# CO-REPRESENTATION NEURAL HYPERGRAPH DIFFUSION FOR EDGE-DEPENDENT NODE CLASSIFICATION

## ABSTRACT

Hypergraphs are widely employed to represent complex higher-order relations in real-world applications. Most hypergraph learning research focuses on node-level or edge-level tasks. A practically relevant but more challenging task, edge-dependent node classification (ENC), is only recently proposed. In ENC, a node can have different labels across different hyperedges, which requires the modeling of node-edge pairs instead of single nodes or hyperedges. Existing solutions for this task are based on message passing and model interactions in within-edge and within-node structures as *multi-input single-output* functions. This brings **three limitations**: (1) non-adaptive representation size, (2) non-adaptive messages, and (3) insufficient direct interactions among nodes or edges. To tackle these limitations, we propose **CoNHD**, a new ENC solution that models both within-edge and within-node interactions as *multi-input multi-output* functions. Specifically, we represent these interactions as a hypergraph diffusion process on node-edge co-representations. We further develop a neural implementation for this diffusion process, which can adapt to a specific ENC dataset. Extensive experiments demonstrate the effectiveness and efficiency of the proposed CoNHD method.

## 1 INTRODUCTION

Real-world applications often involve intricate higher-order relations that cannot be represented by traditional graphs with pairwise connections (Milo et al., 2002; Battiston et al., 2020; Lambiotte et al., 2019; Zhang et al., 2023). Hypergraphs, where an edge can connect more than two nodes, provide a flexible structure to represent these relations (Berge, 1984; Bretto, 2013; Gao et al., 2020; Antelmi et al., 2023). Many hypergraph learning methods are proposed to obtain effective node or edge representations (Liu et al., 2024; Wang et al., 2024; Jo et al., 2021). These methods, however, are insufficient for predicting labels related to node-edge pairs. To initiate the development of effective solutions for such scenarios, Choe et al. (2023) propose a new problem namely *edge-dependent node classification* (ENC), where a node can have different labels across different edges. Addressing this problem requires modeling the node features unique to each edge, which is more complex than other tasks and requires considering the hypergraph structure. The ENC task has valuable real-world applications, such as predicting the score of a player in different matches for the game industry (Choe et al., 2023) or determining the role of a protein in various pathways (Kanehisa et al., 2024). Moreover, the predicted labels from ENC can also serve as additional features for improving performance on downstream tasks (Choe et al., 2023), including ranking aggregation (Chitra & Raphael, 2019), node clustering (Hayashi et al., 2020), product-return prediction (Li et al., 2018), and anomaly detection (Lee et al., 2022). Despite its significant practical value, the ENC task still remains under-explored.

To address hypergraph-related problems (Kim et al., 2024; Saxena et al., 2024; Yan et al., 2024), including the ENC problem (Choe et al., 2023), message passing-based hypergraph neural networks (HGNNs) have become a standard solution (Huang & Yang, 2021; Chien et al., 2022; Arya et al., 2024). Since message passing has various meanings in literature (Gilmer et al., 2017; Kim et al., 2024), to avoid confusion, in this paper, *message passing* refers specifically to the HGNN architecture illustrated in Fig. 1(a), which employs a two-stage aggregation process. The first stage aggregates messages from nodes to update the edge representation, while the second stage aggregates messages from edges to update the node representation. The edge and node representations are then

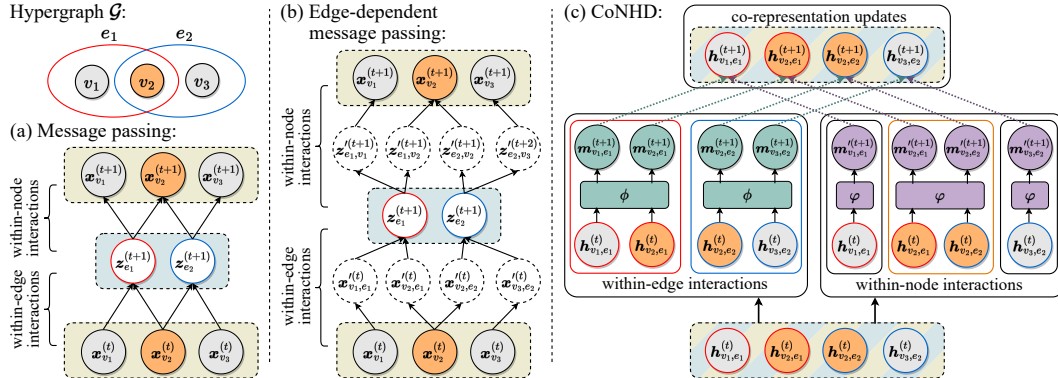

Figure 1: **Different HGNN architectures.** (a,b) The (edge-dependent) message passing framework aggregates (edge-dependent) messages from neighboring nodes to update the edge representation and then from neighboring edges back to update the node representation. (c) Our proposed CoNHD learns a co-representation for each node-edge pair based on diffusion, rather than separate node and edge representations. We define two multi-input multi-output functions $\phi$ and $\varphi$ as the neural implementation of the diffusion operators, which model the within-edge and within-node interactions without aggregation and output diverse diffusion information for different node-edge pairs. The within-edge and within-node diffusion information is then used to update the co-representations.

concatenated and used to predict edge-dependent node labels (Choe et al., 2023). Message passing is simple and intuitive, but does it yield the most effective solution?

To study the characteristics of message passing, we analyze its input-output relations. The two aggregation stages in message passing essentially model interactions in two key types of local structures: *within-edge structures* (different nodes within the same edge) and *within-node structures* (different edges within the same node). Here we treat nodes and edges in a hypergraph as symmetric concepts using hypergraph duality (Scheinerman & Ullman, 2013). In the original hypergraph, nodes are contained within a hyperedge, while in the dual hypergraph, edges (dual nodes) can be similarly viewed as being contained within a node (dual edge). Message passing models interactions in the within-edge and within-node structures as *multi-input single-output* aggregation functions, which brings **three limitations**:

- **Non-adaptive representation size.** When modeling within-node structures, messages from numerous edges are aggregated to a fixed-size node representation vector. This can cause potential information loss for large-degree nodes, which have more neighboring edges and should have larger representation size (Aponte et al., 2022). Since low-degree nodes do not require large representation size, simply increasing the embedding dimension for all nodes is not an effective solution as it not only leads to excessive computational and memory costs, but also introduces challenges like overfitting and optimization difficulties (Luo et al., 2021; Goodfellow et al., 2016). Analogously, when modeling within-edge structures, the same problem exists.

- **Non-adaptive messages.** Since the aggregation process mixes information from different edges to a single node representation and cannot represent specific information for each edge, the node can only pass the same message to the different edges it is part of. However, different edges may focus on different properties of the node and should receive different adaptive messages. Some methods, as shown in Fig. 1(b), attempt to extract edge-dependent information from a single node representation to solve this problem (Aponte et al., 2022; Wang et al., 2023a; Choe et al., 2023). This extraction process requires the model to learn how to recover edge-dependent information from a mixed node representation in each convolution layer, which increases the learning difficulty and may fail to fully recover edge-dependent information.

- **Insufficient direct interactions among nodes or edges.** When modeling within-edge structures, it is crucial to consider direct interactions among nodes to update node representations, rather than solely interactions from nodes to the edge. Similarly, in within-node structures, direct interactions among edges should be considered. These direct nodes-to-nodes and edges-to-edges interactions, which require multiple outputs for different elements, have been shown to benefit hypergraph learning (Pei et al., 2024) and are particularly critical for the ENC task.

Apart from the above three limitations, most message passing-based methods also suffer from the common oversmoothing issue in HGNNs (Wang et al., 2023a; Yan et al., 2024), which hinders the utilization of long-range information and limits the model performance.

To tackle the oversmoothing issue, some diffusion-inspired GNNs and HGNNs are proposed, which demonstrate strong potential for constructing deep models (Wang et al., 2023a; Chamberlain et al., 2021; Thorpe et al., 2022; Gravina et al., 2023). Hypergraph diffusion methods (Liu et al., 2021; Fountoulakis et al., 2021; Veldt et al., 2023) model the information diffusion process from nodes to neighboring nodes in the same hyperedge. These methods, without edge representations, are insufficient for solving the ENC problem. Moreover, the neural implementation of traditional hypergraph diffusion (Wang et al., 2023a) is still following the two-stage message passing framework and suffers from the above three limitations.

To make hypergraph diffusion applicable to the ENC problem, we first extend the concept of hypergraph diffusion by using node-edge co-representations. We show that this diffusion process can model interactions in both within-edge and within-node structures as multi-input multi-output equivariant functions, which can address the three limitations of existing message passing-based ENC solutions. We further propose a neural implementation named **Co**-representation **N**eural **H**ypergraph **D**iffusion (**CoNHD**), which can learn suitable diffusion dynamics from a specific ENC dataset instead of relying on handcrafted regularization functions.

**Our main contributions** are summarized as follows:

1. We define **co-representation hypergraph diffusion**, a new concept that generalizes hypergraph diffusion using node-edge co-representations, which addresses the three major limitations of existing message passing-based ENC solutions.

2. We propose **CoNHD**, a neural implementation for the co-representation hypergraph diffusion process, which in a natural way leads to a novel and effective HGNN architecture that can effectively learn suitable diffusion dynamics from data.

3. We conduct extensive experiments to validate the **effectiveness** and **efficiency** of CoNHD. The results demonstrate that CoNHD achieves the best performance on ten ENC datasets without sacrificing efficiency.

## 2 RELATED WORK

**Hypergraph Neural Networks.** Inspired by the success of graph neural networks (GNNs) (Kipf & Welling, 2017; Wu et al., 2020; 2022), hypergraph neural networks (HGNNs) have been proposed for modeling complex higher-order relations (Kim et al., 2024; Duta et al., 2023). HyperGNN (Feng et al., 2019; Gao et al., 2022) and HCHA (Bai et al., 2021) define hypergraph convolution based on the clique expansion graph. HyperGCN (Yadati et al., 2019) reduces the clique expansion graph into an incomplete graph with mediators. To directly utilize higher-order structures, HNHN (Dong et al., 2020) and HyperSAGE (Arya et al., 2020; 2024) model the convolution layer as a two-stage message passing process, where messages are first aggregated from nodes to edges and then back to nodes. UniGNN (Huang & Yang, 2021) and AllSet (Chien et al., 2022) show that most existing HGNNs can be represented in this two-stage message passing framework. $HDS^{ode}$ improves message passing by modeling it as an ODE-based dynamic system (Yan et al., 2024). Recent research explores edge-dependent message passing, where edge-dependent node messages are extracted before feeding them into the aggregation process (Aponte et al., 2022; Wang et al., 2023a; Telyatnikov et al., 2023). LEGCN (Yang et al., 2022) and MultiSetMixer (Telyatnikov et al., 2023) can generate multiple representations for a single node. However, both model interactions as an aggregation function, which produces the same output for different elements. This limitation leads to significantly reduced performance compared to our multi-output design, as demonstrated in our ablation experiments (Section 5.3). Further discussion of the weaknesses of these two methods compared to our method can be found in Appendix F. While most existing methods focus on node-level or edge-level tasks (Liu et al., 2024; Benko et al., 2024; Chen et al., 2023; Behrouz et al., 2023), the ENC problem remains less explored. Choe et al. (2023) are the first to explore the ENC problem and propose WHATsNet, a solution based on edge-dependent message passing. Different from our method, WHATsNet employs an aggregation after the equivariant operator to produce a single node or edge representation, which still follows the single-output design. Message passing

has become the dominant framework for HGNN research, but it cannot effectively address the ENC problem as it suffers from the three limitations discussed in the introduction.

**(Hyper)graph Diffusion.** (Hyper)graph diffusion (Gleich & Mahoney, 2015; Chamberlain et al., 2021) models the diffusion information as the gradients derived from minimizing a regularized target function, which regularizes the node representations within the same edge. This ensures that the learned node representations converge to the solution of the optimization target instead of an oversmoothed solution (Yang et al., 2021; Thorpe et al., 2022). The technique was first introduced to achieve local and global consistency on graphs (Zhu et al., 2003; Zhou et al., 2003), and was then generalized to hypergraphs (Zhou et al., 2007; Antelmi et al., 2023). Zhou et al. (2007) propose a regularization function by reducing the higher-order structure in a hypergraph using clique expansion. To directly utilize the higher-order structures, Hein et al. (2013) propose a regularization function based on the total variation of the hypergraph. Other regularization functions are designed to improve parallelization ability and introduce non-linearity (Jegelka et al., 2013; Tudisco et al., 2021b;a; Liu et al., 2021). To efficiently calculate the diffusion process when using complex regularization functions, some advanced optimization techniques have been investigated (Zhang et al., 2017; Li et al., 2020). Recently, some works explore the neural implementation of (hyper)graph diffusion processes (Chamberlain et al., 2021; Li et al., 2022; Thorpe et al., 2022; Gravina et al., 2023; Wang et al., 2023a;b), which demonstrate strong robustness against the oversmoothing issue. While hypergraph diffusion methods have shown effectiveness in various tasks like ranking, motif clustering, and signal processing (Li & Milenkovic, 2017; Takai et al., 2020; Zhang et al., 2019; Schaub et al., 2021), they are all restricted to node representations and cannot address the ENC problem.

In this paper, we extend hypergraph diffusion using node-edge co-representations and propose a neural implementation. Most related to our work is ED-HNN (Wang et al., 2023a), which is designed to approximate any traditional hypergraph diffusion process. However, ED-HNN is based on message passing, which still models interactions in within-edge and within-node structures as multi-input single-output aggregation functions and suffers from the three limitations discussed in the introduction. Our method is the first that models both within-edge and within-node interactions as multi-input multi-output functions, which effectively tackles these limitations and demonstrates significant improvements in our experiments.

## 3 PRELIMINARIES

**Notations.** Let $\mathcal{G} = (\mathcal{V}, \mathcal{E})$ denote a hypergraph, where $\mathcal{V} = \{v_1, v_2, \ldots, v_n\}$ represents a set of $n$ nodes, and $\mathcal{E} = \{e_1, e_2, \ldots, e_m\}$ represents a set of $m$ hyperedges. Each edge $e_i \in \mathcal{E}$ is a non-empty subset of $\mathcal{V}$ and can contain an arbitrary number of nodes. $\mathcal{E}_v = \{e \in \mathcal{E} | v \in e\}$ represents the set of edges that contain node $v$, and $d_v = |\mathcal{E}_v|$ and $d_e = |e|$ are the degrees of node $v$ and edge $e$, respectively. We use $v_i^e$ and $e_j^v$ to respectively denote the $i$-th node in edge $e$ and the $j$-th edge in $\mathcal{E}_v$. $\boldsymbol{X}^{(0)} = [\boldsymbol{x}_{v_1}^{(0)}, \ldots, \boldsymbol{x}_{v_n}^{(0)}]^\top$ is the initial node feature matrix.

**Message Passing-based HGNNs.** Message passing (Huang & Yang, 2021; Chien et al., 2022) has become a standard framework for most HGNNs, which models the interactions in within-edge and within-node structures as two multi-input single-output aggregation functions $f_{\mathcal{V} \to \mathcal{E}}$ and $f_{\mathcal{E} \to \mathcal{V}}$:

$$\boldsymbol{z}_e^{(t+1)} = f_{\mathcal{V} \to \mathcal{E}}(\boldsymbol{X}_e^{(t)}; \boldsymbol{z}_e^{(t)}), \tag{1}$$

$$\tilde{\boldsymbol{x}}_v^{(t+1)} = f_{\mathcal{E} \to \mathcal{V}}(\boldsymbol{Z}_v^{(t+1)}; \boldsymbol{x}_v^{(t)}), \tag{2}$$

$$\boldsymbol{x}_v^{(t+1)} = f_{\text{skip}}(\tilde{\boldsymbol{x}}_v^{(t+1)}, \boldsymbol{x}_v^{(t)}, \boldsymbol{x}_v^{(0)}). \tag{3}$$

Here $\boldsymbol{x}_v^{(t)}$ and $\boldsymbol{z}_e^{(t)}$ are the node and edge representations in the $(t)$-th iteration. $\boldsymbol{x}_v^{(0)}$ is the initial node features, and $\boldsymbol{z}_e^{(0)}$ is typically initialized by a zero vector or the average of the feature vectors of the nodes in this edge. $\boldsymbol{X}_e^{(t)}$ denotes the representations of nodes contained in edge $e$, *i.e.*, $\boldsymbol{X}_e^{(t)} = \left[\boldsymbol{x}_{v_1^e}^{(t)}, \ldots, \boldsymbol{x}_{v_{d_e}^e}^{(t)}\right]^\top$. Similarly, $\boldsymbol{Z}_v^{(t)} = \left[\boldsymbol{z}_{e_1^v}^{(t)}, \ldots, \boldsymbol{z}_{e_{d_v}^v}^{(t)}\right]^\top$ denotes the representations of edges containing node $v$. $f_{\text{skip}}$ represents the optional skip connection of the original features, which can help mitigate the oversmoothing issue (Huang & Yang, 2021). $f_{\mathcal{V} \to \mathcal{E}}$ and $f_{\mathcal{E} \to \mathcal{V}}$ take multiple representations from neighboring nodes or edges as inputs, and output a single edge or node representation. These two single-output aggregation functions lead to the three limitations indicated in the introduction.

**Hypergraph Diffusion.** Hypergraph diffusion learns node representations $\boldsymbol{X} = \left[ \boldsymbol{x}_{v_1}, \dots, \boldsymbol{x}_{v_n} \right]^\top$, where $\boldsymbol{x}_{v_i} \in \mathbb{R}^d$, by minimizing a hypergraph-regularized target function (Tudisco et al., 2021a; Prokopchik et al., 2022). For brevity, we use $\boldsymbol{X}_e = \left[ \boldsymbol{x}_{v_1^e}, \dots, \boldsymbol{x}_{v_{d_e}^e} \right]^\top$ to denote the representations of nodes contained in the edge $e$. The target function is the weighted summation of some non-structural and structural regularization functions. The non-structural regularization function is independent of the hypergraph structure, which is typically defined as a squared loss function based on the node attribute vector $\boldsymbol{a}_v$ (composed of initial node features $\boldsymbol{x}_v^{(0)}$ (Takai et al., 2020) or observed node labels (Tudisco et al., 2021a)). The structural regularization functions incorporate the hypergraph structure and apply regularization to multiple node representations within the same hyperedge. Many structural regularization functions are designed by heuristics (Zhou et al., 2007; Hein et al., 2013; Hayhoe et al., 2023; Tudisco et al., 2021b). For instance, the clique expansion (CE) regularization functions (Zhou et al., 2007), defined as $\Omega_{\mathrm{CE}}(\boldsymbol{X}_e) := \sum_{v,u \in e} \|\boldsymbol{x}_v - \boldsymbol{x}_u\|_2^2$, encourages the representations of all nodes in an edge to become similar.

**Definition 1** (Node-Representation Hypergraph Diffusion). *Given a non-structural regularization function $\mathcal{R}_v(\cdot; \boldsymbol{a}_v) : \mathbb{R}^d \to \mathbb{R}$ and a structural regularization function $\Omega_e(\cdot) : \mathbb{R}^{d_e \times d} \to \mathbb{R}$, the node-representation hypergraph diffusion learns representations by solving the following optimization problem*

$$\boldsymbol{X}^\star = \arg\min_{\boldsymbol{X}} \left\{ \sum_{v \in \mathcal{V}} \mathcal{R}_v(\boldsymbol{x}_v; \boldsymbol{a}_v) + \lambda \sum_{e \in \mathcal{E}} \Omega_e(\boldsymbol{X}_e) \right\}. \tag{4}$$

Here $\Omega_e(\cdot)$ is also referred to as the edge regularization function. $\boldsymbol{X}^\star$ denotes the matrix of all learned node representations, which can be used for predicting the node labels.

## 4 METHODOLOGY

Our goal is to provide a new HGNN framework that views the within-edge and within-node interactions from a multi-input multi-output perspective, which can address the three limitations discussed in the introduction. To achieve this goal, we need to:

**(1) Redefine the inputs and outputs.** Existing message passing-based HGNNs treat the single edge or node representation as the output for the within-edge or within-node interactions, respectively. A new kind of representations is needed to disentangle the single output to multiple outputs.

**(2) Redefine the interaction process.** An expressive enough and learnable function is needed to model the interactions from inputs to outputs, while considering the symmetry in hypergraph data.

In Section 4.1, we introduce co-representations into hypergraph diffusion to meet requirement (1). We further show that this extension naturally satisfies part of requirement (2) by preserving the permutation equivariance in hypergraphs. In Section 4.2, we present a learnable neural implementation of the diffusion process. We carefully design the architecture to ensure both requirements are satisfied, leading to the novel HGNN framework demonstrated in Fig. 1 (c).

### 4.1 CO-REPRESENTATION HYPERGRAPH DIFFUSION

In this section, we extend the current hypergraph diffusion concept using co-representations of node-edge pairs. This extension not only enables the application of hypergraph diffusion in addressing the ENC problem, but also disentangles the mixed node/edge representations into fine grained co-representations to address the three limitations discussed in the introduction. Let us first formally introduce the ENC problem.

**Problem 1** (Edge-Dependent Node Classification (ENC) (Choe et al., 2023)). *Given (1) a hypergraph $\mathcal{G} = (\mathcal{V}, \mathcal{E})$, (2) observed edge-dependent node labels $y_{v,e}$ for $\mathcal{E}' \subset \mathcal{E}$ ($\forall v \in e, \forall e \in \mathcal{E}'$), and (3) an initial node feature matrix $\boldsymbol{X}^{(0)}$, the ENC problem is to predict the unobserved edge-dependent node labels $y_{v,e}$ for $\mathcal{E} \setminus \mathcal{E}'$ ($\forall v \in e, \forall e \in \mathcal{E} \setminus \mathcal{E}'$).*

In ENC, the label $y_{v,e}$ is associated with both the node $v$ and the edge $e$. We extend hypergraph diffusion to learn a co-representation $\boldsymbol{h}_{v,e} \in \mathbb{R}^d$ for each node-edge pair $(v, e)$. We name this co-representation hypergraph diffusion. Let $\boldsymbol{H} = [\dots, \boldsymbol{h}_{v,e}, \dots]^\top$ denote the collection of all co-

representation vectors. We use $\boldsymbol{H}_e = [\boldsymbol{h}_{v_1^e,e}, \ldots, \boldsymbol{h}_{v_{d_e}^e,e}]^\top$ and $\boldsymbol{H}_v = [\boldsymbol{h}_{v,e_1^v}, \ldots, \boldsymbol{h}_{v,e_{d_v}^v}]^\top$ to represent the co-representations associated with an edge $e$ or a node $v$, respectively.

**Definition 2** (Co-Representation Hypergraph Diffusion). *Given a non-structural regularization function $\mathcal{R}_{v,e}(\cdot; \boldsymbol{a}_{v,e}) : \mathbb{R}^d \to \mathbb{R}$, structural regularization functions $\Omega_e(\cdot) : \mathbb{R}^{d_e \times d} \to \mathbb{R}$ and $\Omega_v(\cdot) : \mathbb{R}^{d_v \times d} \to \mathbb{R}$, the co-representation hypergraph diffusion learns node-edge co-representations by solving the following optimization problem*

$$\boldsymbol{H}^\star = \arg\min_{\boldsymbol{H}} \left\{ \sum_{v \in \mathcal{V}} \sum_{e \in \mathcal{E}_v} \mathcal{R}_{v,e}(\boldsymbol{h}_{v,e}; \boldsymbol{a}_{v,e}) + \lambda \sum_{e \in \mathcal{E}} \Omega_e(\boldsymbol{H}_e) + \gamma \sum_{v \in \mathcal{V}} \Omega_v(\boldsymbol{H}_v) \right\}. \tag{5}$$

Here $\mathcal{R}_{v,e}(\cdot; \boldsymbol{a}_{v,e})$ is independent of the hypergraph structure, where $\boldsymbol{a}_{v,e}$ can be any related attributes of the node-edge pair $(v, e)$ (*e.g.*, node features, edge features, or observed edge-dependent node labels). $\Omega_e(\cdot)$ and $\Omega_v(\cdot)$ are referred to as the edge and node regularization functions, respectively. They apply regularization to co-representations associated with the same node or edge, which can be implemented as the structural regularization functions designed for traditional node-representation hypergraph diffusion (Zhou et al., 2007; Hein et al., 2013; Hayhoe et al., 2023).

Depending on whether the regularization functions are differentiable, we can solve Eq. 5 using one of two standard optimization methods: gradient descent (GD) or alternating direction method of multipliers (ADMM) (Boyd et al., 2011). When the regularization functions are differentiable, we initialize $\boldsymbol{h}_{v,e}^{(0)} = \boldsymbol{a}_{v,e}$, and then solve it using GD with a step size $\alpha$:

$$\boldsymbol{h}_{v,e}^{(t+1)} = \boldsymbol{h}_{v,e}^{(t)} - \alpha(\nabla \mathcal{R}_{v,e}(\boldsymbol{h}_{v,e}^{(t)}; \boldsymbol{a}_{v,e}) + \lambda[\nabla \Omega_e(\boldsymbol{H}_e^{(t)})]_v + \gamma[\nabla \Omega_v(\boldsymbol{H}_v^{(t)})]_e), \tag{6}$$

where $\nabla$ is the gradient operator. $[\cdot]_v$ and $[\cdot]_e$ represent the gradient vector associated with node $v$ and edge $e$, respectively. For example, $[\nabla \Omega_e(\boldsymbol{H}_e^{(t)})]_v$ represents the gradient w.r.t. $\boldsymbol{h}_{v,e}^{(t)}$.

When the regularization functions are not all differentiable, we can apply ADMM with the proximity term $\mathbf{prox}_{\lambda \Omega_e / \rho}(\cdot)$ to find the optimal solution (see Appendix J for the details).

Similar to traditional hypergraph diffusion, we refer to $\nabla \Omega_e(\cdot)$ and $\mathbf{prox}_{\lambda \Omega_e / \rho}(\cdot)$ in the GD or ADMM method as *edge diffusion operators*, which model interactions in within-edge structures and generate information that should "diffuse" to each node-edge pair. $\nabla \Omega_v(\cdot)$ and $\mathbf{prox}_{\gamma \Omega_v / \rho}(\cdot)$ are referred to as *node diffusion operators*.

The edges and nodes are inherently unordered, hence in designing structural regularization functions it is important to ensure the outputs are consistent regardless of the input ordering. We say a function $g : \mathbb{R}^{n \times d} \to \mathbb{R}^{d'}$ is permutation invariant, if for any action $\pi$ from the row permutation group $\mathbb{S}_n$, the relation $g(\pi \cdot \boldsymbol{I}) = g(\boldsymbol{I})$ holds for any input matrix $\boldsymbol{I} \in \mathbb{R}^{n \times d}$. Similarly, function $g : \mathbb{R}^{n \times d} \to \mathbb{R}^{n \times d'}$ is permutation equivariant, if for any $\pi \in \mathbb{S}_n$, the relation $g(\pi \cdot \boldsymbol{I}) = \pi \cdot g(\boldsymbol{I})$ holds for all $\boldsymbol{I} \in \mathbb{R}^{n \times d}$. In traditional hypergraph diffusion, the diffusion operators derived from invariant regularization functions have been proven to be permutation equivariant (Wang et al., 2023a).

**Proposition 1** (Wang et al. (2023a)). *With permutation invariant structural regularization functions, the diffusion operators are permutation equivariant.*

Since the edge and node regularization functions are defined as structural regularization functions in traditional hypergraph diffusion, Proposition 1 applies to both the edge and node diffusion operators in our co-representation hypergraph diffusion as well. This critical property shows that our co-representation hypergraph diffusion process models the complex interactions in within-edge and within-node structures as multi-input multi-output equivariant functions, while ensuring the outputs commute according to the input ordering.

Next, we state the relation between the proposed co-representation hypergraph diffusion and the node-representation hypergraph diffusion.

**Proposition 2.** *The traditional node-representation hypergraph diffusion is a special case of the co-representation hypergraph diffusion, while the opposite is not true.*

We leave all the proofs to Appendix B. Node-representation hypergraph diffusion is equivalent to imposing a strict constraint that all the co-representations associated with the same node must be identical, resulting in a single unified node representation. We relax this constraint by incorporating node regularization functions into the optimization objective, allowing multiple co-representations associated with the same node to differ while still being constrained by certain regularization terms.

## 4.2 NEURAL IMPLEMENTATION

Traditional hypergraph diffusion relies on handcrafting structural regularization functions, which requires good insights in the dataset. In this section, we propose **Co**-representation **N**eural **H**ypergraph **D**iffusion (**CoNHD**), which is a neural implementation of the diffusion process and can easily adapt to a specific dataset. This implementation leads to the novel HGNN architecture illustrated in Fig. 1(c).

We provide a GD-based implementation of our model architecture following the update rules in Eq. 6. The $(t + 1)$-th layer can be represented as:

$$\texttt{GD-based:} \qquad \boldsymbol{M}_e^{(t+1)} = \phi(\boldsymbol{H}_e^{(t)}), \ \boldsymbol{M}_v'^{(t+1)} = \varphi(\boldsymbol{H}_v^{(t)}), \qquad (7)$$

$$\boldsymbol{h}_{v,e}^{(t+1)} = \psi([\boldsymbol{h}_{v,e}^{(t)}, \boldsymbol{m}_{v,e}^{(t+1)}, \boldsymbol{m}_{v,e}'^{(t+1)}, \boldsymbol{h}_{v,e}^{(0)}]), \qquad (8)$$

Here $\boldsymbol{M}_e^{(t)} = [\boldsymbol{m}_{v_1^e,e}^{(t)}, \dots, \boldsymbol{m}_{v_{d_e}^e,e}^{(t)}]^\top$ and $\boldsymbol{M}_v'^{(t)} = [\boldsymbol{m}_{v,e_1^v}'^{(t)}, \dots, \boldsymbol{m}_{v,e_{d_v}^v}'^{(t)}]$ are the within-edge and within-node diffusion information generated using the neural diffusion operators $\phi$ and $\varphi$, which can be implemented by any permutation equivariant network. $\psi(\cdot)$ is implemented as a linear layer, which collects diffusion information and updates the co-representations. $\boldsymbol{h}_{v,e}^{(0)}$ is the initial feature vector, which corresponds to the non-structural regularization term in Eq. 5. We provide the ADMM-based implementation in Appendix J.

According to Proposition 1, $\phi$ and $\varphi$ should satisfy the permutation equivariance property. Previous research only models the composition of within-edge and within-node interactions as an equivariant function, while each interaction is still an invariant aggregation function (Wang et al., 2023a). Although WHATsNet (Choe et al., 2023) utilizes the equivariant module in both interactions, the multiple outputs serve only as an intermediate results, with an aggregation module applied at the end. As a result, the composition is still an invariant aggregation function and only a single node or edge representation is updated in this process. In contrast, our method removes the unnecessary aggregation process and is the first to model interactions in both within-edge and within-node structures as two distinct equivariant functions, which output different information to update multiple co-representations. Our ablation experiments in Section 5.3 show the effectiveness of this design.

We explore two popular equivariant neural networks, UNB (Segol & Lipman, 2020; Wang et al., 2023a) and ISAB (Chien et al., 2022), for the implementation of the diffusion operators $\phi$ and $\varphi$. The details can be found in Appendix C. Apart from these two equivariant networks explored in our experiments, it is worth noting that our proposed CoNHD is a general HGNN architecture, where the neural diffusion operators can be implemented as any other equivariant network.

To demonstrate the expressiveness of CoNHD, we compare it with the message passing framework defined in Eq. 1-3, which can cover most existing HGNNs (Huang & Yang, 2021; Chien et al., 2022). Since the message passing framework can only generate separate representations for nodes and edges, following (Choe et al., 2023), we regard the concatenation of node and edge representations as the final embeddings, which can be used to predict edge-dependent node labels.

**Proposition 3.** *With the same embedding dimension, CoNHD is expressive enough to represent the message passing framework, while the opposite is not true.*

Proposition 3 demonstrates that CoNHD is more expressive than all methods following the message passing framework. Notably, despite the increased expressiveness, the complexity of CoNHD is still linear to the number of node-edge pairs, *i.e.*, $\sum_{e \in \mathcal{E}} d_e$, which is the same as message passing-based methods. We provide theoretical complexity analysis in Appendix D.1.

## 5 EXPERIMENTS

In this section, we present experiments to evaluate the effectiveness and efficiency of the proposed CoNHD method for predicting edge-dependent node labels, as well as to assess the impact of the critical multi-output design. Additional experiments are provided in Appendix I, where we further validate the performance on other tasks including downstream tasks and the traditional node classification task, examine its capacity to mitigate the oversmoothing issue in constructing deep models, and explore the benefits of the direct interactions among nodes and edges.

Table 1: **Performance of edge-dependent node classification. Bold** numbers represent the best results, while underlined numbers indicate the second-best. "O.O.M." means "out of memory". Shaded cells indicate that our method significantly outperforms the best baseline (p-value $< 0.05$, based on the Wilcoxon signed-rank test). "A.R." denotes the average ranking among all datasets.

| Method | Email-Enron Micro-F1 | Email-Enron Macro-F1 | Email-Eu Micro-F1 | Email-Eu Macro-F1 | Stack-Biology Micro-F1 | Stack-Biology Macro-F1 | Stack-Physics Micro-F1 | Stack-Physics Macro-F1 | Coauth-DBLP Micro-F1 | Coauth-DBLP Macro-F1 | A.R. of Micro-F1 |
|---|---|---|---|---|---|---|---|---|---|---|---|
| GraphSAGE | $0.775 \pm 0.005$ | $0.714 \pm 0.007$ | $0.658 \pm 0.001$ | $0.564 \pm 0.005$ | $0.689 \pm 0.010$ | $0.598 \pm 0.014$ | $0.660 \pm 0.011$ | $0.523 \pm 0.018$ | $0.474 \pm 0.002$ | $0.401 \pm 0.008$ | 12.3 |
| GAT | $0.736 \pm 0.056$ | $0.611 \pm 0.103$ | $0.618 \pm 0.002$ | $0.580 \pm 0.024$ | $0.692 \pm 0.015$ | $0.628 \pm 0.010$ | $0.725 \pm 0.024$ | $0.636 \pm 0.043$ | $0.575 \pm 0.005$ | $0.558 \pm 0.007$ | 8.6 |
| ADGN | $0.790 \pm 0.001$ | $0.723 \pm 0.001$ | $0.667 \pm 0.001$ | $0.622 \pm 0.006$ | $0.714 \pm 0.002$ | $0.651 \pm 0.001$ | $0.686 \pm 0.014$ | $0.537 \pm 0.019$ | $0.505 \pm 0.006$ | $0.440 \pm 0.020$ | 9.1 |
| HyperGNN | $0.725 \pm 0.004$ | $0.674 \pm 0.003$ | $0.633 \pm 0.001$ | $0.533 \pm 0.008$ | $0.689 \pm 0.002$ | $0.624 \pm 0.007$ | $0.686 \pm 0.004$ | $0.630 \pm 0.002$ | $0.540 \pm 0.004$ | $0.519 \pm 0.002$ | 10.5 |
| HNHN | $0.738 \pm 0.028$ | $0.637 \pm 0.023$ | $0.643 \pm 0.004$ | $0.552 \pm 0.014$ | $0.640 \pm 0.005$ | $0.592 \pm 0.006$ | $0.506 \pm 0.053$ | $0.422 \pm 0.043$ | $0.486 \pm 0.004$ | $0.478 \pm 0.008$ | 11.5 |
| HCHA | $0.666 \pm 0.010$ | $0.464 \pm 0.002$ | $0.620 \pm 0.000$ | $0.497 \pm 0.001$ | $0.589 \pm 0.007$ | $0.465 \pm 0.060$ | $0.622 \pm 0.003$ | $0.481 \pm 0.007$ | $0.451 \pm 0.007$ | $0.334 \pm 0.048$ | 15.9 |
| HAT | $0.817 \pm 0.001$ | $0.753 \pm 0.004$ | $0.669 \pm 0.001$ | $0.638 \pm 0.002$ | $0.661 \pm 0.005$ | $0.606 \pm 0.005$ | $0.708 \pm 0.005$ | $0.643 \pm 0.009$ | $0.503 \pm 0.004$ | $0.483 \pm 0.006$ | 8.0 |
| UniGCNII | $0.734 \pm 0.010$ | $0.656 \pm 0.010$ | $0.630 \pm 0.005$ | $0.565 \pm 0.013$ | $0.610 \pm 0.004$ | $0.433 \pm 0.007$ | $0.671 \pm 0.022$ | $0.492 \pm 0.016$ | $0.497 \pm 0.003$ | $0.476 \pm 0.002$ | 14.4 |
| AllSet | $0.796 \pm 0.014$ | $0.719 \pm 0.020$ | $0.666 \pm 0.005$ | $0.624 \pm 0.021$ | $0.571 \pm 0.054$ | $0.446 \pm 0.081$ | $0.728 \pm 0.039$ | $0.646 \pm 0.046$ | $0.495 \pm 0.038$ | $0.487 \pm 0.040$ | 9.4 |
| HDS$^{ode}$ | $0.805 \pm 0.001$ | $0.740 \pm 0.006$ | $0.651 \pm 0.000$ | $0.577 \pm 0.001$ | $0.708 \pm 0.001$ | $0.643 \pm 0.004$ | $0.737 \pm 0.001$ | $0.635 \pm 0.008$ | $0.558 \pm 0.001$ | $0.550 \pm 0.002$ | 7.6 |
| LEGCN | $0.783 \pm 0.001$ | $0.728 \pm 0.007$ | $0.639 \pm 0.001$ | $0.535 \pm 0.004$ | $0.668 \pm 0.002$ | $0.572 \pm 0.006$ | $0.701 \pm 0.003$ | $0.575 \pm 0.018$ | $0.499 \pm 0.003$ | $0.490 \pm 0.002$ | 7.6 |
| MultiSetMixer | $0.818 \pm 0.001$ | $0.755 \pm 0.005$ | $0.670 \pm 0.001$ | $0.636 \pm 0.005$ | $0.709 \pm 0.001$ | $0.643 \pm 0.003$ | $0.754 \pm 0.001$ | $0.679 \pm 0.004$ | $0.559 \pm 0.001$ | $0.554 \pm 0.001$ | 6.2 |
| HNN | $0.763 \pm 0.003$ | $0.679 \pm 0.007$ | O.O.M. | O.O.M. | $0.618 \pm 0.015$ | $0.568 \pm 0.013$ | $0.683 \pm 0.005$ | $0.617 \pm 0.005$ | $0.488 \pm 0.006$ | $0.482 \pm 0.006$ | 13.2 |
| ED-HNN | $0.778 \pm 0.001$ | $0.713 \pm 0.004$ | $0.648 \pm 0.001$ | $0.558 \pm 0.004$ | $0.688 \pm 0.005$ | $0.606 \pm 0.002$ | $0.726 \pm 0.002$ | $0.617 \pm 0.006$ | $0.514 \pm 0.016$ | $0.484 \pm 0.024$ | 9.6 |
| WHATsNet | $0.826 \pm 0.001$ | $0.761 \pm 0.003$ | $0.671 \pm 0.000$ | $0.645 \pm 0.003$ | $0.742 \pm 0.002$ | $0.685 \pm 0.003$ | $0.770 \pm 0.003$ | $0.707 \pm 0.004$ | $0.604 \pm 0.003$ | $0.592 \pm 0.004$ | 5.3 |
| CoNHD (UNB) (*ours*) | $0.905 \pm 0.001$ | $0.858 \pm 0.004$ | $0.708 \pm 0.001$ | $0.689 \pm 0.001$ | $0.748 \pm 0.003$ | $0.694 \pm 0.005$ | $0.776 \pm 0.001$ | $\mathbf{0.712 \pm 0.005}$ | $\mathbf{0.620 \pm 0.002}$ | $\mathbf{0.604 \pm 0.002}$ | 1.9 |
| CoNHD (ISAB) (*ours*) | $\mathbf{0.911 \pm 0.001}$ | $\mathbf{0.871 \pm 0.002}$ | $\mathbf{0.709 \pm 0.001}$ | $\mathbf{0.690 \pm 0.002}$ | $\mathbf{0.749 \pm 0.002}$ | $\mathbf{0.695 \pm 0.004}$ | $\mathbf{0.777 \pm 0.001}$ | $0.710 \pm 0.001$ | $0.619 \pm 0.002$ | $\mathbf{0.604 \pm 0.003}$ | 1.1 |

| Method | Coauth-AMiner Micro-F1 | Coauth-AMiner Macro-F1 | Cora-Outsider Micro-F1 | Cora-Outsider Macro-F1 | DBLP-Outsider Micro-F1 | DBLP-Outsider Macro-F1 | Citeseer-Outsider Micro-F1 | Citeseer-Outsider Macro-F1 | Pubmed-Outsider Micro-F1 | Pubmed-Outsider Macro-F1 | A.R. of Macro-F1 |
|---|---|---|---|---|---|---|---|---|---|---|---|
| GraphSAGE | $0.441 \pm 0.013$ | $0.398 \pm 0.012$ | $0.520 \pm 0.009$ | $0.518 \pm 0.007$ | $0.490 \pm 0.029$ | $0.427 \pm 0.083$ | $0.704 \pm 0.005$ | $0.704 \pm 0.005$ | $0.677 \pm 0.003$ | $0.663 \pm 0.002$ | 12.5 |
| GAT | $0.623 \pm 0.006$ | $0.608 \pm 0.009$ | $0.531 \pm 0.009$ | $0.521 \pm 0.008$ | $0.563 \pm 0.003$ | $0.548 \pm 0.003$ | $0.704 \pm 0.011$ | $0.702 \pm 0.011$ | $0.677 \pm 0.003$ | $0.670 \pm 0.002$ | 8.3 |
| ADGN | $0.452 \pm 0.009$ | $0.415 \pm 0.014$ | $0.533 \pm 0.007$ | $0.524 \pm 0.005$ | $0.559 \pm 0.005$ | $0.548 \pm 0.001$ | $0.706 \pm 0.008$ | $0.705 \pm 0.008$ | $0.669 \pm 0.003$ | $0.667 \pm 0.002$ | 9.6 |
| HyperGNN | $0.566 \pm 0.002$ | $0.551 \pm 0.004$ | $0.532 \pm 0.015$ | $0.528 \pm 0.013$ | $0.571 \pm 0.005$ | $0.566 \pm 0.005$ | $0.696 \pm 0.006$ | $0.696 \pm 0.006$ | $0.658 \pm 0.003$ | $0.654 \pm 0.002$ | 9.9 |
| HNHN | $0.520 \pm 0.002$ | $0.514 \pm 0.002$ | $0.539 \pm 0.016$ | $0.535 \pm 0.015$ | $0.581 \pm 0.001$ | $0.580 \pm 0.001$ | $0.694 \pm 0.017$ | $0.693 \pm 0.016$ | $0.674 \pm 0.004$ | $0.670 \pm 0.004$ | 11.0 |
| HCHA | $0.468 \pm 0.020$ | $0.447 \pm 0.040$ | $0.505 \pm 0.009$ | $0.445 \pm 0.058$ | $0.542 \pm 0.007$ | $0.509 \pm 0.018$ | $0.622 \pm 0.038$ | $0.620 \pm 0.037$ | $0.655 \pm 0.002$ | $0.648 \pm 0.002$ | 15.9 |
| HAT | $0.543 \pm 0.002$ | $0.533 \pm 0.003$ | $0.548 \pm 0.015$ | $0.544 \pm 0.017$ | $0.588 \pm 0.002$ | $0.586 \pm 0.002$ | $0.691 \pm 0.018$ | $0.690 \pm 0.019$ | $0.676 \pm 0.003$ | $0.673 \pm 0.003$ | 6.9 |
| UniGCNII | $0.520 \pm 0.001$ | $0.507 \pm 0.001$ | $0.519 \pm 0.019$ | $0.509 \pm 0.023$ | $0.540 \pm 0.004$ | $0.537 \pm 0.006$ | $0.674 \pm 0.014$ | $0.671 \pm 0.023$ | $0.621 \pm 0.006$ | $0.617 \pm 0.006$ | 14.2 |
| AllSet | $0.577 \pm 0.005$ | $0.570 \pm 0.002$ | $0.523 \pm 0.018$ | $0.502 \pm 0.016$ | $0.585 \pm 0.008$ | $0.515 \pm 0.013$ | $0.686 \pm 0.010$ | $0.681 \pm 0.009$ | $0.679 \pm 0.006$ | $0.660 \pm 0.010$ | 10.7 |
| HDS$^{ode}$ | $0.561 \pm 0.003$ | $0.552 \pm 0.003$ | $0.537 \pm 0.009$ | $0.529 \pm 0.010$ | $0.554 \pm 0.004$ | $0.548 \pm 0.002$ | $0.703 \pm 0.008$ | $0.703 \pm 0.008$ | $0.669 \pm 0.004$ | $0.664 \pm 0.005$ | 7.4 |
| LEGCN | $0.520 \pm 0.002$ | $0.511 \pm 0.002$ | $0.698 \pm 0.008$ | $0.689 \pm 0.008$ | $0.676 \pm 0.012$ | $0.675 \pm 0.016$ | $0.733 \pm 0.015$ | $0.731 \pm 0.016$ | $0.703 \pm 0.002$ | $0.698 \pm 0.002$ | 7.7 |
| MultiSetMixer | $0.593 \pm 0.005$ | $0.585 \pm 0.005$ | $0.542 \pm 0.013$ | $0.538 \pm 0.011$ | $0.561 \pm 0.044$ | $0.552 \pm 0.035$ | $0.706 \pm 0.007$ | $0.705 \pm 0.007$ | $0.668 \pm 0.001$ | $0.666 \pm 0.001$ | 5.8 |
| HNN | $0.543 \pm 0.002$ | $0.533 \pm 0.002$ | $0.522 \pm 0.008$ | $0.354 \pm 0.008$ | $0.527 \pm 0.006$ | $0.409 \pm 0.083$ | $0.527 \pm 0.028$ | $0.436 \pm 0.094$ | $0.673 \pm 0.006$ | $0.668 \pm 0.006$ | 12.8 |
| ED-HNN | $0.503 \pm 0.006$ | $0.479 \pm 0.008$ | $0.532 \pm 0.011$ | $0.511 \pm 0.014$ | $0.599 \pm 0.002$ | $0.559 \pm 0.013$ | $0.709 \pm 0.007$ | $0.709 \pm 0.007$ | $0.668 \pm 0.008$ | $0.656 \pm 0.009$ | 11.6 |
| WHATsNet | $0.632 \pm 0.004$ | $0.625 \pm 0.006$ | $0.526 \pm 0.014$ | $0.519 \pm 0.014$ | $0.587 \pm 0.014$ | $0.582 \pm 0.008$ | $0.711 \pm 0.010$ | $0.710 \pm 0.009$ | $0.677 \pm 0.004$ | $0.670 \pm 0.004$ | 5.0 |
| CoNHD (UNB) (*ours*) | $0.646 \pm 0.003$ | $0.640 \pm 0.004$ | $0.769 \pm 0.028$ | $0.767 \pm 0.028$ | $0.884 \pm 0.011$ | $0.883 \pm 0.011$ | $0.827 \pm 0.013$ | $\mathbf{0.826 \pm 0.013}$ | $0.896 \pm 0.003$ | $0.895 \pm 0.003$ | 1.8 |
| CoNHD (ISAB) (*ours*) | $\mathbf{0.650 \pm 0.003}$ | $\mathbf{0.646 \pm 0.004}$ | $\mathbf{0.800 \pm 0.019}$ | $\mathbf{0.797 \pm 0.020}$ | $\mathbf{0.903 \pm 0.002}$ | $\mathbf{0.902 \pm 0.002}$ | $\mathbf{0.828 \pm 0.010}$ | $\mathbf{0.826 \pm 0.010}$ | $\mathbf{0.899 \pm 0.004}$ | $\mathbf{0.898 \pm 0.004}$ | 1.1 |

## 5.1 EFFECTIVENESS AND EFFICIENCY ON THE ENC TASK

**Datasets.** We conduct experiments on ten ENC datasets, with detailed descriptions and statistics provided in Appendix G. These datasets include all six datasets in (Choe et al., 2023), which are Email (`Email-Enron` and `Email-Eu`), StackOverflow (`Stack-Biology` and `Stack-Physics`), and Co-authorship networks (`Coauth-DBLP` and `Coauth-AMiner`). Notably, `Email-Enron` and `Email-Eu` have relatively large node degrees, while `Email-Enron` has relatively large edge degrees as well. Additionally, as real-world hypergraph structures typically contain noise (Cai et al., 2022), to examine the model performance on such scenarios, four newly introduced datasets (`Cora-Outsider`, `DBLP-Outsider`, `Citeseer-Outsider`, and `Pubmed-Outsider`) are derived by transforming the outsider identification problem (Zhang et al., 2020) into the ENC problem. In these datasets, we randomly replace half of the nodes in each edge with other nodes, and the task is to predict whether each node belongs to the corresponding edge.

**Baselines.** We compare our GD-based CoNHD model to ten baseline HGNN methods. For CoNHD, we compare two variants with different neural diffusion operator implementations, UNB (Eq. A3) and ISAB (Eq. A4). The HGNN baselines include seven models following the traditional message passing framework (HyperGNN (Feng et al., 2019), HNHN (Dong et al., 2020), HCHA (Bai et al., 2021), HAT (Hwang et al., 2021), UniGCNII (Huang & Yang, 2021), AllSet (Chien et al., 2022), and HDS$^{ode}$ (Yan et al., 2024)) and five models that utilize edge-dependent node information (LEGCN (Yang et al., 2022), MultiSetMixer (Telyatnikov et al., 2023), HNN (Aponte et al., 2022), ED-HNN (Wang et al., 2023a), and WHATsNet (Choe et al., 2023))Since a hypergraph can also be viewed as a bipartite graph with ENC labels on the new edges, we add three traditional GNN methods (GraphSAGE (Hamilton et al., 2017), GAT (Veličković et al., 2018), and a graph diffusion-based method ADGN (Gravina et al., 2023)) as our baselines.

**Effectiveness.** As shown in Table 1, CoNHD consistently achieves the best performance across all datasets in terms of both Micro-F1 and Macro-F1 metrics. Notably, CoNHD shows very significant improvements on the `Email-Enron` and `Email-Eu` datasets. As indicated before, the main difference between these two datasets and the others is that they have relatively large-degree nodes or edges. All the baseline methods based on single node or edge representations can easily cause

potential information loss for large degree nodes or edges in the aggregation process. In contrast, the number of co-representations in CoNHD is adaptive to the node and edge degrees.

Additionally, CoNHD achieves very significant improvements on the four outsider identification datasets, while GNN methods and message passing-based methods fail to identify these outsiders. This suggests that mixing information from these noise outsiders into a single edge representation significantly degrades the performance of message passing-based methods. Our method, with the co-representation design, can distinguish information from normal nodes and outsiders, thereby achieving superior performance. While on some simple datasets with very low node and edge degrees (see Table A1), such as `Stack-Physics`, the performance improvement is less pronounced compared to other datasets. In these datasets, each hyperedge only contains a very limited number of nodes (about 2 on average, similar to normal graphs), which is relatively simple and cannot fully demonstrate the ability of different HGNNs in modeling complex higher-order interactions. Nevertheless, our method still consistently achieves the best performance on these datasets, and the improvement is statistically significant with a p-value less than 0.05 in most cases.

The performance gap between the two neural diffusion operator implementations is minimal. While theoretically the UNB implementation can approximate any equivariant functions, the ISAB implementation overall demonstrates better performance in our experiments. This might be attributed to the practical effectiveness of the self-attention mechanism.

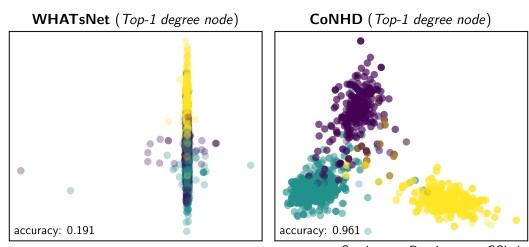

Figure 2: **Visualization of embeddings in the `Email-Enron` dataset using LDA.** The embeddings learned by CoNHD exhibit clearer distinctions based on the edge-dependent labels compared to the embeddings learned by WHATsNet.

To demonstrate whether the model can learn separable embeddings for the same node, similar to (Choe et al., 2023), we use LDA to visualize the embeddings associated with the largest-degree node in the `Email-Enron` dataset. As shown in Fig. 2, CoNHD can learn more separable embeddings than WHATsNet. We show more examples in Appendix I.4.

**Efficiency.** The performance and training time on `Email-Enron` and `Email-Eu` are illustrated in Fig. 3. All experiments are conducted on a single NVIDIA A100 GPU. Only models using mini-batch training are considered in the comparison. Some methods are excluded as their implementation is based on full-batch training, which is impractical when handling large real-world hypergraphs. The overall best baseline, WHATsNet, sacrifices efficiency to improve performance. In contrast, our proposed method, CoNHD, not only achieves the best performance but also maintains high efficiency. In each layer, CoNHD only incorporates

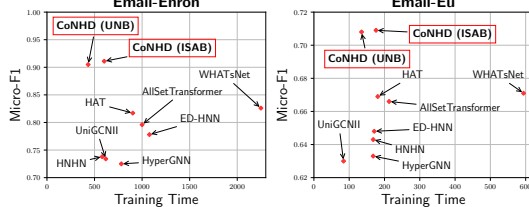

Figure 3: **Comparison of the performance and training time (minutes).** CoNHD demonstrates significant improvements in terms of Micro-F1 while maintaining good efficiency. The same conclusion holds for Marco-F1 (results not shown).

rates direct neighbors of the node-edge pairs, which can reduce the computational costs compared to message passing. Additionally, our method avoids the unnecessary extraction and aggregation process in edge-dependent message passing methods, which not only increases the expressiveness but also improves the efficiency. We provide more analysis in Appendix D.2.

## 5.2 Approximation of Co-representation Hypergraph Diffusion Processes

To validate whether CoNHD, as a neural implementation for co-representation hypergraph diffusion, can effectively approximate the diffusion processes, we conduct experiments on semi-synthetic diffusion datasets using common regularization functions.

**Setup.** We use the `Senate` (Fowler, 2006b) dataset with 1-dimensional feature initialization (Wang et al., 2023a). We perform the co-representation hypergraph diffusion process using three common structural regularization functions: `CE` (Zhou et al., 2007), `TV` (Hein et al., 2013; Hayhoe et al., 2023), and `LEC` (Jegelka et al., 2013; Veldt et al., 2023). More details can be found in Appendix H.

We compare both ADMM-based (Eq. A8-A10) and GD-based (Eq. 7-8) CoNHD model (with ISAB operator) to two baseline methods, ED-HNN (Wang et al., 2023a) and WHATsNet (Choe et al., 2023). ED-HNN is a universal approximator for any node-representation hypergraph diffusion process, while WHATsNet is the overall best baseline in the ENC experiments.

**Results.** The results in terms of Mean Absolute Error (MAE) are reported in Table 2. All methods demonstrate superior performance in approximating the invariant operator derived from differentiable `CE` functions. Conversely, approximating the equivariant operators derived from non-differentiable functions, `TV` and `LEC`, are more challenging. The proposed

Table 2: **MAE($\downarrow$) of approximating diffusion processes with common regularization functions.**

| Method | CE | TV | LEC |
|---|---|---|---|
| ED-HNN | $0.0132 \pm 0.0028$ | $0.0394 \pm 0.0011$ | $0.2057 \pm 0.0004$ |
| WHATsNet | $0.0065 \pm 0.0019$ | $0.0380 \pm 0.0007$ | $0.2056 \pm 0.0014$ |
| CoNHD (ADMM) | $\underline{0.0012} \pm 0.0001$ | $0.0293 \pm 0.0000$ | $\mathbf{0.0532} \pm 0.0031$ |
| CoNHD (GD) | $\mathbf{0.0011} \pm 0.0003$ | $\mathbf{0.0292} \pm 0.0001$ | $\underline{0.0561} \pm 0.0056$ |

method CoNHD can achieve the lowest MAE results compared to the baseline methods in all settings. While the ADMM-based implementation is theoretically more suitable for approximating non-differentiable regularization functions, it demonstrates minimal performance differences compared to the GD-based implementation in practice.

## 5.3 ABLATION STUDY

One critical design choice in the proposed CoNHD method is the use of equivariant functions without aggregation. Previous work (Choe et al., 2023) add aggregation after the equivariant functions to generate a single node or edge representation, where the composition is still a single-output invariant function and leads to the three limitations discussed in the introduction. To investigate the effectiveness of our design choice, we apply a mean aggregation to the outputs of our equivariant functions. This reduces the diffusion operators to invariant single-output functions with the same output for different node-edge pairs. We conduct experiments on `Email-Enron` and `Email-Eu`.

As shown in Table 3, CoNHD with two equivariant operators achieves the highest performance, exhibiting significant improvements compared to the variant with two invariant operators. Furthermore, variants with just one equivariant operator still outperform the fully invariant model. This suggests that equivariance benefits the modeling of both within-edge and within-node interactions. We also notice that the performance gap between the full equivariant model and the variant with only the equiv-

Table 3: **Effectiveness of the equivariance in two diffusion operators $\phi$ and $\varphi$.** ✓ and ✗ indicate whether the corresponding operator is equivariant or invariant, respectively. Shaded cells indicate the variants with equivariance significantly outperform the one with only invariant operators.

| Method | $\phi$ | $\varphi$ | Email-Enron | | Email-Eu | |
|---|---|---|---|---|---|---|
| | | | Micro-F1 | Macro-F1 | Micro-F1 | Macro-F1 |
| CoNHD (UNB) | ✗ | ✗ | $0.827 \pm 0.000$ | $0.769 \pm 0.004$ | $0.673 \pm 0.000$ | $0.645 \pm 0.001$ |
| | ✗ | ✓ | $0.876 \pm 0.001$ | $0.817 \pm 0.006$ | $0.698 \pm 0.001$ | $0.677 \pm 0.002$ |
| | ✓ | ✗ | $0.903 \pm 0.001$ | $0.855 \pm 0.004$ | $0.707 \pm 0.000$ | $0.688 \pm 0.002$ |
| | ✓ | ✓ | $\mathbf{0.905} \pm 0.001$ | $\mathbf{0.858} \pm 0.004$ | $\mathbf{0.708} \pm 0.001$ | $\mathbf{0.689} \pm 0.001$ |
| CoNHD (ISAB) | ✗ | ✗ | $0.829 \pm 0.001$ | $0.765 \pm 0.007$ | $0.673 \pm 0.001$ | $0.647 \pm 0.002$ |
| | ✗ | ✓ | $0.878 \pm 0.001$ | $0.823 \pm 0.005$ | $0.698 \pm 0.001$ | $0.678 \pm 0.003$ |
| | ✓ | ✗ | $0.910 \pm 0.001$ | $0.870 \pm 0.003$ | $0.707 \pm 0.001$ | $0.689 \pm 0.001$ |
| | ✓ | ✓ | $\mathbf{0.911} \pm 0.001$ | $\mathbf{0.871} \pm 0.002$ | $\mathbf{0.709} \pm 0.001$ | $\mathbf{0.690} \pm 0.002$ |

ariant within-edge operator $\phi$ is not significant. This might imply that within-edge interactions can provide the majority of the information needed for predicting the ENC labels in these datasets.

## 6 CONCLUSION

In this paper, we develop CoNHD, a novel HGNN based on hypergraph diffusion. CoNHD explicitly models within-edge and within-node interactions among co-representations as multi-input multi-output functions, which demonstrates three advantages: adaptive representation size, diverse diffusion information, and sufficient direct interactions among nodes or edges (see Appendix A for more details on how CoNHD achieves this). Our experiments demonstrate: (1) CoNHD achieves best performance on ten real-world ENC datasets without sacrificing efficiency. (2) CoNHD can effectively approximate the co-representation hypergraph diffusion process with common regularization functions. (3) Implementing interactions as multi-input multi-output equivariant functions without aggregation is essential for performance improvements. In Appendix I, we further show that CoNHD can achieve superior performance on downstream tasks and traditional node classification tasks, and mitigate the oversmoothing issue when constructing deep models.

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

# Appendix

## Contents

## A    ADVANTAGES OF CoNHD

Message passing models interactions in both within-edge and within-node structures as multi-input single-output functions, leading to the three limitations as outlined in the introduction. In this section, we highlight how the proposed CoNHD method addresses these limitations, offering the following three corresponding advantages:

- **Adaptive representation size.** In message passing, messages from numerous edges are aggregated to a fixed-size node representation vector, which can cause potential information loss for large-degree nodes. CoNHD addresses this limitation by introducing node-edge co-representations, which avoids aggregating information to a single node or edge representation. For larger-degree nodes or edges, they are contained by more node-edge pairs and therefore are associated with more co-representations, while the lower-degree nodes have less co-

representations. The number of co-representations adaptively scales with node or edge degrees, which can prevent potential information loss for large-degree nodes.

- **Diverse diffusion information.** In message passing, since the aggregation process mixes information from different edges to a single node representation and cannot differentiate specific information for each edge, the node can only pass the same message to different edges. In CoNHD, a node can have multiple co-representations that related to each hyperedge, and thus can generate diverse diffusion information in the interactions within different hyperedges. Additionally, CoNHD reduces the unnecessary aggregation process and avoids mixing different information into a single node or edge representations. The edge-dependent node information is preserved in the co-representations at each convolution layer, which obviates the necessity of extracting edge-dependent node information from a mixed node representation and reduces the learning difficulty compared to those edge-dependent message passing methods.

- **Sufficient direct interactions among nodes or edges.** In message passing, the single-output aggregation process is unable to capture direct nodes-to-nodes or edges-to-edges interactions, as they require multiple outputs for different elements. In CoNHD, the interactions in within-edge and within-node structures are designed as multi-input multi-output functions among multiple node-edge co-representations, which includes not only interactions between nodes and edges, but also direct nodes-to-nodes and edges-to-edges interactions.

Previous efforts have attempted to address the limitations of non-adaptive messages and insufficient interactions by extracting edge-dependent node messages (Aponte et al., 2022; Wang et al., 2023a; Choe et al., 2023; Telyatnikov et al., 2023) or introducing a three-stage message passing process (Pei et al., 2024). However, these approaches each tackle only a specific limitation, leaving the others unresolved and even introducing additional learning difficulties (like introducing an additional extraction process). Moreover, none of the previous methods can solve the limitation of non-adaptive representation size. After carefully analyzing the fundamental causes of these three limitations, we identified that they all stem from the single-output design in message passing. CoNHD overcomes these challenges through a multi-input multi-output design based on co-representations, offering a unified and elegant HGNN architecture with the above three advantages.

# B DERIVATIONS AND PROOFS

## B.1 DERIVATION OF ADMM OPTIMIZATION PROCESS (EQUATION A5-A7)

We use the ADMM method (Boyd et al., 2011) to solve the optimization problem in Eq. 5 with non-differentiable functions. We first introduce an auxiliary variable $\boldsymbol{A}_e$ for each edge $e$, and an auxiliary variable $\boldsymbol{B}_v$ for each node $v$. Then the problem in Eq. 5 can be formulated as:

$$
\begin{aligned}
\min_{\boldsymbol{H}} \quad & \sum_{v \in \mathcal{V}} \sum_{e \in \mathcal{E}_v} \mathcal{R}_{v,e}(\boldsymbol{h}_{v,e}; \boldsymbol{a}_{v,e}) + \lambda \sum_{e \in \mathcal{E}} \Omega_e(\boldsymbol{A}_e) + \gamma \sum_{v \in \mathcal{V}} \Omega_v(\boldsymbol{B}_v), \\
\text{s.t.} \quad & \forall e \in \mathcal{E} : \boldsymbol{A}_e = \boldsymbol{H}_e, \\
& \forall v \in \mathcal{V} : \boldsymbol{B}_v = \boldsymbol{H}_v.
\end{aligned}
$$

Then the scaled form augmented Lagrangian function can be transformed as:

$$
\begin{aligned}
L_\rho = & \sum_{v \in \mathcal{V}} \sum_{e \in \mathcal{E}_v} \mathcal{R}_{v,e}(\boldsymbol{h}_{v,e}; \boldsymbol{a}_{v,e}) + \lambda \sum_{e \in \mathcal{E}} \Omega_e(\boldsymbol{A}_e) + \gamma \sum_{v \in \mathcal{V}} \Omega_v(\boldsymbol{B}_v) \\
& + \sum_{e \in \mathcal{E}} \frac{\rho}{2} \left( \|\boldsymbol{A}_e - \boldsymbol{H}_e + \boldsymbol{P}_e\|_F^2 - \|\boldsymbol{P}_e\|_F^2 \right) \\
& + \sum_{v \in \mathcal{V}} \frac{\rho}{2} \left( \|\boldsymbol{B}_v - \boldsymbol{H}_v + \boldsymbol{Q}_v\|_F^2 - \|\boldsymbol{Q}_v\|_F^2 \right),
\end{aligned}
$$

where $\boldsymbol{P}_e$ and $\boldsymbol{Q}_v$ are the scaled dual variables (with scaling factor $\frac{1}{\rho}$). Then we can use the primal-dual algorithms in ADMM to find the optimal solutions (Boyd et al., 2011).

The primal steps can be calculated as follows:

$$\boldsymbol{A}_e^{(t+1)} := \arg\min_{\boldsymbol{A}_e} L_\rho$$

$$= \arg\min_{\boldsymbol{A}_e} \frac{\lambda}{\rho}\Omega_e(\boldsymbol{A}_e) + \frac{1}{2}\|\boldsymbol{A}_e - \boldsymbol{H}_e^{(t)} + \boldsymbol{P}_e^{(t)}\|_F^2$$

$$= \mathbf{prox}_{\lambda\Omega_e/\rho}(\boldsymbol{H}_e^{(t)} - \boldsymbol{P}_e^{(t)}), \ \forall e \in \mathcal{E},$$

$$\boldsymbol{B}_v^{(t+1)} := \arg\min_{\boldsymbol{B}_v} L_\rho$$

$$= \arg\min_{\boldsymbol{B}_v} \frac{\gamma}{\rho}\Omega_v(\boldsymbol{B}_v) + \frac{1}{2}\|\boldsymbol{B}_v - \boldsymbol{H}_v^{(t)} + \boldsymbol{Q}_v^{(t)}\|_F^2$$

$$= \mathbf{prox}_{\gamma\Omega_v/\rho}(\boldsymbol{H}_v^{(t)} - \boldsymbol{Q}_v^{(t)}), \ \forall v \in \mathcal{V},$$

$$\boldsymbol{h}_{v,e}^{(t+1)} := \arg\min_{\boldsymbol{h}_{v,e}} L_\rho$$

$$= \arg\min_{\boldsymbol{h}_{v,e}} \mathcal{R}_{v,e}(\boldsymbol{h}_{v,e}; \boldsymbol{a}_{v,e}) + \frac{\rho}{2}\|\boldsymbol{h}_{v,e} - [\boldsymbol{A}_e^{(t+1)}]_v - [\boldsymbol{P}_e^{(t)}]_v\|_2^2$$

$$+ \frac{\rho}{2}\|\boldsymbol{h}_{v,e} - [\boldsymbol{B}_v^{(t+1)}]_e - [\boldsymbol{Q}_v^{(t)}]_e\|_2^2$$

$$= \arg\min_{\boldsymbol{h}_{v,e}} \frac{1}{2\rho}\mathcal{R}_{v,e}(\boldsymbol{h}_{v,e}; \boldsymbol{a}_{v,e})$$

$$+ \frac{1}{2}\left\|\boldsymbol{h}_{v,e} - \frac{1}{2}\big([\boldsymbol{A}_e^{(t+1)}]_v + [\boldsymbol{P}_e^{(t)}]_v + [\boldsymbol{B}_v^{(t+1)}]_e + [\boldsymbol{Q}_v^{(t)}]_e\big)\right\|_2^2$$

$$= \mathbf{prox}_{\mathcal{R}_{v,e}(\cdot;\boldsymbol{a}_{v,e})/2\rho}\Big(\frac{1}{2}\big([\boldsymbol{A}_e^{(t+1)}]_v + [\boldsymbol{P}_e^{(t)}]_v + [\boldsymbol{B}_v^{(t+1)}]_e + [\boldsymbol{Q}_v^{(t)}]_e\big)\Big), \ \forall e \in \mathcal{E}_v, \ \forall v \in \mathcal{V}.$$

The dual steps can be calculated as follows:

$$\boldsymbol{P}_e^{(t+1)} := \boldsymbol{P}_e^{(t)} + \boldsymbol{A}_e^{(t+1)} - \boldsymbol{H}_e^{(t+1)}, \ \forall e \in \mathcal{E},$$

$$\boldsymbol{Q}_v^{(t+1)} := \boldsymbol{Q}_v^{(t)} + \boldsymbol{B}_v^{(t+1)} - \boldsymbol{H}_v^{(t+1)}, \ \forall v \in \mathcal{V}.$$

By defining $\boldsymbol{U}_e^{(t+1)} = \boldsymbol{A}_e^{(t+1)} + \boldsymbol{P}_e^{(t)}$ and $\boldsymbol{Z}_v^{(t+1)} = \boldsymbol{B}_v^{(t+1)} + \boldsymbol{Q}_v^{(t)}$, the update process can be simplified as follows:

$$\boldsymbol{U}_e^{(t+1)} = \mathbf{prox}_{\lambda\Omega_e/\rho}(2\boldsymbol{H}_e^{(t)} - \boldsymbol{U}_e^{(t)}) + \boldsymbol{U}_e^{(t)} - \boldsymbol{H}_e^{(t)}, \ \forall e \in \mathcal{E},$$

$$\boldsymbol{Z}_v^{(t+1)} = \mathbf{prox}_{\gamma\Omega_v/\rho}(2\boldsymbol{H}_v^{(t)} - \boldsymbol{Z}_v^{(t)}) + \boldsymbol{Z}_v^{(t)} - \boldsymbol{H}_v^{(t)}, \ \forall v \in \mathcal{V},$$

$$\boldsymbol{h}_{v,e}^{(t+1)} = \mathbf{prox}_{\mathcal{R}_{v,e}(\cdot;\boldsymbol{a}_{v,e})/2\rho}\Big(\frac{1}{2}\big([\boldsymbol{U}_e^{(t+1)}]_v + [\boldsymbol{Z}_v^{(t+1)}]_e\big)\Big), \ \forall e \in \mathcal{E}_v, \ \forall v \in \mathcal{V}.$$

### B.2 PROOF OF PROPOSITION 1

**Proposition 1** (Wang et al. (2023a)). *With permutation invariant structural regularization functions, the diffusion operators are permutation equivariant.*

*Proof.* Proved in Proposition 2 in (Wang et al., 2023a). □

### B.3 PROOF OF PROPOSITION 2

**Proposition 2.** *The traditional node-representation hypergraph diffusion is a special case of the co-representation hypergraph diffusion, while the opposite is not true.*

*Proof.* We first rewrite the node-representation hypergraph diffusion defined in Eq. 4 as a constraint optimization problem, then show that it is a special case of co-representation hypergraph diffusion defined in Eq. 5.

For each $v \in \mathcal{V}$, we introduce a set of new variables $\{\boldsymbol{h}_{v,e_i} | e_i \in \mathcal{E}_v\}$, satisfying $\boldsymbol{h}_{v,e_1} = \boldsymbol{x}_v$, and $\boldsymbol{h}_{v,e_i} = \boldsymbol{h}_{v,e_j}$ for any $e_i, e_j \in \mathcal{E}_v$. Then the objective function in Eq. 4 becomes:

$$\sum_{v \in \mathcal{V}} \mathcal{R}_v(\boldsymbol{x}_v; \boldsymbol{a}_v) + \lambda \sum_{e \in \mathcal{E}} \Omega_e(\boldsymbol{X}_e) = \sum_{v \in \mathcal{V}} \sum_{e \in \mathcal{E}_v} \frac{1}{d_v} \mathcal{R}_v(\boldsymbol{x}_v; \boldsymbol{a}_v) + \lambda \sum_{e \in \mathcal{E}} \Omega_e(\boldsymbol{X}_e)$$
$$= \sum_{v \in \mathcal{V}} \sum_{e \in \mathcal{E}_v} \frac{1}{d_v} \mathcal{R}_v(\boldsymbol{h}_{v,e}; \boldsymbol{a}_v) + \lambda \sum_{e \in \mathcal{E}} \Omega_e(\boldsymbol{H}_e),$$

The original problem in Eq. 4 can be reformulated as a constraint optimization problem:

$$\arg\min_{\boldsymbol{H}} \quad \sum_{v \in \mathcal{V}} \sum_{e \in \mathcal{E}_v} \frac{1}{d_v} \mathcal{R}_v(\boldsymbol{h}_{v,e}; \boldsymbol{a}_v) + \lambda \sum_{e \in \mathcal{E}} \Omega_e(\boldsymbol{H}_e), \tag{A1}$$
$$\text{s.t.} \quad \forall v \in \mathcal{V}, \forall e_i, e_j \in \mathcal{E}_v : \boldsymbol{h}_{v,e_i} = \boldsymbol{h}_{v,e_j},$$

where the optimal solutions satisfy $\boldsymbol{h}_{v,e}^* = \boldsymbol{x}_v^*$.

We now show that this constraint optimization is a special case of co-representation hypergraph diffusion. We can set $\mathcal{R}_{v,e}(\cdot; \boldsymbol{a}_{v,e}) = \frac{1}{d_v} \mathcal{R}_v(\cdot; \boldsymbol{a}_v)$, and use the CE regularization functions (Zhou et al., 2007) for the node regularization functions in Eq. 5, *i.e.*, $\Omega_{\text{CE}}(\boldsymbol{H}_v) := \sum_{e_i, e_j \in \mathcal{E}_v} \|\boldsymbol{h}_{v,e_i} - \boldsymbol{h}_{v,e_j}\|_2^2$. Then Eq. 5 can be reformulated as follows:

$$\arg\min_{\boldsymbol{H}} \sum_{v \in \mathcal{V}} \sum_{e \in \mathcal{E}_v} \frac{1}{d_v} \mathcal{R}_v(\boldsymbol{h}_{v,e}; \boldsymbol{a}_v) + \lambda \sum_{e \in \mathcal{E}} \Omega_e(\boldsymbol{H}_e) + \gamma \sum_{v \in \mathcal{V}} \Omega_{\text{CE}}(\boldsymbol{H}_v). \tag{A2}$$

The node regularization term in A2 is exactly the exterior penalty function (Yeniay, 2005) for the given equality constraints in Eq. A1. Thus when $\gamma \to \infty$, Eq. A2 yields the same optimal solutions as Eq. A1.

To show that the opposite is not true, we only need to consider the cases that the co-representations according to the same node are not identical. As the node-representation hypergraph diffusion only have one representation for each node, it cannot represent the multiple co-representations in the co-representation hypergraph diffusion. □

### B.4 PROOF OF PROPOSITION 3

**Proposition 3.** *With the same co-representation dimension, CoNHD is expressive enough to represent the message passing framework, while the opposite is not true.*

*Proof.* We prove the proposition using the GD-based implementation of CoNHD, which can be easily extended to the ADMM-based implementation.

First, we prove that CoNHD is expressive enough to represent any model within the message passing framework. We initialize the co-representations as $\boldsymbol{h}_{v,e}^{(0)} = \big[\boldsymbol{x}_v^{(0)}, \boldsymbol{z}_e^{(0)}\big]$. For brevity, we assume $\boldsymbol{x}_v^{(0)} \in \mathbb{R}^{\frac{d}{2}}$ and $\boldsymbol{z}_e^{(0)} \in \mathbb{R}^{\frac{d}{2}}$. We will show that given $\boldsymbol{h}_{v,e}^{(2t)} = \big[\boldsymbol{x}_v^{(t)}, \boldsymbol{z}_e^{(t)}\big]$, two layers of CoNHD are expressive enough to generate $\boldsymbol{h}_{v,e}^{(2(t+1))} = \big[\boldsymbol{x}_v^{(t+1)}, \boldsymbol{z}_e^{(t+1)}\big]$, where $\boldsymbol{x}_v^{(t)}$ and $\boldsymbol{z}_e^{(t)}$ exactly correspond to the node and edge representations in the $t$-th layer of the message passing framework defined in Eq. 1-3.

When $\phi$ and $\varphi$ are implemented by universal equivariant neural diffusion operators like UNB, they are expressive enough to represent any equivariant mapping. With the same co-representation dimension, we can use one layer in CoNHD to represent the nodes-to-edge aggregation process in the message passing framework. In this layer, we reduce $\phi$, $\varphi$, and $\psi$ as follows:

$$\boldsymbol{M}_e^{(2t+1)} = \phi(\boldsymbol{H}_e^{(2t)}) = \big[\boldsymbol{0}_{d_e \times \frac{d}{2}}, \boldsymbol{1}_{d_e} \cdot \big(f_{\mathcal{V} \to \mathcal{E}}(\boldsymbol{X}_e^{(t)}; \boldsymbol{z}_e^{(t)})\big)^\top\big]$$
$$= \big[\boldsymbol{0}_{d_e \times \frac{d}{2}}, \boldsymbol{1}_{d_e} \cdot \boldsymbol{z}_e^{(t+1)\top}\big],$$
$$\boldsymbol{M}_v'^{(2t+1)} = \varphi(\boldsymbol{H}_v^{(2t)}) = \big[\boldsymbol{0}_{d_v \times \frac{d}{2}}, \boldsymbol{0}_{d_v \times \frac{d}{2}}\big],$$
$$\boldsymbol{h}_{v,e}^{(2t+1)} = \psi([\boldsymbol{h}_{v,e}^{(2t)}, \boldsymbol{m}_{v,e}^{(2t+1)}, \boldsymbol{m}_{v,e}'^{(2t+1)}, \boldsymbol{h}_{v,e}^{(0)}]) = \big[\boldsymbol{x}_v^{(t)}, \boldsymbol{z}_e^{(t+1)}\big],$$

where $\mathbf{1}_n$ represents a $n$-dimensional all one vector, which is used to construct a matrix with repeated row elements. $\mathbf{0}_{m \times n}$ represents a $(m \times n)$-dimensional all zero matrix. In this layer, we use $\phi$ to represent the aggregation process and ignore the output of $\varphi$.

We use another layer to represent the edges-to-node aggregation process and the skip connection. In this layer, we reduce $\phi$, $\varphi$, and $\psi$ as follows:

$$\boldsymbol{M}_e^{(2(t+1))} = \phi(\boldsymbol{H}_e^{(2t+1)}) = \left[ \mathbf{0}_{d_e \times \frac{d}{2}}, \mathbf{0}_{d_e \times \frac{d}{2}} \right],$$

$$\boldsymbol{M}_v^{\prime(2(t+1))} = \varphi(\boldsymbol{H}_v^{(2t+1)}) = \left[ \mathbf{1}_{d_v} \cdot \left( f_{\mathcal{E} \to \mathcal{V}}(\boldsymbol{Z}_v^{(t+1)}; \boldsymbol{x}_v^{(t)}) \right)^\top, \mathbf{0}_{d_e \times \frac{d}{2}} \right]$$

$$= \left[ \mathbf{1}_{d_v} \cdot \tilde{\boldsymbol{x}}_v^{(t+1)\top}, \mathbf{0}_{d_e \times \frac{d}{2}} \right],$$

$$\boldsymbol{h}_{v,e}^{(2(t+1))} = \psi([\boldsymbol{h}_{v,e}^{(2t+1)}, \boldsymbol{m}_{v,e}^{(2(t+1))}, \boldsymbol{m}_{v,e}^{\prime(2(t+1))}, \boldsymbol{h}_{v,e}^{(0)}])$$

$$= \left[ f_{\text{skip}}(\boldsymbol{x}_v^{(t)}, \tilde{\boldsymbol{x}}_v^{(t+1)}, \boldsymbol{x}_v^{(0)}), \boldsymbol{z}_e^{(t+1)} \right]$$

$$= \left[ \boldsymbol{x}_v^{(t+1)}, \boldsymbol{z}_e^{(t+1)} \right].$$

In this layer, we use $\varphi$ to represent the aggregation process and ignore the output of $\phi$. Besides, we set the update function $\psi$ to represent the skip connection part. The final output $\boldsymbol{x}_v^{(t+1)}$ and $\boldsymbol{z}_e^{(t+1)}$ are the $(t+1)$-th node and edge representation in the message passing framework. Therefore, CoNHD is expressive enough to represent any model within the message passing framework.

To show that the opposite is not true, we only need to construct a counter-example. Since $\phi$ is equivariant, it can generate different diffusion information $\boldsymbol{m}_{v,e}^{(t)}$ for different node-edge pair $(v, e)$. We can simply set $\boldsymbol{h}_{v,e}^{(t)} = \boldsymbol{m}_{v,e}^{(t)}$ in the update function $\psi$, which lead to different representation for each node-edge pair $(v, e)$. However, the message passing framework can only generate the same edge representation for each edge, which constraints that the first $\frac{d}{2}$ dimension of the co-representations for different node-edge pairs are the same and cannot generate the same $\boldsymbol{h}_{v,e}^{(t)}$. Therefore, any model within the message passing framework cannot represent CoNHD. $\square$

## C IMPLEMENTATION OF THE DIFFUSION OPERATORS

We explore two popular equivariant network architectures, UNB (Segol & Lipman, 2020; Wang et al., 2023a) and ISAB (Chien et al., 2022), for the implementation of the diffusion operators $\phi$ and $\varphi$. Apart from these two equivariant networks explored in our experiments, it is worth noting that our proposed CoNHD is a general HGNN architecture and can be combined with any other equivariant neural network.

**UNweighted Block (UNB).** UNB is a widely investigated set-equivariant neural network (Zaheer et al., 2017; Qi et al., 2017; Segol & Lipman, 2020; Wang et al., 2023a). It first generates global information by an unweighted pooling operation, and then concatenates it with each element to generate the output for the corresponding element using a MLP. As it utilizes an unweighted pooling to aggregate global set information, we refer to this implementation as UNweighted Block (UNB). The UNB module can be represented as follows:

$$\text{UNB}: \quad [\text{UNB}(\tilde{\boldsymbol{H}})]_i = \text{MLP}\left( \left[ \tilde{\boldsymbol{h}}_i, \sum_{\tilde{\boldsymbol{h}}_j \in \boldsymbol{H}} \text{MLP}(\tilde{\boldsymbol{h}}_j) \right] \right). \tag{A3}$$

Here $\tilde{\boldsymbol{H}} = [\tilde{\boldsymbol{h}}_1, \ldots, \tilde{\boldsymbol{h}}_{n_H}]^\top \in \mathbb{R}^{n_H \times d}$ represents a matrix with $n_H$ co-representation vectors, which can be replaced by $\boldsymbol{H}_e$ or $\boldsymbol{H}_v$ for the within-edge or within-node diffusion operators, respectively. $\text{MLP}(\cdot)$ is a Multi-Layer Perceptron (MLP). This simple implementation can approximate **any** continuous permutation equivariant functions (Segol & Lipman, 2020; Wang et al., 2023a), leading to a universal approximator for our diffusion operators. Besides, its time complexity is linear to the number of the input co-representations.

**Induced Set Attention Block (ISAB).** The static unweighted operation ignores the importance of different elements, limiting its ability to capture interactions in practice (Lee et al., 2019; Kim et al., 2021; 2022). Therefore, we consider another implementation using the ISAB module in Set Transformer (Lee et al., 2019), which is based on self-attention. The ISAB module can be formulated as follows:

$$\text{ISAB:} \quad \text{ISAB}(\tilde{\boldsymbol{H}}) = \text{MAB}(\tilde{\boldsymbol{H}}, \text{MAB}(\boldsymbol{W}^I, \tilde{\boldsymbol{H}})), \tag{A4}$$

$$\text{where} \quad \text{MAB}(\boldsymbol{Q}, \boldsymbol{K}) = \text{LN}(\boldsymbol{M} + \text{RFF}(\boldsymbol{M})), \ \boldsymbol{M} = \text{LN}(\boldsymbol{Q}, \text{MULTIHEAD}(\boldsymbol{Q}, \boldsymbol{K}, \boldsymbol{K})),$$

$$\text{MULTIHEAD}(\boldsymbol{Q}, \boldsymbol{K}, \boldsymbol{V}) = [\boldsymbol{O}_1, \ldots, \boldsymbol{O}_h] \cdot \boldsymbol{W}^O,$$

$$\boldsymbol{O}_i = \omega(\boldsymbol{Q}\boldsymbol{W}_i^Q(\boldsymbol{K}\boldsymbol{W}_i^K)^\top)\boldsymbol{V}\boldsymbol{W}_i^V.$$

Here $\boldsymbol{W}^I$, $\boldsymbol{W}^O$, $\boldsymbol{W}^Q$, $\boldsymbol{W}^K$, and $\boldsymbol{W}^V$ are all trainable weights. $\text{LN}(\cdot)$ denotes the layer normalization. $\text{RFF}(\cdot)$ is a row-wise feed-forward layer. $\text{MULTIHEAD}(\cdot)$ is the multihead attention mechanism and $\omega$ is the softmax function. ISAB utilizes a fixed number of inducing points $\boldsymbol{I} \in \mathbb{R}^{k \times d}$ to reduce the quadratic complexity in self attention to linear complexity (Lee et al., 2019), which can increase the efficiency when modeling hypergraphs with larger node and hyperedge degrees.

Some previous works also explore Set Transformer in their message passing-based HGNN implementations (Chien et al., 2022; Choe et al., 2023). However, AllSet (Chien et al., 2022) employs the invariant module instead of the equivariant ISAB module in Set Transformer, leading to a single-output implementation. WHATsNet (Choe et al., 2023) investigates the equivariant ISAB module but with another aggregation module after the ISAB module, which still degrades to a single-output implementation. In contrast, due to the introduced co-representations, CoNHD reduces the necessity of generating single node or edge representations and therefore removes the uncessary aggregation process. The implementation relies solely on the permutation equivariant module ISAB without aggregation, which can be more expressive compared to AllSet and WHATsNet, and can solve the three limitations in these message passing-based methods.

## D EFFICIENCY OF CoNHD

In this section, we provide a theoretical analysis of the time and space complexity to evaluate the computational and memory efficiency of our method. We then discuss the additional computational efficiency advantages of our approach compared to message passing-based HGNNs under the same mini-batch training setup.

### D.1 COMPLEXITY ANALYSIS

**Time Complexity.** We discuss the time complexity of two GD-based implementations using UNB or ISAB operators, while the ADMM-based implementations have similar results.

Both the UNB and ISAB operators have linear complexity with the number of the input co-representations. For the UNB implementation, the first MLP and the sum pooling only calculate once for all elements in the set, and the second MLP calculates in an element-wise manner. We set the same hidden size as the co-representation size for MLPs. With co-representation dimension $d$, the overall complexity for the within-edge and within-node interactions in each layer is $\mathcal{O}(\sum_{e \in \mathcal{E}}(d_e d^2) + \sum_{v \in \mathcal{V}}(d_v d^2)) = \mathcal{O}(d^2 \sum_{e \in \mathcal{E}} d_e)$. This equation follows from the fact that the sum of node degrees is equal to the sum of edge degrees, *i.e.*, $\sum_{v \in \mathcal{V}} d_v = \sum_{e \in \mathcal{E}} d_e$. The ISAB implementation requires dot products between the input co-representations and $k$ inducing points. The overall complexity for the within-edge and within-node interactions in each layer is $\mathcal{O}(\sum_{e \in \mathcal{E}}(d_e kd + (d_e + k)d^2) + \sum_{v \in \mathcal{V}}(d_v kd + (d_v + k)d^2)) = \mathcal{O}((dk + d^2) \sum_{e \in \mathcal{E}} d_e + \sum_{e \in \mathcal{E}} kd^2 + \sum_{v \in \mathcal{V}} kd^2)$. When $k$ is small (in our experiments, $k = 4$), this complexity can be simplified as $\mathcal{O}(d^2 \sum_{e \in \mathcal{E}} d_e)$, which is consistent with the UNB implementation. For the update function, the complexity is $\mathcal{O}(d^2 \sum_{e \in \mathcal{E}} d_e)$. Therefore, the overall complexity of CoNHD is $\mathcal{O}(Ld^2 \sum_{e \in \mathcal{E}} d_e)$, where $L$ is the number of layers.

The overall time complexity is linear to the number of node-edge pairs in the input hypergraph, *i.e.*, $\sum_{e \in \mathcal{E}} d_e$, which is the same as other HGNNs within the message passing framework (*e.g.*, the overall best baseline WHATsNet (Choe et al., 2023) in the ENC experiments).

**Space Complexity.** To maintain consistency, we use the same notations as those used in the time complexity analysis. Since the number of input co-representations in each layer of our model depends on the number of node-hyperedge pairs, *i.e.*, $\sum_{e \in \mathcal{E}} d_e$, the size of the inputs is $\mathcal{O}(d \sum_{e \in \mathcal{E}} d_e)$. For within-edge and within-node interactions, both UNB or ISAB implementation utilizes some

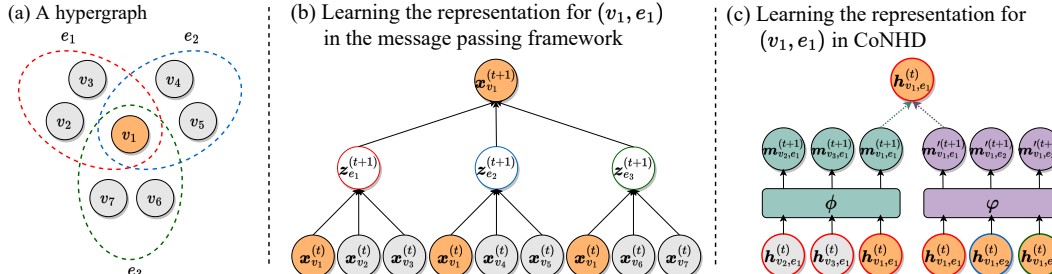

Figure A1: **Learning the representation for a node-edge pair in mini-batch training.** (a) An example hypergraph, where node $v_1$ and all edges have degree 3. We want to learn the representation for node-edge pair $(v_1, e_1)$. (b) In the message passing framework, each layer performs a two-stage aggregation process: from nodes to edges and from edges back to nodes. This process involves all nodes in the neighboring edges of node $v_1$. Some nodes (*e.g.*, $v_4$, $v_5$, $v_6$, and $v_7$) are not direct neighbors for the node-edge pair $(v_1, e_1)$. (c) In contrast, CoNHD focuses solely on the direct neighbors in each layer, including neighboring edges of $v_1$ (*i.e.*, $e_1$, $e_2$, and $e_3$) and neighboring nodes of $e_1$ (*i.e.*, $v_1$, $v_2$, and $v_3$). This not only ensures the diffusion information is from the most related neighbors, but also reduces the subgraph size in each layer and improves the efficiency.

MLPs to perform feature transformation, where the size of each MLP is $d \times d$. In UNB, two MLPs are utilized, while in ISAB, six MLPs are required due to the implementation of self-attention. Therefore, the total size of weights in UNB or ISAB should be $\mathcal{O}(d^2)$. For the ISAB implementation, additional inducing points are required to reduce the complexity of self-attention, with a size of $\mathcal{O}(kd)$. Similar to the case in time complexity analysis, when $k$ is small ($k = 4$ in our experiments), the size of these inducing points can be ignored compared to the weights. The sizes of the outputs for the within-edge and within-node interactions are both $\mathcal{O}(d \sum_{e \in \mathcal{E}} d_e)$. In the final co-representation update process, the input co-representations, initial features, and updated information from within-edge and within-node interactions are concatenated to form a $4d$-dimensional vector. This is then passed through a linear layer to output the updated co-representations, where the weight size is $\mathcal{O}(4d^2)$. Therefore, the total space complexity of $L$ layers after removing the constants is $\mathcal{O}(L(d^2 + d \sum_{e \in \mathcal{E}} d_e)) = \mathcal{O}(Ld(d + \sum_{e \in \mathcal{E}} d_e))$.

The overall space complexity is linear to the number of node-edge pairs in the input hypergraph, *i.e.*, $\sum_{e \in \mathcal{E}} d_e$. This is the same as those edge-dependent message passing-based methods, like the best baseline WHATsNet (Choe et al., 2023), which generates multiple edge-dependent node representations for each node in the calculation process.

### D.2    EFFICIENCY ADVANTAGES IN MINI-BATCH TRAINING

Despite the same theoretical computational complexity, CoNHD exhibits additional efficiency advantages in mini-batch training as shown in Section 5.1. Compared to full-batch training, mini-batch training is a more common setting for training on large real-world hypergraphs, which can reduce memory consumption. In mini-batch training, the overlapping of the subgraphs across different batches introduces additional computational overhead compared to full-batch training. The computational load scales with the size of the neighboring subgraph, *i.e.*, the number of neighboring edges and nodes.

For convenience, we assume all the node degrees and edge degrees are equal to $d_n$. To calculate the representation for each node-edge pair, in each layer, the message passing framework needs to calculate the two-stage aggregation process $\mathcal{V} \to \mathcal{E}$ and $\mathcal{E} \to \mathcal{V}$. This leads to a neighboring subgraph containing $d_n$ edges and $d_n^2$ nodes. As shown in Fig. A1(b), this process not only increases the computational complexity but also includes nodes that are not direct neighbors for the target node-edge pair, which may affect the learning results. In contrast, each layer of CoNHD only contains direct neighboring edges and nodes, as shown in Fig. A1(c), resulting in a smaller subgraph with $d_n$ edges and $d_n$ nodes. This can greatly improve the efficiency when handling complex hypergraphs with large node and edge degrees.

Additionally, our method further improves efficiency by reducing the unnecessary extraction and aggregation processes in edge-dependent message passing methods like WhatsNet (Choe et al., 2023).

---

**Algorithm 1:** CoNHD-GD for edge-dependent node classification.

---

**Input** : A hypergraph $\mathcal{G} = (\mathcal{V}, \mathcal{E})$, an initial node feature matrix $\boldsymbol{X}^{(0)} = [\boldsymbol{x}_{v_1}^{(0)}, \ldots, \boldsymbol{x}_{v_n}^{(0)}]^\top$.

**Output:** Predicted edge-dependent node labels $\hat{y}_{v,e}$.

**Initialize** $\forall v \in \mathcal{V}, \forall e \in \mathcal{E}_v : \boldsymbol{h}_{v,e}^{(0)} = \boldsymbol{x}_v^{(0)}$;

**for** $\ell = 1, \ldots, L$ **do**

    *// within-edge interactions (Eq. 7).*

    $\forall e \in \mathcal{E} : \boldsymbol{M}_e^{(\ell)} \leftarrow \phi(\boldsymbol{H}_e^{(\ell-1)})$;

    *// within-node interactions (Eq. 7).*

    $\forall v \in \mathcal{V} : \boldsymbol{M}_v'^{(\ell)} \leftarrow \varphi(\boldsymbol{H}_v^{(\ell-1)})$;

    *// co-representation updates (Eq. 8).*

    $\forall v \in \mathcal{V}, \forall e \in \mathcal{E}_v : \boldsymbol{h}_{v,e}^{(\ell)} \leftarrow \psi([\boldsymbol{h}_{v,e}^{(\ell-1)}, \boldsymbol{m}_{v,e}^{(\ell)}, \boldsymbol{m}_{v,e}'^{(\ell)}, \boldsymbol{h}_{v,e}^{(0)}])$;

**end**

**for** $v \in \mathcal{V}, e \in \mathcal{E}_v$ **do**

    Predict edge-dependent node label $\hat{y}_{v,e}$ using co-representation $\boldsymbol{h}_{v,e}^{(L)}$.

**end**

---

Edge-dependent message passing first extract edge-dependent information from a single node representation. This force the model learn to extract edge specific information from a single mixed node representation, which increases the learning difficulty. However, this extracted information is then aggregated back to a single node representation. Therefore, the model needs to repeat the "extract-aggregate" process in each layer of the HGNNs. In contrast, our method learns node-edge co-representations directly without any aggregation, which do not need to aggregate to a single node or edge representation and thus also reduce the unnecessary extraction process. Therefore, our method can avoid this complex "extract-aggregate" process. This not only reduces the learning difficulty by keeping the edge-dependent information directly without the need of extraction, but also further improves the efficiency of our method.

# E   ALGORITHMS OF CONHD

Algorithm 1 and 2 describe the forward propagation of the CoNHD model for edge-dependent node classification using GD-based and ADMM-based implementations, respectively. $\phi$ and $\varphi$ are implemented as UNB (Eq. A3) or ISAB (Eq. A4) in our experiments. Although the proposed CoNHD model can accept any features related to the node-edge pairs, we initialize the co-representations using only the node features to follow the setup of the original ENC problem (Choe et al., 2023). To predict the final labels $\hat{y}_{v,e}$, an MLP with the corresponding co-representation $\boldsymbol{h}_{v,e}^{(L)}$ as input is utilized to output the logits for each class following previous work (Choe et al., 2023).

# F   DISCUSSION ON OTHER HYPERGRAPH NEURAL NETWORKS

LEGCN (Yang et al., 2022) and MultiSetMixer (Telyatnikov et al., 2023) are two approaches that do not follow the message passing framework defined in Eq. 1-3 and can explicitly generate edge-dependent node representations, which are similar to the concept of co-representations in the proposed CoNHD model. Unfortunately, LEGCN transforms the hypergraph structure into a traditional graph structure, which loses some higher-order group information. Instead, MultiSetMixer preserves the hypergraph structure but still models it as a two-stage message passing process, where messages from nodes are aggregated to a single edge representation and then back to nodes. Both of these two methods model within-edge and within-node interactions as multi-input single-output functions, which can only generate the same output for different node-edge pairs and therefore limits their expressiveness.

**LEGCN.** LEGCN converts a hypergraph into a traditional graph using line expansion, and utilizes graph convolution (Kipf & Welling, 2017) to learn representations of the new nodes on the expanded

---

**Algorithm 2:** CoNHD-ADMM for edge-dependent node classification.

---

**Input** : A hypergraph $\mathcal{G} = (\mathcal{V}, \mathcal{E})$, an initial node feature matrix $\boldsymbol{X}^{(0)} = [\boldsymbol{x}_{v_1}^{(0)}, \ldots, \boldsymbol{x}_{v_n}^{(0)}]^\top$.

**Output:** Predicted edge-dependent node labels $\hat{y}_{v,e}$.

**Initialize** $\forall v \in \mathcal{V}, \forall e \in \mathcal{E}_v : \boldsymbol{h}_{v,e}^{(0)} = \boldsymbol{x}_v^{(0)}, \boldsymbol{m}_{v,e}^{(0)} = \boldsymbol{h}_{v,e}^{(0)}, \boldsymbol{m}_{v,e}'^{(0)} = \boldsymbol{h}_{v,e}^{(0)}$;

**for** $\ell = 1, \ldots, L$ **do**

  *// within-edge interactions (Eq. A8).*

  $\forall e \in \mathcal{E} : \boldsymbol{M}_e^{(\ell)} \leftarrow \phi(2\boldsymbol{H}_e^{(\ell-1)} - \boldsymbol{M}_e^{(\ell-1)}) + \boldsymbol{M}_e^{(\ell-1)} - \boldsymbol{H}_e^{(\ell-1)}$;

  *// within-node interactions (Eq. A9).*

  $\forall v \in \mathcal{V} : \boldsymbol{M}_v'^{(\ell)} \leftarrow \varphi(2\boldsymbol{H}_v^{(\ell-1)} - \boldsymbol{M}_v'^{(\ell-1)}) + \boldsymbol{M}_v'^{(\ell-1)} - \boldsymbol{H}_v^{(\ell-1)}$;

  *// co-representation updates (Eq. A10).*

  $\forall v \in \mathcal{V}, \forall e \in \mathcal{E}_v : \boldsymbol{h}_{v,e}^{(\ell)} \leftarrow \psi([\boldsymbol{m}_{v,e}^{(\ell)}, \boldsymbol{m}_{v,e}'^{(\ell)}, \boldsymbol{h}_{v,e}^{(0)}])$;

**end**

**for** $v \in \mathcal{V}, e \in \mathcal{E}_v$ **do**

  Predict edge-dependent node label $\hat{y}_{v,e}$ using co-representation $\boldsymbol{h}_{v,e}^{(L)}$.

**end**

---

graph. The $(t+1)$-th layer of LEGCN (without normalization) can be formulated as:

$$\boldsymbol{m}_e^{(t+1)} = \sum_{v_i \in e} \boldsymbol{h}_{v_i,e}^{(t)}, \; \boldsymbol{m}_v'^{(t+1)} = \sum_{e_j \in \mathcal{E}_v} \boldsymbol{h}_{v,e_j}^{(t)},$$

$$\boldsymbol{h}_{v,e}^{(t+1)} = \sigma(\lambda \boldsymbol{m}_e^{(t+1)} + \gamma \boldsymbol{m}_v'^{(t+1)}) \boldsymbol{W}^{(t)},$$

where $\sigma(\cdot)$ denotes the non-linear activation function. $\boldsymbol{m}_e^{(t)}$ and $\boldsymbol{m}_v^{(t)}$ are the within-edge information and the within-node information, respectively.

Compared to CoNHD, LEGCN exhibits three main limitations. *(1) Lack of differentiation between within-edge and within-node interactions.* In a line expansion graph, the new vertices (node-edge pairs) associated with the same edge or the same node are connected by homogeneous edges, which overlooks the difference between these two kinds of relations. Although LEGCN utilizes different scalar weights to balance within-edge or within-node messages, this is still not expressive enough compared to two different neural diffusion operators in CoNHD. *(2) Non-adaptive messages.* LEGCN still follows the single-output setting which only generates one shared within-edge message $\boldsymbol{m}_e^{(t)}$ and one shared within-node message $\boldsymbol{m}_v^{(t)}$ using sum pooling, instead of diverse messages for different node-edge pairs. *(3) High computational complexity.* The motivation for LEGCN is to reduce the hypergraph structure to a graph structure. This reduction loses the higher-order group information and requires additional computation for different node-edge pairs. For example, the learning for representations of node-edge pairs $(v_1, e_1)$ and $(v_1, e_2)$ are calculated separately, although they have the same within-node messages $\boldsymbol{m}_{v_1}'^{(t+1)}$ which only needs to be computed once. In contrast, our proposed CoNHD method generates all diffusion information within the same node $v_1$ together using the node diffusion operator with linear complexity.

**MultiSetMixer.** MultiSetMixer replaces each node representation in the message passing framework with several edge-dependent node representations. However, it still follows the message passing framework and aggregates different node representations into a single edge representation. This models the within-edge interactions as multi-input single-output functions. The $(t+1)$-th layer of MultiSetMixer can be formulated as:

$$\boldsymbol{m}_e^{(t+1)} = \frac{1}{d_e} \sum_{v_i \in e} \boldsymbol{h}_{v_i,e}^{(t)} + \text{MLP}(\text{LN}(\frac{1}{d_e} \sum_{v_i \in e} \boldsymbol{h}_{v_i,e}^{(t)})),$$

$$\boldsymbol{h}_{v,e}^{(t+1)} = \boldsymbol{h}_{v,e}^{(t)} + \text{MLP}(\text{LN}(\boldsymbol{h}_{v,e}^{(t)})) + \boldsymbol{m}_e^{(t+1)}.$$

MultiSetMixer also generates one shared within-edge message $\boldsymbol{m}_e^{(t)}$, which loses specific messages for different node-edge pair. This formulation still suffers from the three main limitations of the message passing framework. Besides, it does not incorporate within-node interactions, which cannot model the relations among different representations associated with the same node.

Table A1: **Full statistics of all datasets.**

| | Dataset | Num. of Nodes | Num. of Edges | Avg. $d_v$ | Avg. $d_e$ | Med. $d_v$ | Med. $d_e$ | Max. $d_v$ | Max. $d_e$ | Min. $d_v$ | Min. $d_e$ |
|---|---|---|---|---|---|---|---|---|---|---|---|
| **Edge-dependent Node Classification** | Email-Enron | 21,251 | 101,124 | 55.83 | 11.73 | 8 | 6 | 18,168 | 948 | 1 | 3 |
| | Email-Eu | 986 | 209,508 | 549.54 | 2.59 | 233 | 2 | 8,659 | 59 | 1 | 2 |
| | Stack-Biology | 15,490 | 26,823 | 3.63 | 2.10 | 1 | 2 | 1,318 | 12 | 1 | 1 |
| | Stack-Physics | 80,936 | 200,811 | 5.93 | 2.39 | 1 | 2 | 6,332 | 48 | 1 | 1 |
| | Coauth-DBLP | 108,484 | 91,266 | 2.96 | 3.52 | 1 | 3 | 236 | 36 | 1 | 2 |
| | Coauth-AMiner | 1,712,433 | 2,037,605 | 3.03 | 2.55 | 1 | 2 | 752 | 115 | 1 | 1 |
| | Cora-Outsider | 1,904 | 1,905 | 7.87 | 7.87 | 7 | 6 | 32 | 43 | 1 | 4 |
| | DBLP-Outsider | 34,106 | 40,240 | 9.76 | 8.27 | 9 | 6 | 51 | 202 | 1 | 4 |
| | Citeseer-Outsider | 767 | 1,420 | 10.49 | 5.67 | 9 | 5 | 141 | 26 | 2 | 4 |
| | Pubmed-Outsider | 3,450 | 14,075 | 32.91 | 8.07 | 28 | 6 | 167 | 171 | 6 | 4 |
| **Downstream Application** | Halo | 5,507 | 31,028 | 34.75 | 6.17 | 20 | 7 | 505 | 12 | 1 | 2 |
| | H-Index/AMiner | 187,297 | 115,196 | 2.09 | 3.39 | 1 | 3 | 191 | 66 | 1 | 2 |
| | DBLP | 2,123 | 1,000 | 1.83 | 3.88 | 1 | 4 | 22 | 25 | 1 | 2 |
| | Etail | 6,000 | 9,675 | 5.57 | 3.45 | 6 | 3 | 10 | 13 | 1 | 1 |
| **Node Classification** | Senate | 282 | 315 | 19.18 | 17.17 | 15 | 19 | 63 | 31 | 1 | 4 |
| | House | 1,290 | 340 | 9.18 | 34.83 | 7 | 40 | 44 | 81 | 1 | 2 |
| | Walmart | 88,860 | 69,906 | 5.18 | 6.59 | 2 | 5 | 5,733 | 25 | 1 | 2 |
| | Congress | 1,718 | 83,105 | 426.25 | 8.81 | 273 | 6 | 3,964 | 25 | 1 | 2 |
| | Cora-CA | 2,708 | 1,072 | 1.69 | 4.28 | 2 | 3 | 23 | 43 | 0 | 2 |
| | DBLP-CA | 41,302 | 22,363 | 2.41 | 4.45 | 2 | 3 | 18 | 202 | 1 | 2 |

# G ADDITIONAL DETAILS OF THE DATASETS

Table A1 provides a comprehensive overview of the datasets used in our experiments.

## G.1 DATASETS FOR EDGE-DEPENDENT NODE CLASSIFICATION

We use ten real-world edge-dependent node classification datasets. Six of them are from (Choe et al., 2023), which are Email (Email-Enron[1] and Email-Eu (Paranjape et al., 2017)), StackOverflow (Stack-Biology[2] and Stack-Physics[2]), and Co-authorship networks (Coauth-DBLP (Swati et al., 2017) and Coauth-AMiner[3]). In Email-Enron and Email-Eu, nodes represent individuals, and emails act as edges connecting them. The edge-dependent node labels denote the role of a user within an email (sender, receiver, or CC'ed). In Stack-Biology and Stack-Physics, nodes represent users while posts on Stack Overflow are hyperedges. The edge-dependent node label indicates the role of a user within a post (questioner, chosen answerer, or other answerer). In Coauth-DBLP and Coauth-AMiner, publications serve as hyperedges connecting authors (nodes) in these datasets. The edge-dependent node label represents the order of an author within a publication (first, last, or others).

Four newly introduced ENC datasets are derived by transforming the outsider identification problem (Zhang et al., 2020) into the ENC problem. We generate these datasets using Cora-CA, DBLP-CA, Citeseer, and Pubmed, which are four hypergraph datasets with original features in (Wang et al., 2023a). For each dataset, we first removed hyperedges with a degree less than or equal to 3. Then, for each remaining hyperedge, we randomly replaced half of the nodes with other nodes (outsiders) and generated five new hyperedges by different replacements. The labels indicate whether each node belongs to the corresponding hyperedge or is an outsider.

## G.2 DATASETS FOR DOWNSTREAM TASKS

For downstream tasks, we utilize all four datasets from (Choe et al., 2023). Halo is a game dataset where the edge-dependent node labels represent the scores of each player (node) in each match (hyperedge). It includes global rankings of all players, which serve as ground truth labels for the ranking aggregation task. H-Index and AMiner are derived from the same hypergraph dataset but are used for different downstream tasks. In these datasets, the edge-dependent node labels correspond to the order of co-authorship for each author (node) in each paper (hyperedge). The dataset

---

[1] https://www.cs.cmu.edu/~enron/

[2] https://archive.org/download/stackexchange

[3] https://www.aminer.org/aminernetwork

includes H-Index information for each author, which can be used for ranking aggregation, and also contains venue information for each paper, which serves as ground truth for the clustering task. `DBLP` is another co-authorship network with venue information for each paper, also used for the clustering task. `Etail` is a synthetic online shopping basket dataset, where the edge-dependent node labels indicate the count of each product in each basket. The product return information can be used as ground truth for the product return prediction task.

### G.3 DATASETS FOR TRADITIONAL NODE CLASSIFICATION

For the traditional node classification task, we use six real-world datasets from (Wang et al., 2023a). In `Senate` (Fowler, 2006b), nodes represent individual US Senators, while each edge connects the sponsor and co-sponsors of a bill introduced in the Senate. The node labels are the political party affiliation of each person. In `House` (Chodrow et al., 2021), nodes represent US House of Representatives and edges represent the groups of members of the same committee. The node labels are the political party of the representatives. In `Walmart` (Amburg et al., 2020), nodes represent products being purchased, while the edges connect the products that are purchased together. The node labels are the product categories. In `Congress` (Fowler, 2006a), nodes represent US Congress persons and edges represent the sponsor and co-sponsors of legislative bills. `Cora-CA`[4] and `DBLP-CA`[5] are two co-authorship datasets. In these two datasets, each node represents each paper and each edge represents the papers co-authored by the same author. The node labels are the category of the papers.

### G.4 DATASETS FOR APPROXIMATING CO-REPRESENTATION HYPERGRAPH DIFFUSION

For the diffusion operator approximation experiment, we generated semi-synthetic diffusion data using the `Senate` (Fowler, 2006b) dataset with the same initial features $\boldsymbol{X}^{(0)}$ as the experiments in (Wang et al., 2023a). Although our proposed CoNHD model can accept any input features related to the node-edge pairs, we only utilize initial node features to fit the input of most HGNNs. Following (Wang et al., 2023a), we sampled one-dimensional node feature by the Gaussion distribution $\mathcal{N}(\mu, \sigma)$, where $\mu = 0$ and $\sigma$ uniformly sampled from $[1, 10]$. We initialized the features of node-edges using the node features, i.e., $\boldsymbol{H}^{(0)} = \{\boldsymbol{x}_v^{(0)} | v \in \mathcal{V}, e \in \mathcal{E}_v\}$. We then generated the labels $\boldsymbol{H}^{(2)}$ by performing two steps of the co-representation hypergraph diffusion process. We consider three different diffusion operators: CE (Zhou et al., 2007), TV ($p = 2$) (Hein et al., 2013), and LEC ($p = 2$) (Jegelka et al., 2013). We applied gradient descent for the differential diffusion operator CE, and ADMM for the non-differential diffusion operators TV and LEC. We set equal weights for the node and edge regularization functions, i.e., $\lambda = \gamma = 1$. We chose $\alpha$ and $\rho$ to make the variance ratio $\mathrm{Var}(\boldsymbol{H}^{(2)})/\mathrm{Var}(\boldsymbol{H}^{(0)})$ in a similar scale. Specifically, we set the step size $\alpha = 0.06$ for CE in gradient descent, and set the scale factor $\rho = 0.07$ for TV and $\rho = 0.5$ for LEC in the ADMM optimization process. To avoid the node features exposed in the training process, we generated 100 pairs $(\boldsymbol{H}^{(0)}, \boldsymbol{H}^{(2)})$ using the same hypergraph structure, where 20 pairs are for the validation set and 20 pairs are for the test set.

## H IMPLEMENTATION DETAILS

To ensure a fair comparison, we follow the experimental setup for edge-dependent node classification in (Choe et al., 2023). All models are tuned using grid search. Specifically, the learning rate is chosen from $\{0.0001, 0.001\}$ and the number of layers is chosen from $\{1, 2\}$. The batch size is set from $\{256, 512\}$ for the `Coauth-AMiner` dataset due to the large node number, while for other datasets the batch size is chosen from $\{64, 128\}$. To maintain consistent computational cost across methods, we fix the embedding dimension for node and edge representations in baseline methods, and co-representations in the proposed CoNHD method, to $128$. The dropout rate is set to $0.7$. We run the models for 100 epochs with early stopping. For the implementation of the ISAB operator, we set the number of inducing points to $4$ as WHATsNet, and use $2$ attention layers. During training, we sample 40 neighboring edges of a node, while we do not sample neighboring nodes of an edge since the final label is related to all nodes in an edge. HCHA (Bai et al., 2021) and HNN (Aponte

---

[4] `https://people.cs.umass.edu/~mccallum/data.html`
[5] `https://www.aminer.cn/citation`

et al., 2022) are run in full-batch training with more epochs as in (Choe et al., 2023). As different diffusion steps utilize the same diffusion operators in hypergraph diffusion, we share the weights in different layers of the proposed CoNHD model. We employ the same relative positional encoding as in the experiments of WHATsNet (Choe et al., 2023), which has shown effectiveness in predicting edge-dependent node labels. For traditional GNN methods, we transform the hypergraph into a bipartite graph, where the new nodes represent the nodes and hyperedges in the original hypergraph. The edge-dependent node labels are predicted using the learned features of the new edges in the bipartite graph.

For the diffusion operator approximation experiment, we use the `Senate` (Fowler, 2006b) dataset with 1-dimensional feature initialization (Wang et al., 2023a). As shown in Proposition 2, the node-representation hypergraph diffusion in (Wang et al., 2023a) is a special case of the co-representation hypergraph diffusion. Here we conduct experiments using the more general co-representation hypergraph diffusion processes. Although our proposed CoNHD model can accept any input features related to the node-edge pairs, we only utilize initial node features to fit the input of other baseline HGNNs. To generate the labels, we perform the co-representation hypergraph diffusion process using three common structural regularization functions: `CE` (Zhou et al., 2007), `TV` (Hein et al., 2013; Hayhoe et al., 2023), and `LEC` (Jegelka et al., 2013; Veldt et al., 2023). Most hyperparameters follow the same setting as the ENC experiment. To ensure the expressive power of all models, we use a relatively large embedding dimension 256. The number of layers is fixed to 2, which is consistent with the steps of the diffusion process for generating the labels.

We conduct all experiments on a single NVIDIA A100 GPU with 40GB of GPU memory. To ensure statistically significant results, we repeat each experiment with 5 different random seeds and report the mean performance along with the standard deviation.

# I   SUPPLEMENTARY EXPERIMENTAL RESULTS

## I.1   APPLICATION TO DOWNSTREAM TASKS

The ENC task has been shown to be beneficial for many downstream applications (Choe et al., 2023). In this experiment, we investigate whether the edge-dependent labels predicted by our CoNHD method can enhance performance on these downstream tasks.

**Setup.** We follow (Choe et al., 2023) and conduct experiments on three specific tasks: Ranking Aggregation (`Halo`, `H-Index`), Clustering (`DBLP`, `AMiner`), and Product Return Prediction (`Etail`). For these downstream tasks, the ENC labels are first predicted and then used as supplementary input to enhance performance of other algorithms. It is important to note that the extent of improvement relies not only on the performance of the models in the ENC task, but also on the relevance between the downstream task and the ENC task. Therefore, the performance of the downstream task can demonstrate the practical usefulness of the ENC task, but does not directly reflect the performance of the models. We present the dataset statistics in Table A1 and provide a detailed description in Appendix G.2.

For Ranking Aggregation, the predicted edge-dependent node labels are used as edge-dependent weights, which are input into a random-walk-based method (Chitra & Raphael, 2019) to predict the global ranking results. For Clustering, the hypergraph clustering algorithm RDC-Spec (Hayashi et al., 2020) is employed to predict the clustering results of all publications, with the edge-dependent node labels serving as weights for each author (node) in each paper (hyperedge). For Product Return Prediction, the HyperGo algorithm (Li et al., 2018) is used to predict the product return probability based on the counts of each product in each basket (edge-dependent node labels).

**Results.** We present the ENC prediction results on the downstream datasets in Table A2 and the performance on downstream tasks in Table A3. Similar to the results in the main ENC experiments in Section 5.1, Table A2 demonstrates that our method consistently achieves superior performance on ENC tasks across all datasets.

Table A3 shows that incorporating predicted edge-dependent node labels as additional information improves downstream task performance compared to cases where these labels are not used. Furthermore, compared to the best baseline WHATsNet, our method delivers better downstream performance across all three tasks. This improvement can be attributed to the higher performance of

Table A2: **Performance of edge-dependent node classification on downstream datasets.**

| Method | Halo | | H-Index/AMiner | | DBLP | | Etail | |
|---|---|---|---|---|---|---|---|---|
| | Micro-F1 | Macro-F1 | Micro-F1 | Macro-F1 | Micro-F1 | Macro-F1 | Micro-F1 | Macro-F1 |
| WHATsNet | $0.377 \pm 0.002$ | $0.352 \pm 0.006$ | $0.631 \pm 0.027$ | $0.561 \pm 0.044$ | $0.625 \pm 0.092$ | $0.553 \pm 0.128$ | $0.622 \pm 0.004$ | $0.461 \pm 0.007$ |
| CoNHD (*ours*) | $\mathbf{0.396} \pm 0.003$ | $\mathbf{0.381} \pm 0.007$ | $\mathbf{0.661} \pm 0.027$ | $\mathbf{0.605} \pm 0.040$ | $\mathbf{0.768} \pm 0.094$ | $\mathbf{0.740} \pm 0.127$ | $\mathbf{0.751} \pm 0.008$ | $\mathbf{0.696} \pm 0.008$ |

Table A3: **Performance on Downstream Tasks.**

(a) Ranking Aggregation (Acc.↑)

| Method | Halo | H-Index |
|---|---|---|
| RW w/o Labels | 0.532 | 0.654 |
| RW w/ WHATsNet | 0.714 | 0.693 |
| RW w/ CoNHD | **0.723** | **0.695** |
| RW w/ GroundTruth | 0.711 | 0.675 |

(b) Clustering (NMI↑)

| Method | DBLP | AMiner |
|---|---|---|
| RDC-Spec w/o Labels | 0.163 | 0.338 |
| RDC-Spec w/ WHATsNet | 0.184 | 0.352 |
| RDC-Spec w/ CoNHD | 0.196 | 0.354 |
| RDC-Spec w/ GroundTruth | **0.221** | **0.359** |

(c) Product Return (F1↑)

| Method | Etail |
|---|---|
| HyperGO w/o Labels | 0.718 |
| HyperGO w/ WHATsNet | 0.723 |
| HyperGO w/ CoNHD | 0.733 |
| HyperGO w/ GroundTruth | **0.738** |

our method on the ENC prediction tasks, leading to higher quality predicted edge-dependent node labels. Interestingly, in the ranking aggregation task, the downstream performance using predicted labels even surpasses that achieved using the ground truth labels. This suggests that the ground truth labels may contain some noise, while the predicted labels better capture the underlying smooth structure of the label space and further can enhance the downstream task performance.

## I.2 TRADITIONAL NODE CLASSIFICATION TASK

While our method is specifically designed for learning co-representations and is naturally suited for the ENC task, it can also be extended to address other tasks. In this experiment, we explore the potential of our proposed CoNHD on the traditional node classification task, which the most common task in existing hypergraph learning research.

**Setup.** We conduct experiments on both synthetic and real-world datasets. The synthetic datasets are generated with varying levels of controlled heterophily, following the synthetic strategy in (Wang et al., 2023a). For real-world datasets, we utilize six datasets from (Wang et al., 2023a). The dataset statistics are presented in Table A1. It is important to note that some of the datasets in (Wang et al., 2023a) contain a significant proportion of isolated nodes (up to 80 percent), which are not connected to any other nodes in the hypergraph. This means that performance on these datasets is largely dominated by these isolated nodes, making them less effective for evaluating the true capability of a hypergraph learning algorithm. Consequently, we focus on the six datasets with a higher proportion of connected nodes.

**Results.** Table A4 presents the accuracy results on the synthetic heterophily datasets. All methods perform better performance in homophily scenarios with lower levels of heterophily, while performance declines as the heterophily level increases. Overall, our method consistently achieves the best performance across all cases. This can be attributed to the separate co-representations in our method, which prevent mixing of heterophic information among neighbouring nodes.

Table A5 reports the accuracy results on real-world datasets. Our method achieves the best performance on most of the datasets, while on the remaining two datasets, it still delivers highly competitive accuracy compared to the best baselines. Although our method is specifically designed for ENC, these results demonstrate its general applicability and potential beyond the ENC task. Additionally, in the experiments, we simply aggregate co-representations according to the same node using the mean function to generate node representations. More effective aggregation strategies can be further explored in the future research.

## I.3 PERFORMANCE OF CONSTRUCTING DEEP HGNNS

Oversmoothing is a well-known challenge in constructing deep HGNNs (Wang et al., 2023a; Yan et al., 2024), which hinders the utilization of long-range information and limits the model performance. To examine whether our method can alleviate the oversmoothing issue, we conduct experiments on the ENC task using HGNNs with different number of layers.

Table A4: **Accuracy of node classification on synthetic heterophily datasets.**

| Method | heterophily level | | | | | | |
|--------|---|---|---|---|---|---|---|
| | 1 | 2 | 3 | 4 | 5 | 6 | 7 |
| AllSet | $95.58 \pm 0.86$ | $91.96 \pm 0.92$ | $87.21 \pm 1.02$ | $81.73 \pm 1.83$ | $76.06 \pm 1.78$ | $69.08 \pm 1.42$ | $64.66 \pm 2.69$ |
| ED-HNN | $96.14 \pm 0.45$ | $92.34 \pm 0.48$ | $87.88 \pm 0.59$ | $83.01 \pm 0.87$ | $77.70 \pm 0.93$ | $72.69 \pm 1.38$ | $70.09 \pm 1.93$ |
| WHATsNet | $97.22 \pm 0.35$ | $93.45 \pm 0.62$ | $89.33 \pm 0.70$ | $84.02 \pm 0.92$ | $78.20 \pm 1.42$ | $72.78 \pm 1.70$ | $70.59 \pm 1.62$ |
| CoNHD (*ours*) | $\mathbf{98.21 \pm 0.23}$ | $\mathbf{95.10 \pm 0.34}$ | $\mathbf{90.75 \pm 0.39}$ | $\mathbf{84.97 \pm 0.79}$ | $\mathbf{78.51 \pm 0.71}$ | $\mathbf{73.90 \pm 1.32}$ | $\mathbf{71.13 \pm 1.70}$ |

Table A5: **Accuracy of node classification on real-world datasets.**

| Method | Senate | House | Walmart | Congress | Cora-CA | DBLP-CA |
|--------|--------|-------|---------|----------|---------|---------|
| AllSet | $51.83 \pm 5.22$ | $69.33 \pm 2.20$ | $65.46 \pm 0.25$ | $92.16 \pm 1.05$ | $83.63 \pm 1.47$ | $91.53 \pm 0.23$ |
| ED-HNN | $64.79 \pm 5.14$ | $\mathbf{72.45 \pm 2.28}$ | $66.91 \pm 0.41$ | $\mathbf{95.00 \pm 0.99}$ | $83.97 \pm 1.55$ | $91.90 \pm 0.19$ |
| WHATsNet | $64.93 \pm 4.99$ | $71.89 \pm 3.08$ | $67.53 \pm 0.57$ | $91.72 \pm 0.63$ | $82.57 \pm 1.39$ | $91.15 \pm 0.25$ |
| CoNHD (*ours*) | $\mathbf{68.73 \pm 5.74}$ | $72.20 \pm 1.57$ | $\mathbf{68.68 \pm 0.57}$ | $94.88 \pm 1.12$ | $\mathbf{84.00 \pm 1.09}$ | $\mathbf{91.99 \pm 0.19}$ |

**Setup.** A series of models with varying depths, ranging from 1 to 64, are trained and evaluated on the `Citeseer-Outsider` dataset. We compare CoNHD with WHATsNet, the overall best-performing baseline for the ENC task. We also include EDHNN and HDS$^{ode}$ in our comparison, which are two recent HGNNs that have been shown to mitigate the oversmoothing problem.

**Results.** As shown in Fig. A2, the performance of WHATsNet drops sharply when the depth exceeds 4 layers. In contrast, the performance of EDHNN and HDS$^{ode}$ remains stable as the number of layers increases, but they do not demonstrate significant gains with deeper architectures. Our proposed CoNHD method, in contrast, continues to improve the performance as the number of layers increases, and the performance converges after 16 layers. This suggests that CoNHD benefits from deeper architectures, effectively leveraging long-range information to enhance performance. Both EDHNN and CoNHD are based on hypergraph diffusion, which has

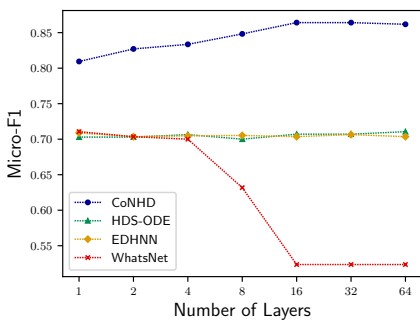

Figure A2: **Performance of HGNNs with varying numbers of layers on the `Citeseer-Outsider` dataset.** CoNHD achieves the best performance across all settings.

demonstrated potential in addressing the oversmoothing issue (Wang et al., 2023a; Chamberlain et al., 2021). Additionally, CoNHD further introduces co-representations, allowing the same node to have distinct representations when interacting within different hyperedges. This approach ensures that diffused information remains diverse, preventing the learned representations from becoming uniform, thereby helping to mitigate the oversmoothing issue.

### I.4 MORE VISUALIZATIONS OF THE LEARNED EMBEDDINGS

We visualize the learned embeddings of node-edge pairs on the `Email-Enron` dataset using LDA. As the same node can have different labels in different edges, we choose the three largest-degree nodes and present the node-edge embeddings associated with each of them. For the small-degree nodes, as these nodes are incident in fewer hyperedges, we visualize the node-edge embeddings of the total 300 smallest-degree nodes. As shown in Fig. A3, CoNHD can learn more separable embeddings compared to the best baseline method WHATsNet on large-degree nodes. For the small-degree nodes, the embeddings from both methods can show clear distinction based on the edge-dependent node labels. CoNHD implements the interactions as multi-input multi-output functions, which can preserve specific information for each node-edge pair and avoid potential information loss. This leads to significant performance improvements on the ENC task, especially for complex hypergraphs with large-degree nodes and edges.

### I.5 ABLATION EXPERIMENTS ON THE DIRECT INTERACTIONS

Traditional message passing-based HGNNs only model the interactions between nodes and edges, neglecting direct interactions among nodes and among edges. Pei et al. (2024) empirically demonstrate that incorporating such direct interactions can enhance the performance of HGNNs on the traditional node classification task. However, the impact of direct interactions on the ENC task re-

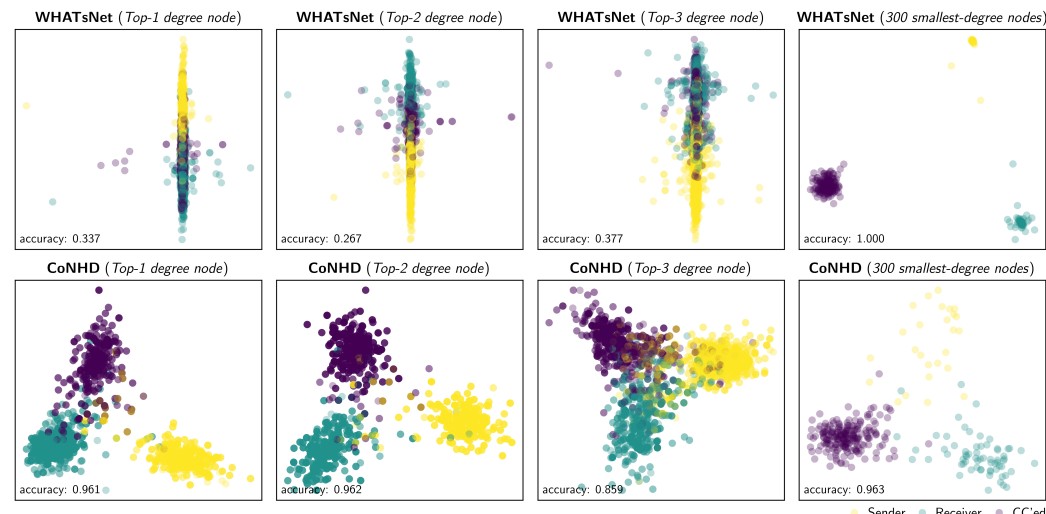

Figure A3: **More visualizations of embeddings in the `Email-Enron` dataset using LDA.** WHATsNet fails to learn separable embedding for node-edge pairs associated with large-degree nodes, while the embeddings learned by CoNHD exhibit clearer distinctions. For small-degree nodes, both methods can learn separable embeddings for node-edge pairs.

Table A6: **Effectiveness of the direct interactions among nodes and among edges.** "D.I." denotes "Direct Interactions".

| Method | Email-Enron | | Email-Eu | | Cora-Outsider | |
| --- | --- | --- | --- | --- | --- | --- |
| | Micro-F1 | Macro-F1 | Micro-F1 | Macro-F1 | Micro-F1 | Macro-F1 |
| CoNHD w/o D.I. | $0.897_{\pm 0.001}$ | $0.848_{\pm 0.003}$ | $0.707_{\pm 0.001}$ | $0.688_{\pm 0.002}$ | $0.749_{\pm 0.010}$ | $0.745_{\pm 0.010}$ |
| CoNHD w/ D.I. | $\mathbf{0.911}_{\pm 0.001}$ | $\mathbf{0.871}_{\pm 0.002}$ | $\mathbf{0.709}_{\pm 0.001}$ | $\mathbf{0.690}_{\pm 0.002}$ | $\mathbf{0.800}_{\pm 0.019}$ | $\mathbf{0.797}_{\pm 0.020}$ |

| Method | DBLP-Outsider | | Citeseer-Outsider | | Pubmed-Outsider | |
| --- | --- | --- | --- | --- | --- | --- |
| | Micro-F1 | Macro-F1 | Micro-F1 | Macro-F1 | Micro-F1 | Macro-F1 |
| CoNHD w/o D.I. | $0.828_{\pm 0.008}$ | $0.825_{\pm 0.009}$ | $0.813_{\pm 0.018}$ | $0.810_{\pm 0.018}$ | $0.863_{\pm 0.004}$ | $0.861_{\pm 0.004}$ |
| CoNHD w/ D.I. | $\mathbf{0.903}_{\pm 0.002}$ | $\mathbf{0.902}_{\pm 0.002}$ | $\mathbf{0.828}_{\pm 0.010}$ | $\mathbf{0.826}_{\pm 0.010}$ | $\mathbf{0.899}_{\pm 0.004}$ | $\mathbf{0.898}_{\pm 0.004}$ |

mains unexplored. In this section, we conduct experiments to evaluate the performance differences of our method on the ENC task with and without these direct interactions among nodes and edges.

**Setup.** In our proposed method, the interactions among co-representations naturally encompass not only interactions between nodes and edges but also direct interactions among nodes and among edges. To isolate the effects of direct interactions among nodes and edges, we treat each co-representation $h_{v,e}^{(t)}$ as the concatenation of the node part $x_{v,e}^{(t)}$ and the edge part $z_{v,e}^{(t)}$, i.e., $h_{v,e}^{(t)} = [x_{v,e}^{(t)}, z_{v,e}^{(t)}]$. Notably, unlike the separate node and edge representations in message passing-based HGNNs, the node part $x_{v,e}^{(t)}$ and the edge part $z_{v,e}^{(t)}$ here remain specific to each node-edge pair. Consequently, each node and edge still maintains multiple representations that adapt to their respective degrees. This ensures that the advantages of adaptive representation size and adaptive diffusion of information are preserved in the modified variant. The only difference lies in the reduction of direct interactions among nodes and among edges.

In this experiment, we adopt the better-performing ISAB implementation of our proposed CoNHD method. Two $\phi$ functions and two $\varphi$ functions are utilized to independently generate within-edge and within-node information for the two parts. The generated information from the node part is used to update the edge part $z_{v,e}^{(t+1)}$, while the generated information from the edge part is employed to update the node part. We conduct experiments on six ENC datasets with relatively large node degrees, while most of them also have relatively large edge degrees. These datasets might incorporate more higher-order interactions, making them well-suited to study the performance differences with and without direct interactions.

**Results.** As shown in Table A6, our proposed CoNHD model, which incorporates direct interactions among nodes and edges, outperforms the variant without these interactions. This highlights the ef-

fectiveness of direct interactions in the ENC task. The degree of improvement varies across datasets. For simpler datasets with relatively low edge degrees, such as `Email-Eu`, the performance gains are less pronounced compared to those observed on more complex datasets. These simpler datasets contain fewer higher-order interactions, limiting the ability to fully demonstrate the benefits of direct interactions. This observation further supports that our method achieves greater improvements on complex datasets with large node and edge degrees, aligning with the conclusion drawn from our main experiments.

## J    DIFFUSION WITH NON-DIFFERENTIABLE REGULARIZATION FUNCTIONS

**Optimization with ADMM method.** When the regularization functions are not all differentiable (*e.g.*, the total variation (TV) regularization functions (Hein et al., 2013; Hayhoe et al., 2023) or the Lovász extension cardinality-based (LEC) regularization functions (Jegelka et al., 2013; Veldt et al., 2023)), we can apply ADMM to find the optimal solution. We first introduce auxiliary variables $U_e$ and $Z_v$ for each edge and node, respectively. The variables are initialized as $h_{v,e}^{(0)} = a_{v,e}$, $U_e^{(0)} = H_e^{(0)}$, and $Z_v^{(0)} = H_v^{(0)}$, and then iteratively updated as follows:

$$U_e^{(t+1)} = \mathbf{prox}_{\lambda \Omega_e / \rho}(2H_e^{(t)} - U_e^{(t)}) + U_e^{(t)} - H_e^{(t)}, \tag{A5}$$

$$Z_v^{(t+1)} = \mathbf{prox}_{\gamma \Omega_v / \rho}(2H_v^{(t)} - Z_v^{(t)}) + Z_v^{(t)} - H_v^{(t)}, \tag{A6}$$

$$h_{v,e}^{(t+1)} = \mathbf{prox}_{\mathcal{R}_{v,e}(\cdot; a_{v,e})/2\rho}\Big(\frac{1}{2}\big([U_e^{(t+1)}]_v + [Z_v^{(t+1)}]_e\big)\Big). \tag{A7}$$

Here $\mathbf{prox}_g(I) := \arg\min_{I'} \big(g(I') + \frac{1}{2}\|I' - I\|_F^2\big)$ is the proximity operator (Boyd et al., 2011) of a function $g$, in which $\|\cdot\|_F^2$ denotes the Frobenius norm. The proximity operator of a lower semi-continuous convex function is 1-Lipschitz continuous (Parikh et al., 2014), enabling its approximation by neural networks. $\rho$ is the scaling factor in the ADMM method. We leave the derivation of Eq. A5-A7 to Appendix B.1.

**Neural Implementation.** We provide a variant of our CoNHD model following the update rules of the ADMM optimization in Eq. A5-A7The $(t + 1)$-th layer can be represented as:

$$\texttt{ADMM-based:} \qquad M_e^{(t+1)} = \phi(2H_e^{(t)} - M_e^{(t)}) + M_e^{(t)} - H_e^{(t)}, \tag{A8}$$

$$M_v'^{(t+1)} = \varphi(2H_v^{(t)} - M_v'^{(t)}) + M_v'^{(t)} - H_v^{(t)}, \tag{A9}$$

$$h_{v,e}^{(t+1)} = \psi([m_{v,e}^{(t+1)}, m_{v,e}'^{(t+1)}, h_{v,e}^{(0)}]). \tag{A10}$$

Here we use the same notations as the GD-based implementation in Section 4.2. $M_e^{(t)} = [m_{v_1^e,e}^{(t)}, \ldots, m_{v_{d_e}^e,e}^{(t)}]^\top$ and $M_v'^{(t)} = [m_{v,e_1^v}'^{(t)}, \ldots, m_{v,e_{d_v}^v}'^{(t)}]$ are the within-edge and within-node diffusion information generated using the neural diffusion operators $\phi$ and $\varphi$, which can be implemented by any permutation equivariant network. $\psi(\cdot)$ is implemented as a linear layer, which collects diffusion information and updates the co-representations.

Due to the dependency on historical auxiliary variables, the ADMM-based implementation needs to preserve the historical diffusion information $M_e^{(t)}$ and $M_v'^{(t)}$ from the last step. This results in higher memory consumption compared to the GD-based implementation, as also discussed in Appendix C of (Wang et al., 2023a). In Section 5.2, we compare the empirical performance of the ADMM-based and GD-based implementations in approximating operators derived from differentiable and non-differentiable regularization functions. While the ADMM-based implementation shows slightly better performance for the non-differentiable case, the performance gap between the two implementations is minimal. Consequently, we adopt the simpler GD-based implementation for the majority of our experiments.

