# OpenReview forum: "Co-Representation Neural Hypergraph Diffusion for Edge-Dependent Node Classification"
_ICLR.cc/2025/Conference — Submitted to ICLR 2025_

### Official Review · Reviewer_WHeD · 2024-11-01

**Soundness:** 2
**Presentation:** 3
**Contribution:** 1
**Rating:** 3
**Confidence:** 4

**Summary:**

The paper introduces Co-representation Neural Hypergraph Diffusion (CoNHD) as a novel solution for edge-dependent node classification (ENC) in hypergraphs. Experimental results demonstrate that CoNHD outperforms the selected baselines.

**Strengths:**

The paper is well-organized and easy to follow, and the proposed model demonstrates superior empirical performance compared to the baselines in the selected setting.

**Weaknesses:**

1. There is a lack of rigorous guarantees that the three main limitations mentioned in the introduction are thoroughly addressed. Regarding the non-adaptive representation size, to my understanding, the weight matrix W in the final classification layer of the proposed model appears to have fixed dimensions and does not adapt based on the node degree. Concerning the non-adaptive messages, there lack theoretical or empirical results to support that the proposed approach can indeed produce distinct features for the same node across different hyperedges. For example, how different are $h_{v1,e2}$ and $h_{v1,e2}$. As for the insufficient direct interactions, node-to-edge and edge-to-node message-passing operations indirectly exchange information between connected nodes and edges. What additional information gain does the proposed operation provide compared to these node-to-edge and edge-to-node message passing? Is this gain purely intuitive, or is there any theoretical support?

2. The real-world value of this paper remains unclear. To my knowledge, [1] is the first paper to study the hypergraph edge-dependent node classification (ENC) task. In [1], the authors present the ENC task as a benchmark dataset and suggest its primary real-world application lies in generating informative node features for problems such as ranking aggregation and clustering. Conversely, in this paper, the authors argue that the ENC task itself holds significant real-world importance, providing an example involving the classification of the order of authors within a paper. However, it is difficult to envision a scenario where classifying the order of authors for a given paper would be necessary, as both the co-authorship and the order of authors are typically provided alongside the paper. This raises the question: are there any real-world applications that can be naturally modelled as an ENC task?

[1] Choe, Minyoung, et al. "Classification of edge-dependent labels of nodes in hypergraphs." Proceedings of the 29th ACM SIGKDD Conference on Knowledge Discovery and Data Mining. 2023.

**Questions:**

1. To my understanding, most hypergraph neural networks [1-3] are designed for hyperedge prediction and node classification tasks. The ENC task can be viewed as a sub-task of node classification. I was curious whether this method is effective only within this sub-task or if it could also be useful for other types of tasks.

2. Proposition 3 demonstrates that the proposed method can represent existing message-passing hypergraph neural networks. However, I was wondering if this property necessarily implies that the proposed method is more powerful than existing methods across all hypergraph-related tasks. For example, while an MLP is often considered a universal function approximator, there is no evidence to suggest that an MLP can outperform hypergraph neural networks on all hypergraph-related tasks.

[1] Chien, Eli, et al. "You are AllSet: A Multiset Function Framework for Hypergraph Neural Networks." International Conference on Learning Representations.

[2] Chen, Can, and Yang-Yu Liu. "A survey on hyperlink prediction." IEEE Transactions on Neural Networks and Learning Systems (2023).

[3] Antelmi, Alessia, et al. "A survey on hypergraph representation learning." ACM Computing Surveys 56.1 (2023): 1-38.

---

> ### Author Response · Authors · 2024-11-20
> **Response to Reviewer WHeD**
>
> We thank Reviewer WHeD for the valuable comments and feedback. Below we respond to the concerns point by point.

---

> ### Author Response · Authors · 2024-11-20
> **Response to Weakness 1 (part 1/2)**
>
> **W1:** How our method addresses the three main limitations
>
> Thank you for this insightful comment. Below we clarify the three main limitations and explain how our method addresses each of them.
>
> **(1) Adaptive representation size in our method.**
>
> We respectfully think the reviewer might have some misunderstandings about this point. The representation size in our paper refers to the node/edge representation size, instead of the weight in the final classification layer. In message passing-based HGNNs, each node ($v_i$) only has one fixed-length representation vector ($x_{v_i}$), which is not adaptive to the node degree and can cause potential information loss. In contrast, our method introduces co-representations $(h_{v_i, e_1}, h_{v_i, e_2},..., h_{v_i, e_n})$, where n is equal to the degree of node $v_i$. Although the final classification layer $f(\cdot)$ only accepts a fixed-length input vector, we can use different co-representations as input to predict different edge-dependent node labels (e.g., $y_{v_i, e_1} = f(h_{v_i, e_1})$, $y_{v_i, e_2} = f(h_{v_i, e_2})$). In this case, we ensure that the representation size is adaptive to the node/edge degree, and this size is not related to the fixed-dimensional weight of the final classification layer.
>
> **(2) Adaptive diffusion information in our method.**
>
> First, we want to emphasize that this is a common issue for most existing HGNNs, which is first discussed in [a]. Some recent research has shown that introducing edge-dependent messages (adaptive messages) can help improve the performance on hypergraph-related tasks [a,b,c]. However, in each layer, these methods focus on extracting edge-dependent information from a single node representation, while the extracted information is then aggregated back to a single node representation. Therefore, the model needs to repeat the "extract-aggregate-extract" process in each layer, which increases the learning difficulties. In contrast, our method learns node-edge co-representations directly without any aggregation, avoiding the need of this extraction and aggregation process. This also alleviates some computational burden of our method.
>
> In **Figure 2** in our original manuscript, we visualize the embeddings (i.e., $h_{v_1, e_1}$, $h_{v_1, e_2}$, …) of the largest-degree node ($v_1$). The results not only demonstrate that our method can produce different embeddings for the same node across different hyperedges, but also show that the embeddings exhibit clear distinctions based on the edge-dependent node labels. More visualizations can be found in Appendix I.4.

---

> ### Author Response · Authors · 2024-11-20
> **Response to Weakness 1 (part 2/2)**
>
> **(3) Sufficient direct interactions among nodes or edges in our method.**
>
> We consider the limitation of lacking direct interactions based on recent research [d], which empirically shows the effectiveness of direct interactions among hyperedges in hypergraph learning. Below we demonstrate the advantage of such direct interactions.
>
> Considering two edges $e_1$ and $e_2$ with a common node $v_1$, when adding direct interactions, information can be passed directly from $e_1$ to update the representation of $e_2$, i.e., $z_{e_1} \xrightarrow[]{m_{e_1, e_2}} z_{e_2}$, where $m_{e_1, e_2}$ represents the specific information for $e_2$ that should be obtained from $e_1$ and cannot be obtained from other nodes or edges. In traditional message passing, information need to be passed through the intermediate node, i.e., $z_{e_1} \xrightarrow[]{m_{e_1, v_1}} x_{v_1} \xrightarrow[]{m_{v_1, e_2}} z_{e_2}$. To avoid information loss in $m_{v_1, e_2}$ compared to $m_{e_1, e_2}$, we need to ensure each preceding variables, i.e., $x_{v_1}$ and $m_{e_1, v_1}$, contains the complete information of $m_{e_1, e_2}$. When information needed in $m_{e_1, e_2}$ is large, this information may be lost in the preceding variables as they need to carry other information (e.g., information about node $v_1$). Additionally, considering the reverse direction of $z_{e_2} \xrightarrow[]{m_{e_2, e_1}} z_{e_1}$, then in traditional message passing, $x_{v_1}$ should also carry information of $m_{e_2, e_1}$. This means the intermediate node $x_{v_1}$ should carry information for the interactions among all neighboring edges, which is more likely to cause information loss. In contrast, with direct interactions, messages do not need to pass through this intermediate node and therefore can solve the problem.
>
> In our method, the interactions among co-representations naturally include direct interactions among nodes or among edges. Since the information gain is highly related to the extent that the interactions among nodes or edges can contribute to task performance in a specific dataset, we added an ablation experiment to empirically compare the performance with and without direct interactions. We provide more details in Appendix I.5 in the updated manuscript. The results are as follows:
>
> - *Micro-F1 Results*
>
> |Method|Email-Enron|Email-Eu|Cora-Outsider|DBLP-Outsider|Citeseer-Outsider|Pubmed-Outsider|
> |---|---|---|---|---|---|---|
> |CoNHD w/o Direct interactions|0.897±0.001|0.707±0.001|0.749±0.010|0.827±0.008|0.813±0.018|0.863±0.004|
> |CoNHD w/ Direct interactions|**0.911±0.001**|**0.709±0.001**|**0.800±0.019**|**0.903±0.002**|**0.828±0.010**|**0.899±0.004**|
>
> - *Macro-F1 Results*
>
> |Method|Email-Enron|Email-Eu|Cora-Outsider|DBLP-Outsider|Citeseer-Outsider|Pubmed-Outsider|
> |---|---|---|---|---|---|---|
> |CoNHD w/o Direct interactions|0.848±0.003|0.688±0.002|0.745±0.010|0.825±0.009|0.810±0.018|0.861±0.004|
> |CoNHD w/ Direct interactions|**0.871±0.002**|**0.690±0.002**|**0.797±0.020**|**0.902±0.002**|**0.826±0.010**|**0.898±0.004**|
>
> As shown in the tables, direct interactions enhance performance on the ENC datasets. For simpler datasets with relatively low edge degrees, such as Email-Eu, the performance gain is less pronounced compared to that on other datasets. These simpler datasets contain fewer higher-order interactions, limiting the ability to fully demonstrate the benefits of direct interactions. This observation further supports that our method can achieve greater improvements on complex datasets with large node and edge degrees, which is consistent with the findings from our main results.

---

> > ### Author Response · Authors · 2024-11-20
> > **Response to Weakness 2**
> >
> > **W2:** The real-world value of the ENC task
> >
> > We apologize for not making the real-world value of the ENC task more explicit in our original manuscript due to the page limit. We have revised our manuscript, especially the introduction part, to better highlight the significance of the ENC task. In our original manuscript, the example of co-authorship network is to help the readers better understand the ENC task. Since we study the same problem as [b] (the paper [1] mentioned by the reviewer), the application value of [b] is also applicable to our work. Below, we discuss the real-world significance of the ENC task and its usefulness for downstream applications in detail.
> >
> > **The ENC task itself is helpful for modeling real-world problems.** In the game industry, as indicated in [b], the ENC task can be used to predict the score or winning probability of each player (node) in each match (hyperedge), and therefore improving the matchmaking system to ensure fair competition. In biology/bioinformatics, the ENC task can be used to uncover the role of a protein (node) in different signal pathways (hyperedge), where the role can be enzyme, regulatory factor, or adaptor protein. **Apart from the ENC task itself, the edge-dependent node labels can also be used as additional features to help improve the downstream tasks** [b], including ranking aggregation, clustering, product return prediction. We have conducted experiments on these downstream tasks following [b], and the results are provided in Appendix I.1 of our original manuscript. The experimental results demonstrate that the edge-dependent node labels predicted by our method can help improve the downstream tasks. Additionally, the labels predicted by our model lead to a more significant performance improvement in downstream tasks compared to the labels predicted by the best baseline WHATsNet. This further supports the real-world value of our work.

---

> > > ### Author Response · Authors · 2024-11-20
> > > **Response to Question 1**
> > >
> > > **Q1:** Applicability of our method beyond the ENC task
> > >
> > > Below we first clarify why we choose the ENC task as our main study topic and then discuss the applicability of our method beyond the ENC task.
> > >
> > > **(1) The value of the ENC task for both research and applications.**
> > >
> > > We want to clarify that the ENC task studied in our paper is not a sub-task of traditional node classification. Instead, ENC represents an under-explored area where the predicted label is related to both the node and the hyperedge, which presents significant values for hypergraph research. As demonstrated by recent research [e], on some traditional node classification tasks, simply applying an MLP on the node features without considering the hypergraph structure can also achieve good results. In contrast, in the ENC task, as the node can have different labels across different hyperedges, it forces the model to utilize the hypergraph structure and learn different features of the same node specific to different hyperedges. This necessitates incorporating the hypergraph structure into the modeling process. As indicated in [b], **the ENC task “evaluates the capability of models in capturing features unique to hypergraphs”**. Additionally, this new task **has significant real-world value** (as discussed in our response for W2). Therefore, we choose to study this interesting and valuable topic.
> > >
> > > **(2) Why our method is specifically suited for the ENC task and its applicability to other tasks.**
> > >
> > > The primary difference between the ENC task and other tasks (node-level and edge-level tasks) is that a node can have multiple labels across different hyperedges. Therefore, our proposed co-representation strategy is naturally suited for predicting these labels related to node-edge pairs. Meanwhile, our method can be extended to solve other tasks. For example, we can aggregate co-representations corresponding to the same node to generate a single node representations, i.e., $h_{v_1} = f(h_{v_1, e_1}, h_{v_1, e_2}, \ldots, h_{v_1, e_n})$, and use this node representation to perform traditional node classification. However, **this violates our design principle of avoiding aggregation**, which can cause non-adaptive representation size and potential information loss. Additionally, in traditional node classification, as a node only has one label, the multiple co-representations of a single node may not provide huge improvement on such a task.
> > >
> > > Nevertheless, in our original manuscript, we have provided experimental results on traditional node classification tasks in **Appendix I.2**, which is the most common task in existing hypergraph research. The results demonstrate the superior performance of our method compared to other baselines, highlighting its potential and applicability beyond the ENC task.

---

> > > > ### Author Response · Authors · 2024-11-20
> > > > **Response to Question 2**
> > > >
> > > > **Q2:** The expressiveness and effectiveness of our method
> > > >
> > > > We thank the reviewer for recognizing the expressiveness of our method supported by Proposition 3 in our manuscript. Below, we analyze the relation between expressiveness and effectiveness for HGNNs, and clarify the practical effectiveness of our proposed method.
> > > >
> > > > **(1) The expressiveness of our model is the foundation for its effectiveness.**
> > > >
> > > > The expressiveness of a model can be related to the richness of the hypothesis space $\mathcal{H}$, which represents the set of functions the model can express. We say model 1 with hypothesis space $\mathcal{H}_1$ is more expressive than model 2 with hypothesis space $\mathcal{H}_2$, if and only if $\mathcal{H}_2 \subset \mathcal{H}_1$. If a target function $f^*$ we want to fit from the data lies in $\mathcal{H}_1$ but not in $\mathcal{H}_2$, i.e., $f^* \in \mathcal{H}_1$ and $f^* \notin \mathcal{H}_2$, then the less expressive model 2 will underfit the data, while the more expressive model 1 has the potential to learn the target function and provide better effectiveness. Therefore, the expressiveness of our model is the foundation for its effectiveness. We have demonstrated that our method provides better expressiveness than message passing-based HGNNs.
> > > >
> > > > **(2) The inductive bias of our model help improve the generalizability on hypergraph tasks.**
> > > >
> > > > While the expressiveness decides whether a model can fit the training data, inductive bias is essential for a model to generalize from a limited set of training data to novel unseen examples. In real-world tasks, the training data is always limited, without infinite data points to help the model learn the target function $f^*$. Therefore, it is critical that the inductive bias can guide a model to leverage the inherent structure or symmetries of the data, and fit a function that is more possible to generalize to unseen data points. For example, CNNs satisfy translation equivariance, allowing them to recognize patterns regardless of their position in the visual field. As a generalization of CNNs, **GNNs/HGNNs leverage the inductive biases of local connectivity presented in graph/hypergraph data** [f], therefore outperform other methods like MLPs on graph/hypergraph data. Specific to hypergraph datasets, as each hyperedge can be regarded as an unordered set of nodes, **another important inductive bias is the set permutation invariance/equivariance.** This means the input ordering of the nodes within a hyperedge should not affect the final outputs. In our method, we carefully design our multi-input multi-output functions in the within-edge and within-node interactions as permutation equivariant functions, which is also not satisfied by a simple MLP.
> > > >
> > > > **(3) Extensive experimental results demonstrate the practical effectiveness of our method.**
> > > >
> > > > While it is infeasible to fully uncover all inherent characteristics across different real-world HGNN datasets that could inform the design of inductive biases, we can use empirical results to demonstrate the general applicability and effectiveness of a model. In Table 1 of our manuscript, we have conducted extensive experiments on ten ENC datasets and reported the empirical results. As supported by the results, **our method achieves the best performance compared to other methods on these real-world datasets without sacrificing the efficiency**, which highly supports the effectiveness of our proposed method.

---

> > > > > ### Author Response · Authors · 2024-11-20
> > > > > **References and Additional Remarks**
> > > > >
> > > > > **References**
> > > > >
> > > > > [a] Wang et al. Equivariant hypergraph diffusion neural operators. In *ICLR*, 2023.
> > > > >
> > > > > [b] Choe et al. Classification of edge-dependent labels of nodes in hypergraphs. In *KDD*, 2023.
> > > > >
> > > > > [c] Aponte et al. A hypergraph neural network framework for learning hyperedge-dependent node embeddings. *arXiv:2212.14077*, 2022.
> > > > >
> > > > > [d] Ye et al. Hyperedge interaction-aware hypergraph neural network. *arXiv:2401.15587*, 2024.
> > > > >
> > > > > [e] Tang et al. Hypergraph-MLP: Learning on hypergraphs without message passing. In *ICASSP*, 2024.
> > > > >
> > > > > [f] Bruna et al. Spectral networks and locally connected networks on graphs. *arXiv:1312.6203*, 2013.
> > > > >
> > > > > ---
> > > > >
> > > > > We hope the responses solve your concern and questions, and we are happy to discuss them if you need further clarification. Thank you!

---

> > > > > > ### Comment · Reviewer_WHeD · 2024-11-26
> > > > > > **Thank you for the rebuttal.**
> > > > > >
> > > > > > Thank you for the rebuttal. After reviewing it, I still have two major concerns regarding this work:
> > > > > >
> > > > > >
> > > > > > 1.The rebuttal highlights the significant real-world value of the task, citing applications in the game industry and biology. However, the model is tested on only one dataset from the game industry, and none of the baselines used for comparison are drawn from the game industry. This raises questions about the alignment between the stated importance of the research problem and the scope of the study presented in the paper. Consequently, the research motivation remains unclear and unconvincing.
> > > > > >
> > > > > > 2. The explanation provided for Weakness 1 lacks rigour. Would it be possible to provide a formal mathematical proof to substantiate the argument made in the rebuttal? Additionally, it seems that WhatsNet [1] could also address the limitations outlined in Section 1. For example, it appears to tackle the first mentioned limitation by incorporating information from both node v1 and hyperedge e1 into the classifier to generate the target node labels, as described in their Eq. (10). As such, the contribution of this work, when compared to WhatsNet, seems to be primarily empirical and incremental.
> > > > > >
> > > > > > These issues leave me with significant concerns about the clarity of the motivation and the novelty of the contributions. Therefore, I will keep my score.
> > > > > >
> > > > > > [1] Choe, Minyoung, et al. "Classification of edge-dependent labels of nodes in hypergraphs." Proceedings of the 29th ACM SIGKDD Conference on Knowledge Discovery and Data Mining. 2023.

---

> > > > > > > ### Author Response · Authors · 2024-11-28
> > > > > > > **Response to the Additional Concerns (part 1/3)**
> > > > > > >
> > > > > > > We thank Reviewer WHeD for the further comments. Below we respond to the additional concerns point-by-point.
> > > > > > >
> > > > > > > ---
> > > > > > >
> > > > > > > **Q1:** Research value of this work
> > > > > > >
> > > > > > > The reviewer might be misled by our writing about the potential value of the ENC research problem for specific real-world challenges. This paper is not meant as an application paper, and our proposed method is a general method for the ENC task without incorporating any special design for specific industry areas, focusing on performance in the ENC task and a set of generic downstream tasks.
> > > > > > >
> > > > > > > The research value of the ENC task has been elaborately discussed in previous work [a]. In our paper, we addressed **the same ENC problem**, conducted experiments on **all existing benchmark ENC datasets and downstream datasets**, and **strictly followed the same experimental setup** as [a]. Utilizing these common benchmark datasets ensures a fair and comprehensive comparison of our method with existing approaches. Our method achieves the best performance on all the ENC datasets and the downstream datasets. The reviewer's concern about the game industry dataset (Halo dataset) in our opinion is thus **unnecessary**. This dataset originates from the baseline paper WHATsNet [a], and we strictly followed the same experimental setup to maintain a fair comparison. We believe all these comprehensive experiments and superior results provide strong support for the research value of our work and its potential for future applications when more curated ENC datasets will come available.
> > > > > > >
> > > > > > > We sincerely hope these clarifications can help the reviewer better assess the value of our work.

---

> > > > > > > > ### Author Response · Authors · 2024-11-28
> > > > > > > > **Response to the Additional Concerns (part 2/3)**
> > > > > > > >
> > > > > > > > **Q2:** Comparison to WHATsNet
> > > > > > > >
> > > > > > > > Our proposed method is designed to address the three major limitations stated in our manuscript. Below, we highlight the advantages of our method compared to WHATsNet from the perspective of these three limitations.
> > > > > > > >
> > > > > > > > **(1) Non-adaptive representation size**
> > > > > > > >
> > > > > > > >  - We first give an intuitive explanation of why WHATsNet fails to achieve adaptive representation size, and then provide a formal analysis.
> > > > > > > >
> > > > > > > >  - As indicated by the reviewer, WHATsNet concatenates the single node representation and edge representation, i.e., $[x_{v_i}, z_{e_j}]$, and inputs it into a classifier to predict the edge-dependent node label, i.e., $y_{v_i, e_j}$. However, simply repeating the same node representation in different combinations will not increase the dimension of the representation space. In this case, all the node vectors of the same node in different combinations are the same, which by its design cannot solve the non-adaptive representation size problem. The node representation is still a fixed-size vector and cannot adapt to the node degree. In contrast, our method allows the co-representation vectors of the same node to be different, and the number of co-representations for a node equals to the node degree, which can provide adaptive representation size.
> > > > > > > >
> > > > > > > >  - We now provide a formal analysis:
> > > > > > > >
> > > > > > > > 	- To simplify the analysis, we set the representation vector dimension to 1, which can be easily extended to high-dimensional cases. We denote the number of nodes and edges in a hypergraph $\mathcal{G}=(\mathcal{V}, \mathcal{E})$ as $n_v$ and $n_e$, respectively. The average node degree is $\bar{d}_{v}$, and the average edge degree is $\bar{d}_e$. We denote the number of node-edge pairs as $n_p$, i.e., $n_p = n_e \cdot \bar{d}_e = n_v \cdot \bar{d}_v$. We assume $n_p > (n_v + n_e)$. This assumption holds if $\bar{d}_v > \frac{n_v+n_e}{n_v}$ or $\bar{d}_e > \frac{n_v+n_e}{n_e}$. This condition is typically satisfied by hypergraph datasets, as shown in the statistics in Table A1.
> > > > > > > >
> > > > > > > > 	- For WHATsNet, the total representation for all the nodes and edges in the hypergraph can be denoted as $h \in \mathbb{R}^{(n_v+n_e)}$. The concatenation operation can be represented by a linear mapping matrix $M \in \mathbb{R}^{2n_p \times (n_v + n_e)}$, which copies the corresponding node/edge representation to form the concatenated results with the total dimension $2n_p$. We denote the concatenated results as $h' = M \cdot h$, where $h' \in \mathbb{R}^{2n_p}$. Since $n_p > (n_v+n_e)$, we have $\mathrm{rank}(M) \le (n_v+n_e)$. This means that $h'$ still lies in the subspace with dimension not larger than $(n_v+n_e)$. This dimension is smaller than $n_p$ and is not related to the node/edge degree.
> > > > > > > >
> > > > > > > > 	- In contrast, for our method, the number of co-representations is exactly equal to the number of node-edge pairs, i.e., $n_p$. Thus, the total co-representation for the hypergraph lies in $\mathbb{R}^{n_p}$. Since $n_p = n_e \cdot \bar{d}_e = n_v \cdot \bar{d}_v$, the total representation size is proportional to the average edge/node degree. Additionally, for a node with degree $d_v$, the number of co-representations for this node is exactly $d_v$. Therefore, our method can ensure adaptive representation size, while WHATsNet cannot. $\square$

---

> ### Author Response · Authors · 2024-11-28
> **Response to the Additional Concerns (part 3/3)**
>
> **(2) Non-adaptive messages**
>
> - WHATsNet aims to solve this limitation by extracting edge-dependent node representation from a single node representation. However, in the output of each layer, the extracted information is then aggregated back to a single node representation. Therefore, the model keeps repeating the "extract-aggregate-extract" process in different layers, and it forces the model to learn how to extract edge-dependent information from a single node. In contrast, our method directly preserves the co-representations and avoids this extraction and aggregation process. This also alleviates some computational burden of our method. Since this advantage is intuitive and its benefit is evident from the different designs of WHATsNet and our methods, no mathematical proof is given here.
>
> **(3) Insufficient direct interactions among nodes or edges**
>
> - Although WHATsNet incorporates a within-attention module to extract the edge-dependent node representations from multiple node representations, these extracted representations are only used as temporary variables to update the edge representation. Therefore, WHATsNet cannot directly pass messages among nodes or edges. The information for the interactions among edges still need to be passed through a hub node with limited representation size, which can cause potential information loss. In contrast, from the design of our method, the interactions are among the co-representations, which includes not only node-edge interactions, but also direct node-node and edge-edge interactions. Therefore, our method by design can address this limitation whereas WHATsNet cannot, and thus no mathematical proof is given.
>
> The performance improvement from addressing these three limitations is highly dependent on specific datasets and affected by many factors. Due to the complexity of real datasets, imposing strong assumptions on data distribution for a mathematical proof is unrealistic, as it would not reflect real-world conditions. Therefore, providing empirical evaluation in this case will better reflect the effectiveness of a method on real datasets. We performed experiments on all benchmark ENC datasets and downstream datasets in previous research [a], which demonstrates that our method can achieve the best performance while preserving efficiency. Our ablation studies also support the effectiveness of our method design.
>
> We hope these additional analysis and clarifications can help the reviewer better understand our method and the contribution of our work.
>
> ---
>
> **References**
>
> [a] Choe et al. Classification of edge-dependent labels of nodes in hypergraphs. In *KDD*, 2023.

---

> > ### Comment · Reviewer_WHeD · 2024-12-02
> >
> > Dear authors,
> >
> > Thank you for the response. First, [1] is the first work to tackle the ENC task. It introduces a toy problem (which is the main benchmark used in the paper) to demonstrate that their model can handle this task in a relatively synthetic setting. Subsequently, it supports the real-world value of this task by experiments on Ranking Aggregation, Clustering, and Product Return Prediction. In my opinion, this is acceptable because the goal of [1] is primarily to make the task known to the community. However, as a follow-up work, the submitted paper dedicates significant space to emphasizing the real-world importance of this task in the introduction. Therefore, in my opinion, if the authors aim to highlight the real-world value of the task to underscore the importance of their work, the experimental section should align with this emphasis and leave enough space to demonstrate these real-world values. Instead, all the experiments in the main manuscript are conducted in the toy setting proposed by [1], while the real-world task is only briefly shown in the appendix. Second, according to the authors’ previous rebuttal, the implementation of the adaptive representation size seems to be based on the input to the final classifier. If so, for generating the y_{vi,ej}, both the classifier in WhatsNet and the classifier in this paper should use input information from x_{vi} and x_{ej}. As a result, WhatsNet also seems capable of providing an adaptive representation size.
> >
> > Overall, the main issue with this work, in my opinion, lies in the disconnection between the motivation and the actual content in two aspects: 1) The paper uses significant space to emphasize the extensive real-world applications of ENC, but it does not discuss in detail how the proposed method brings value to these applications. 2) The introduction dedicates a lot of space to theoretically listing three issues with previous methods, but the discussions of these issues are highly intuitive. The current version still requires substantial revisions. Therefore, I have decided to maintain my score.
> >
> > [1] Choe, Minyoung, et al. "Classification of edge-dependent labels of nodes in hypergraphs." Proceedings of the 29th ACM SIGKDD Conference on Knowledge Discovery and Data Mining. 2023.

---

> > > ### Author Response · Authors · 2024-12-02
> > > **More Clarifications to the Additional Concerns**
> > >
> > > We thank Reviewer WHeD for the further comments. Below we provide more clarifications to the concerns raised by the reviewer.
> > >
> > > Considering the research value, we have indicated that this paper is not meant as an application paper, and our main target is to introduce a general and more effective method for the ENC task compared to [a], which has set the benchmark with the aim to encourage new solutions as ours. While the reviewer describes the benchmark datasets used in [a] as toy datasets, these datasets serve as a standard common benchmark to evaluate the effectiveness of a method on ENC. Additionally, we also included all the real downstream datasets in our appendix I.1. Since the revision period has ended, should we consider this as a suggestion to move these downstream experiments to the main paper in our final version in favor of the description of potential applications?
> > >
> > > For the adaptive representation size, we think the reviewer's interpretation of the methods is incorrect. As stated in our original responses, the representation size specifically refers to the size of the representation space of the nodes and edges. In WHATsNet [a], messages from numerous edges are aggregated to a fixed-size node representation vector. This can cause potential information loss for large-degree nodes, which have more neighboring edges and should have larger representation size. To address the misunderstandings from the initial comments of the reviewer, we have explained that this representation size is not related to the size of the weight matrix of the classification layer. Furthermore, in our additional response, we provided a rigorous proof demonstrating that, in WHATsNet, simply concatenating node and edge representations does not increase the size of the representation space. To be precise, in WHATsNet, the total representation space is still not larger than $(n_v+n_e)$, which is not related to the node degree or edge degree. In contrast, we demonstrated that the size of the co-representations in our method is $n_p = n_e \cdot \bar{d}_e = n_v \cdot \bar{d}_v$, which is larger than $(n_v + n_e)$ and is proportional to both node and edge degrees. For each node with degree $d_v$, the representation size of this node is exactly $d_v$ and thus leads to the adaptive representation size.
> > >
> > > Finally, while we have proved the expressiveness of our method compared to other methods, the generalizability on real datasets depends on a large set of assumptions on the underlying dataset distribution. Given the complexity of real datasets, imposing strong dataset distribution assumptions to prove the effectiveness is unrealistic. We have chosen to provide more intuitive analysis, supplemented by experimental results and ablation studies, to substantiate the effectiveness of our method design. Notably, our method achieves superior performance across all ENC datasets and downstream datasets without sacrificing efficiency.
> > >
> > > We hope our further clarifications can help the reviewer better appreciate our work.
> > >
> > > **References**
> > >
> > > [a] Choe et al. Classification of edge-dependent labels of nodes in hypergraphs. In *KDD*, 2023.

---

> ### Comment · Reviewer_WHeD · 2024-12-02
>
> Dear authors,
>
> Thank you for the response.
>
> First, regarding the WhatsNet [1], for classifying a node $v_i$ within a hyperedge $e_j$, the model inputs the features of both $v_i$ and $e_j$ into an MLP. Based on my understanding of MLPs, the process involves first projecting the features of $v_i$ and $e_j$ into a hidden space to generate a representation that encapsulates information from both $v_i$ and $e_j$, denoted as $h_{v_i, e_j}$. The MLP then uses $h_{v_i, e_j}$ to produce the label logits. Moreover, as mentioned in the rebuttal entitled **Response to Weakness 1 (part 1/2)** from the authors to my **w1**, the adaptive representation size in their method is indeed based on the $h_{v_i, e_j}$. Therefore, based on the authors' rebuttal, in practice, WhatsNet also generates $h_{v_i, e_j}$ just like the proposed method, which enables it to achieve the adaptive representation size.
>
> Additionally, I still think that the actual content of the paper should align with the claims made in the introduction. I recommend the authors consider at least two major modifications: 1. Add more real-world experiments and include them in the experimental section, ensuring that the content related to real-world datasets is at least as substantial as that of the toy settings. For guidance, you may refer to the experimental section of [1]. 2. Add a section in the main paper to provide a point-by-point analysis of why previous methods, particularly WhatsNet, cannot address the limitations mentioned in the introduction, and how the proposed method overcomes these limitations. Overall, I believe this paper requires significant revisions, and thus I will retain my current score.
>
> [1] Choe, Minyoung, et al. "Classification of edge-dependent labels of nodes in hypergraphs." Proceedings of the 29th ACM SIGKDD Conference on Knowledge Discovery and Data Mining. 2023.

---

> ### Author Response · Authors · 2024-12-03
> **Further Response to the Questions and Suggestions**
>
> We thank Reviewer WHeD for the additional comments and suggestions. Below we respond to these points in detail.
>
> For the adaptive representation size, the reviewer might have some misunderstandings about the co-representations $h_{v_i, e_j}$ in our method. These co-representations are learned using the hypergraph structure in the HGNN layers, whereas the representations in the classification layer of WHATsNet only incorporate the concatenation $[x_{v_i}, z_{e_j}]$ as inputs. The information loss in the preceding HGNN layers due to non-adaptive representation size cannot be recovered by the final classification layer, which is unable to leverage the neighboring hypergraph structure. We have proved that, in WHATsNet, the output node and edge representations from the HGNN layers, as well as their concatenations, cannot adapt to the node or edge degrees, which can cause potential information loss for complex hypergraphs with large-degree nodes and edges. The information lost in this process cannot be recovered by a simple classification layer without incorporating hypergraph structural information, and the outputs will still lie in a low-dimensional manifold with dimension not larger than that of the input subspace. In contrast, in our method, the representation size of the co-representations in each HGNN layer is adaptive to both the node and edge degrees, which can naturally solve this limitation.
>
> For the first suggestion about experiments, while the reviewer suggests to "refer to the experimental section of [a]", we did already include results for all experiments in [a], due to page limit partly in the appendix. We apologize that the reviewer seems to have missed them because we didn't make the reference to the appendix explicit enough in the main paper. We will make these experiments more explicit in the main paper in our final version. Regarding some other potential applications, [a] also highlights some other potential applications in the introduction, like toxicity prediction and homonyms classification. We believe it is helpful to have such a discussion in the introduction to encourage future application research in the community, even though curated datasets might be unavailable at the current stage. We will carefully indicate some applications without available datasets as potential applications in our final version.
>
> For the second suggestion about method comparisons, our manuscript primarily focuses on comparisons with message passing-based methods, which naturally include WHATsNet. For the detailed comparison specific to WHATsNet, we have provided the comparisons in the previous responses, particularly from the perspective of the three limitations. We appreciate the reviewer’s suggestion and will incorporate these discussions into our final version.
>
> Finally, we sincerely thank the reviewer for the valuable time and effort in helping improve the quality of our manuscript. We still hope these clarifications can help the reviewer better appreciate our work.
>
> **References**
>
> [a] Choe et al. Classification of edge-dependent labels of nodes in hypergraphs. In *KDD*, 2023.

---

### Official Review · Reviewer_y4P5 · 2024-11-02

**Soundness:** 4
**Presentation:** 3
**Contribution:** 3
**Rating:** 8
**Confidence:** 4

**Summary:**

Traditional Hypergraph Neural Networks (HNNs) struggle with Edge dependent node classification (ENC) tasks, where nodes can have different labels across different hyperedges. To address this issue, the authors propose a new method called Co-Representation Neural Hypergraph Diffusion (CoNHD), which represents node-edge pairs as co-representations, which allows flexible and adaptive handling of ENC. Unlike traditional message-passing methods, CoNHD doesn't rely on fixed representation sizes or non-adaptive messages. The authors develop a hypergraph diffusion mechanism based on neural networks. Experimental results on ten real-world datasets demonstrate the effectiveness and computational efficiency of the proposed approach, with CoNHD consistently outperforming several baseline methods, including both traditional hypergraph neural networks (HGNNs) and edge-dependent methods.

**Strengths:**

1. The paper tackles an important but under-explored problem of edge-dependent node classification (ENC) on hypergraphs, where nodes can have different labels across different hyperedges. This has many real-world applications mentioned in the paper.
2. The proposed CoNHD model is theoretically grounded, with proofs provided for its increased expressiveness compared to message passing methods. It can also approximate traditional hypergraph diffusion processes. The proposed framework does not rely on a fixed representation size, but adapts based on the node or edge degree. This seems like an innovative mechanism, with potential applications beyond the mentioned problem.
3. The authors provide a comprehensive experimental study, with results on ten datasets. The proposed CoNHD outperforms all the baselines, based on Micro-F1 and Macro-F1 metrics. CoNHD is also more efficient wrt training time, as shown in Figure 3.
4. Ablation study provided in the paper compares the different neural operators namely UNB and ISAB, which provides insights into the the effectiveness of multi-output equivariant functions.

**Weaknesses:**

1. The writing is a bit dense, particularly in section 4, it can be made a bit more readable by providing high level intuition before directly jumping into the concepts.
2. I don't think the part related to updates with ADMM is needed in the main paper, it can be moved into Appendix. Anyway, the results provided are for GD based implementation. Authors can use that extra space to improve readability.
3. Authors mention over-smoothing as an issue with the traditional message passing frameworks, but do not provide any experimental evidence as why the proposed method is immune to over-smoothing.

**Questions:**

I have mentioned a few points in the Weakness section, I do not have more specific questions.

---

> ### Author Response · Authors · 2024-11-20
> **Response to Reviewer y4P5**
>
> We thank Reviewer y4P5 for the constructive suggestions and the positive feedback. Below we respond to the suggestions and questions point by point.

---

> ### Author Response · Authors · 2024-11-20
> **Response to Weakness 1&2**
>
> **W1 & W2:** Improving readability and optimizing the structure of the manuscript
>
> We thank the reviewer for these thoughtful writing suggestions. In our original manuscript, we added the ADMM part for the completeness of the whole co-representation hypergraph diffusion theory. We agree with the reviewer that it would be better to move the ADMM part to the appendix to have more space for improving readability. We have reorganized this part and added some high-level intuition at the beginning of Section 4.

---

> ### Author Response · Authors · 2024-11-20
> **Response to Weakness 3**
>
> **W3:** Discussion on the over-smoothing issue and supporting experimental results
>
> In our original manuscript, we have provided experimental results of overcoming the over-smoothing issue in Appendix I.3. We have revised our manuscript and made these additional experiments more explicitly described in the main text.
>
> Diffusion-inspired (hyper)graph neural networks model the convolution as the optimization process for a target regularization function [a, b], which can avoid converging to an over-smoothed solution as the number of layers increases [c]. Empirical results from prior work [a] have demonstrated the robustness of diffusion-based models against the over-smoothing issue. Our method, built on hypergraph diffusion, shares the same advantage. Moreover, the co-representations in our method allow the same node to have distinct representations when interacting within different hyperedges. This ensures that the diffusing information can be diverse, preventing the learned representations from becoming uniform and further mitigating the over-smoothing issue. As illustrated in Figure A2 in Appendix I.3, the performance of the best baseline, WHATsNet, drops sharply when the depth exceeds 4 layers. In contrast, our method continues to improve performance up to 16 layers and remains stable even as the number of layers increases to 64. These results demonstrate that **the proposed CoNHD method is better for mitigating the over-smoothing issue and utilizing long-range information**.

---

> > ### Author Response · Authors · 2024-11-20
> > **References and Additional Remarks**
> >
> > **References**
> >
> > [a] Wang et al. Equivariant hypergraph diffusion neural operators. In *ICLR*, 2023.
> >
> > [b] Chamberlain et al. Grand: Graph neural diffusion. In *ICML*, 2021.
> >
> > [c] Yang et al. Graph neural networks inspired by classical iterative algorithms. In *ICML*, 2021.
> >
> > ---
> >
> > We hope the responses can further solidify your positive outlook on our work, and we are happy to discuss them if you need further clarification. Thank you!

---

> > > ### Comment · Reviewer_y4P5 · 2024-11-26
> > > **Thank you for the responses**
> > >
> > > I thank the reviewers for their responses, and efforts to update the draft. I keep my score the same.

---

> > > > ### Author Response · Authors · 2024-11-26
> > > > **Thank You for Your Further Comments**
> > > >
> > > > Dear Reviewer y4P5,
> > > >
> > > > Thank you for your further comments. We greatly appreciate the time and effort you have devoted to reviewing our work and helping us improve the quality of our manuscript.
> > > >
> > > > Best regards,
> > > >
> > > > Authors of Submission 5147

---

### Official Review · Reviewer_JdfB · 2024-11-04

**Soundness:** 2
**Presentation:** 2
**Contribution:** 2
**Rating:** 5
**Confidence:** 4

**Summary:**

This paper proposes a method, called CoNHD to tackle edge-dependent node classification tasks (ENC). Unlike node-level or edge-level tasks, in ENC, each node may have a different label across different hyperedges. The authors identify three limitations for the existing models to handle ENC tasks: (1) non-adaptive representation size, (2) non-adaptive messages, and (3) insufficient direct interactions among nodes or edges. To tackle these challenges, this paper introduces CoNHD which models both within-edge and within-node interactions as multi-input multi-output functions.

**Strengths:**

1. The overall presentation is fair, and audiences can capture the key points in the paper.
2.  The proposed task is intresting and remains underexplored within the research field.

**Weaknesses:**

1. The first concern is regarding the novelty of the proposed model. The proposed method is pretty similar to Line Expansion and MultiSet. Besides, the mini-batch strategy is also introduced in MultiSet. How do LEGCN and MultiSet perform on the benchmark ENC datasets?
2. Although the downstream task, i.e., ENC is interesting, can authors further discuss why this task and why this an important topic to study? Because I think the motivations in the paper are also applicable to general node-level/hyperedge-level tasks.
3. I am still concerned about the efficiency issue, in terms of time and space compelxity, as the method expand every node-hyperedge pair if the node is in the hyperedge. Can authors further discuss the time complexity and space compleity with respect to the hypergraphs, i.e., number of nodes, number of hyperedges, and degrees.

**Questions:**

See weaknesses above.

---

> ### Author Response · Authors · 2024-11-20
> **Response to Reviewer JdfB**
>
> We thank Reviewer JdfB for the valuable comments and feedback. Below we respond to the concerns point by point.

---

> > ### Author Response · Authors · 2024-11-20
> > **Response to Weakness 1 (part 1/2)**
> >
> > **W1:** Comparison with LEGCN and MultiSetMixer
> >
> > Thank you for this insightful comment. The detailed discussion of line expansion (LEGCN) [a] and MultiSetMixer [b] is provided in Appendix F in our original manuscript. We have revised the updated version to make the discussion of these two methods more explicit in the main text. **Both methods model interactions as an aggregation function, which produces the same output for different elements.** This design cannot solve the three main limitations and leads to significantly reduced performance compared to our multi-output design, as demonstrated in our ablation experiments in **Table 3**. Apart from that, these two methods suffer from some other weaknesses. Below we explain the key weaknesses of these two methods compared to the proposed CoNHD method in detail and provide empirical results.
> >
> > **(1) Weaknesses of LEGCN compared to our method**
> >
> > LEGCN transforms the hypergraph into a line expansion graph and performs GCN on the transformed graph. To avoid confusion, we use "vertices" to refer to the new nodes in the line expansion graph. We take a hypergraph with two hyperedges, i.e., $e_1 = (v_1, v_2, v_3)$ and $e_2 = (v_1, v_2, v_4)$, as an example, where the vertices are denoted as $u_{v_1, e_1}, \ldots, u_{v_4, e_2}$, with the corresponding representation vectors $h_{v_1, e_1}, \ldots, h_{v_4, e_2}$. LEGCN poses the following three weaknesses.
> >
> > 1. **LEGCN does not differentiate the two types of interactions between vertices.**
> >  - As indicated in [a], LEGCN treats the line expansion graph as a homogeneous graph. However, the expansion graph should be a heterogeneous graph with two different edge types. For example, the type of the new edge between vertices $u_{v_1, e_1}$ and $u_{v_1, e_2}$, should be different from the type of the new edge between vertices $u_{v_1, e_1}$ and $u_{v_2, e_1}$, which depends on whether the shared part is the node or edge in the original hypergraph. However, LEGCN does not differentiate these two types of new edges and use the same weights to process them. In contrast, our method clearly separates the within-edge and within-node processes and can utilize such heterogeneity.
> > 2. **In each hyperedge, LEGCN only generates the same message to each vertex.**
> > - LEGCN utilizes a simple summation, i.e., $h_{(v_1, e_1)} + h_{(v_2, e_1)} + h_{(v_3, e_1)}$, to aggregate features of the vertices in hyperedge $e_1$. This aggregated message is the same for each vertex in this hyperedge. While in our method, we carefully implement two multi-input multi-output functions $\phi$ and $\varphi$ as set permutation equivariant functions, which can generate different outputs for different elements. In Section 5.3 of our manuscript, we demonstrate that this multi-output design, which can generate different outputs instead of the same output for different elements, is critical to the performance improvement.
> > 3. **High computational complexity of LEGCN.**
> > - LEGCN treats the line expansion result as a graph and performs GCN on this graph. However, it overlooks the hyperedge structure and introduces a lot of duplicated computation. For example, the input messages from neighboring vertices $(u_{v_1, e_1}, u_{v_2, e_1}, u_{v_3, e_1})$ to $u_{v_1, e_1}$ and $u_{v_2, e_1}$ are calculated separately. If the hyperedge $e_1$ contains $n$ nodes, corresponding to $n$ vertices, then the computational complexity is $\mathcal{O}(n^2)$ (each vertex needs $\mathcal{O}(n)$). This high complexity hinders the practical application of LEGCN in real-world scenarios. In contrast, the multiple outputs ($n$ outputs) in our method are generated together by a single function $\phi$ or $\varphi$ with $\mathcal{O}(n)$ time complexity.

---

> > > ### Author Response · Authors · 2024-11-20
> > > **Response to Weakness 1 (part 2/2)**
> > >
> > > **(2) Weaknesses of MultiSetMixer compared to our method**
> > >
> > > MultiSetMixer first aggregates edge-dependent node representations, e.g., $(h_{v_1, e_1}, h_{v_2, e_1}, h_{v_3, e_1})$, to update edge representation, e.g., $z_{e_1}$, and then propagates the edge representation back to update the edge-dependent node representations. Different from our method, it preserves the edge representations and poses the following two weaknesses.
> > >
> > > 1. **MultiSetMixer lacks within-node interactions.**
> > > - MultiSetMixer only models the interaction among representations of vertices in each hyperedge, i.e., the interactions among $(h_{v_1, e_1}, h_{v_2, e_1}, h_{v_3, e_1})$. However, it overlooks the within-node interactions, i.e., the interactions among $(h_{v_1, e_1}, h_{v_1, e_2}, \ldots)$, which are different representations of the same node. **More importantly**, this implementation only allows information flow within the same hyperedge and hinders the information flow across different hyperedges. For example, the information of $h_{v_1, e_1}$ cannot propagate to update $h_{v_1, e_2}$ or any representations in other hyperedges, even with many convolution layers. In contrast, our method incorporates both within-edge and within-node interactions, which allows information flow within the same hyperedge and across different hyperedges.
> > > 2. **MultiSetMixer still suffers from the three main limitations.**
> > > - As MultiSetMixer aggregates messages from nodes into a single edge representation $z_{e_1}$, this aggregation process can still cause potential information loss for large-degree edges. Also, as the aggregated edge representation $z_{e_1}$ cannot differentiate specific information for different nodes, it can only pass the same message back to the nodes instead of adaptive messages. Additionally, the messages are propagated between nodes and edges, without direct interactions among nodes or among edges. In contrast, our method with co-representations and the multi-output design can avoid this unnecessary aggregation process and address the three limitations.
> > >
> > >
> > > **(3) Performance of LEGCN and MultiSetMixer on the ENC task**
> > >
> > > We did not include LEGCN and MultiSetMixer in our main results due to the following reasons. First, the implementation of LEGCN is based on full-batch training with very high memory costs, leading to the "Out-of-Memory" problem when performing the ENC experiments. Second, the original MultiSetMixer paper does not provide source code for their implementation. We re-implement LEGCN and MultiSetMixer under the mini-batch training setting, which is consistent with the setting of our method. We conduct experiments on all ten ENC datasets. The results have been added to Table 1 in our revised manuscript.
> > >
> > > - *Micro-F1 results:*
> > >
> > > |Method|Email-Enron|Email-Eu|Stack-Biology|Stack-Physics|Coauth-DBLP|Coauth-AMiner|Cora-Outsider|DBLP-Outsider|Citeseer-Outsider|Pubmed-Outsider|
> > > |---|---|---|---|---|---|---|---|---|---|---|
> > > |LEGCN|0.783±0.001|0.639±0.001|0.668±0.002|0.701±0.003|0.499±0.003|0.520±0.002|0.698±0.008|0.676±0.016|0.733±0.015|0.703±0.002|
> > > |MultiSetMixer|0.818±0.001|0.670±0.001|0.709±0.001|0.754±0.001|0.559±0.001|0.593±0.005|0.542±0.013|0.561±0.004|0.706±0.007|0.668±0.001|
> > > |CoNHD (*ours*)|**0.911±0.001**|**0.709±0.001**|**0.749±0.002**|**0.777±0.001**|**0.619±0.002**|**0.650±0.003**|**0.800±0.019**|**0.903±0.002**|**0.828±0.010**|**0.899±0.004**|
> > >
> > > - *Macro-F1 results:*
> > >
> > > |Method|Email-Enron|Email-Eu|Stack-Biology|Stack-Physics|Coauth-DBLP|Coauth-AMiner|Cora-Outsider|DBLP-Outsider|Citeseer-Outsider|Pubmed-Outsider|
> > > |---|---|---|---|---|---|---|---|---|---|---|
> > > |LEGCN|0.728±0.007|0.535±0.004|0.572±0.006|0.575±0.018|0.490±0.002|0.511±0.003|0.689±0.008|0.675±0.016|0.731±0.016|0.698±0.002|
> > > |MultiSetMixer|0.755±0.005|0.636±0.005|0.643±0.003|0.679±0.004|0.554±0.001|0.585±0.005|0.538±0.011|0.552±0.003|0.705±0.007|0.666±0.001|
> > > |CoNHD (*ours*)|**0.871±0.002**|**0.690±0.002**|**0.695±0.004**|**0.710±0.004**|**0.604±0.003**|**0.646±0.004**|**0.797±0.020**|**0.902±0.002**|**0.826±0.010**|**0.898±0.004**|
> > >
> > > CoNHD demonstrates superior performance on all ENC datasets compared to LEGCN and MultiSetMixer.

---

> ### Author Response · Authors · 2024-11-20
> **Response to Weakness 2**
>
> **W2:** Motivation for studying the ENC task
>
> Thanks for this thoughtful question. We would like to emphasize the importance of the ENC task, and then discuss why our method is specifically suited for this task, although it can also be applied for solving other tasks.
>
> **(1) ENC is an interesting and important topic for both hypergraph research and applications.**
>
> For hypergraph research, most existing works focus on node-level or edge-level tasks. The ENC task, which relies on node-edge pairs, is under-explored. Additionally, as demonstrated by recent research [c], on the traditional node classification task, simply applying an MLP on the node features without considering the hypergraph structure can also achieve competitive results on some datasets. In contrast, in the ENC task, as the node can have different labels across different hyperedges, it forces the model to utilize the hypergraph structure and learn different features of the same node specific to different hyperedges. As indicated in [d], **the ENC task "evaluates the capability of models in capturing features unique to hypergraphs"**.
>
> For applications, as shown in [d], **the ENC task is very helpful in many real-world scenarios and downstream tasks**. For example, in the game industry, the ENC task can be used to model the score of each player (node) in each match (hyperedge). In electronic commerce, the ENC task can be used to predict the counts of each product (node) in each basket (hyperedge). Moreover, the ENC labels predicted by the model can be used as additional features to improve downstream tasks [d], including ranking aggregation, node-clustering, product-return prediction. We also provide experimental results on these downstream tasks in Appendix I.1 following [d], which demonstrates that the predicted ENC labels can help improve the performance on these downstream tasks.
>
> Therefore, we believe the ENC task is a valuable topic to study. We have revised our manuscript, especially the introduction part, to better highlight the significance of the ENC task.
>
> **(2) Why our method is specifically suited for the ENC task and its applicability to other tasks**
>
> The primary difference between the ENC task and other tasks (node-level and edge-level tasks) is that a node can have multiple labels across different hyperedges. Therefore, our proposed co-representation strategy is naturally suited for predicting these labels related to node-edge pairs. Meanwhile, our method can be extended to solve other tasks. For example, we can aggregate co-representations corresponding to the same node to generate a single node representations, i.e., $h_{v_1} = f(h_{v_1, e_1}, h_{v_1, e_2}, \ldots, h_{v_1, e_n})$, and use this node representation to perform traditional node classification. However, **this violates our design principle of avoiding aggregation**, which can cause non-adaptive representation size and potential information loss. Additionally, in traditional node classification, as a node only has one label, the multiple co-representations of a single node may not provide huge improvement on such a task. Nevertheless, we have conducted experiments on traditional node classification tasks in **Appendix I.2** in our original manuscript, which is the most common task in existing hypergraph research. The results demonstrate the superior performance and potential of our method beyond the ENC task.

---

> > ### Author Response · Authors · 2024-11-20
> > **Response to Weakness 3**
> >
> > **W3:** Time complexity and space complexity
> >
> > **Time complexity:** In Appendix D.1 in our original manuscript, we have provided the time complexity analysis of our method. The overall time complexity is $\mathcal{O}(Ld^2 \sum_{e \in \mathcal{E}} d_e)$. Here $L$ is the number of convolution layers, and $d$ is the embedding dimension. $d_e$ is the edge degree of hyperedge $e$. Therefore, the total complexity is linear to the number of node-edge pairs in the input hypergraph, i.e., $\sum_{e \in E} d_e$. **This is the same as other HGNN methods.** Intuitively, all HGNN methods need to calculate the interaction of nodes in a hyperedge, which is $\mathcal{O}(d_e)$, therefore, the total complexity is linear to the summation of $\mathcal{O}(d_e)$ for all the edges, i.e., $\sum_{e \in \mathcal{E}} d_e$. The difference is that, in other methods, they use the same node representation for the calculation in different hyperedges, while our method can use different node-edge co-representations for the calculation in different hyperedges. The time complexity of existing HGNNs and our method should be on the same scale.
> >
> > **Space complexity:** Thank you for this valuable comment. We have added the space complexity analysis in Appendix D.1 to make our manuscript more complete. The overall space complexity is $\mathcal{O}( Ld ( d + \sum_{e \in \mathcal{E}} d_e) )$, which is also linear to the number of node-edge pairs in the input hypergraph, i.e., $\sum_{e \in \mathcal{E}} d_e$. **This is the same as those edge-dependent message passing-based HGNNs**, including the best baseline WHATsNet in the ENC experiments. Intuitively, these edge-dependent methods also need to generate multiple edge-dependent node representations for each node in the computation process. The number of these generated edge-dependent node representations are the same as the number of node-edge pairs.

---

> > > ### Author Response · Authors · 2024-11-20
> > > **References and Additional Remarks**
> > >
> > > **References**
> > >
> > > [a] Yang et al. Semi-supervised hypergraph node classification on hypergraph line expansion. In *CIKM*, 2022.
> > >
> > > [b] Telyatnikov et al. Hypergraph neural networks through the lens of message passing: a common perspective to homophily and architecture design. *arXiv:2310.07684*, 2023.
> > >
> > > [c] Tang et al. Hypergraph-MLP: Learning on hypergraphs without message passing. In *ICASSP*, 2024.
> > >
> > > [d] Choe et al. Classification of edge-dependent labels of nodes in hypergraphs. In *KDD*, 2023.
> > >
> > > ---
> > >
> > > We hope the responses solve your concern and questions, and we are happy to discuss them if you need further clarification. Thank you!

---

> ### Author Response · Authors · 2024-11-29
>
> Dear Reviewer JdfB,
>
> Thank you for your valuable time and effort in reviewing our work. We have made an effort to address the concerns and questions you raised. Specifically, we have added discussions about the weaknesses of two other baselines as well as empirical results. We also discussed the applicability of our method to other tasks with experimental support, and included a detailed complexity analysis. We sincerely hope that our responses have sufficiently addressed your concerns.
>
> Your support is very important to our work, and we would greatly appreciate it if you could kindly consider raising your score. If you have any additional questions, we would be willing to have a further discussion on them. Thank you once again for your insightful feedback and thoughtful review.
>
> Best regards,
>
> Authors of Submission 5147

---

> > ### Author Response · Authors · 2024-12-02
> > **Gentle Reminder: Discussion Deadline Approaching**
> >
> > Dear Reviewer JdfB,
> >
> > As the discussion period is nearing its conclusion, we would like to confirm whether our responses have sufficiently addressed your questions and concerns. If there are any additional questions you would like to discuss, we would be happy to engage further.
> >
> > Thank you for your valuable feedback and efforts in improving the quality of our manuscript.
> >
> > Best regards,
> >
> > Authors of Submission 5147

---

### Official Review · Reviewer_wbP8 · 2024-11-08

**Soundness:** 3
**Presentation:** 3
**Contribution:** 2
**Rating:** 5
**Confidence:** 4

**Summary:**

The paper proposes a new hypergraph diffusion-based neural network for edge-dependent node classification (ENC). The paper's main contribution is extending hypergraph diffusion using node-hyperedge co-representations to make hypergraph diffusion better to handle the ENC.

**Strengths:**

The authors propose a new approach to solving the ENC problem. The whole process sounds technical. Experiments show that the proposed method is highly competitive.

**Weaknesses:**

The contribution and novelty of the proposed methodology are limited. The paper applies hypergraph diffusion proposed by the previous method EG-GNN. The most significant difference lay in the input and output (single -> multi). The multi-input and multi-output are also explored in the previous message-passing works[1]. Actually, the multi-output or multi-input function can not be an essential difference. e.g. you can concatenate multiple inputs to be a single input, and then use a single input function to process it (e.g. f(x,y) = g([x,y])).  So, combining the edge and node representation into the EG-GNN to improve ENC seems trivial.

The author should add a reference for problem 1.

minor: figure 3, UNP- > UNB

[1]https://arxiv.org/pdf/2102.09844, sec 3.1

**Questions:**

Further, I am still concerned about the setting of this paper. In my understanding, the edge-dependence node classification problem can be equivalent to the equivalent star-expansion graph edge prediction problem. In this real-world problem, can it be directly modeled as a graph instead of a hypergraph, and directly use the graph neural network to process it? if so,  can you analyze the pros and cons of each approach?

---

> ### Author Response · Authors · 2024-11-20
> **Response to Reviewer wbP8**
>
> We thank Reviewer wbP8 for the thoughtful comments and feedback. Below we respond to the concerns and questions point by point.

---

> ### Author Response · Authors · 2024-11-20
> **Response to Weakness 1 (part 1/2)**
>
> **W1:** The contributions and novelty of our method
>
> We respectfully think the reviewer might have some misunderstandings about our method and its contributions. Below we clarify the main points and highlight the value of our work.
>
> **(1) The multi-input single-output design mentioned in our paper is specific to HGNNs, not GNNs**
>
> In our paper, the single-output design is used to describe the process of aggregating messages from multiple nodes to a single hyperedge in HGNNs. This is a specific problem for HGNNs, not GNNs (**therefore it is not related to EGNN** [a] (the paper [1] mentioned by the reviewer)).
>
> In GNNs, each edge only connects to two nodes (e.g., $e_1 = (v_1, v_2)$), therefore the message from neighboring node $v_2$ to node $v_1$ through edge $e_1$ can always be calculated by a function with two inputs ($m_{v_2, v_1} = f(x_{v_2}, x_{v_1})$). (The EGNN paper mentioned by the reviewer also follows this design, where the implementation of $f(\cdot)$ can be found in Eq. 3 of [a])
>
> However, in HGNNs, each edge can connect any number of nodes (e.g., $e_1 = (v_1, v_2, \ldots, v_n)$) ($n$ can be different for different hyperedges). Therefore, when calculating the message from neighboring nodes ($v_2, \ldots, v_n$) to node $v_1$, the typical solution is to aggregate node information to update the edge representation $z_{e_1} = f(x_{v_1}, x_{v_2}, \ldots, x_{v_n})$, which then propagates back to the node $v_1$ (and also, to the other nodes). In this case, we can see the function $f(\cdot)$ needs to aggregate any number of inputs and generate a single output, which can cause potential information loss when $n$ is very large.
>
> In our method, we introduce co-representations to disseminate the single output $z_{e_1}$ to multiple outputs ($m_{v_1, e_1}, m_{v_2, e_1}, \ldots, m_{v_n, e_1}$). The function $f(\cdot)$ becomes $(m_{v_1, e_1}, m_{v_2, e_1}, \ldots, m_{v_n, e_1}) = f(h_{v_1, e_1}, h_{v_2, e_1}, …, h_{v_n, e_1})$, where each output $m_{v_i, e_1}$ can correspond to each input $h_{v_i, e_1}$. It means, even when $n$ is very large, the outputs of the function can also have $n$ outputs, which can naturally solve the potential information loss problem. Also, this design brings some other advantages like diverse diffusion information and direct interactions among nodes and among edges, which will be discussed below.

---

> ### Author Response · Authors · 2024-11-20
> **Response to Weakness 1 (part 2/2)**
>
> **(2) Why concatenation is not a solution for implementing multi-output functions in HGNNs**
>
> It is indeed possible to model a multi-input multi-output function by the following procedure: first concatenate multiple inputs into one single vector, and then apply a feed forward network to generate the output, finally split the single output to multiple vectors, i.e., $[o_1, o_2, \ldots, o_n] = f([x_1, x_2, …, x_n])$.
>
> However, this is not a solution for HGNNs: **First**, a feed forward network only accepts a fixed-length input, while in our case the number of inputs ($n$) can be different across different edges, which means the concatenated vector should have variable size. **Second**, this simple implementation cannot guarantee the set permutation equivariance property. For example, when permuting the inputs, like $[o'_2, o'_1, \ldots, o'_n] = f([x'_2, x'_1, \ldots, x'_n])$, we cannot guarantee that the output $o'_2$ is the same as $o_2$. More formally, we cannot guarantee that, given any element $\pi$ from the permutation group $\mathbb{S}_n$, the equality $f(\pi([x_1, x_2, \ldots, x_n])) = \pi(f([x_1, x_2, \ldots, x_n]))$ holds. This property is essential in HGNNs, as the nodes are inherently unordered, and the concatenation ordering of the node features should not affect the outputs and also the final predicted labels. Therefore, we carefully design the two multi-input multi-output functions $\phi$ and $\varphi$ as permutation equivariant set functions.
>
> **(3) Introducing co-representations to disseminate a single output into multiple outputs is a critical improvement of ED-HNN** [b]
>
> **First**, we respectfully believe it might not be justified to assess the contribution of a work as limited solely due to its simplicity. Actually, we always want a simple but effective solution, which can be easily implemented and achieve good performance in applications. Below, we demonstrate why the co-representation design in our method is a critical improvement of ED-HNN [b]. (Here we use the name "**ED-HNN**" as used in the original paper [b], instead of the name "**EG-GNN**" used by the reviewer. Please correct us if we have any misunderstandings.)
>
> Similar to other message passing-based HGNNs, ED-HNN models the within-edge and within-node interactions as a single-output aggregation process, which brings **three limitations**.
>
> -   **Non-adaptive representation size:** As discussed in (1), existing message passing-based HGNNs, including ED-HNN, model the within-edge interaction as an aggregation function $z_{e_1} = f(x_{v_1}, x_{v_2}, \ldots, x_{v_n})$. This can cause potential information loss when $n$ is very large.
>
> -   **Non-adaptive messages:** The edge representation $z_{e_1}$ cannot distinguish different node-specific information and therefore can only pass the same message $z_{e_1}$ to different nodes, instead of adaptive messages for different nodes.
>
> -   **Insufficient direct interactions among nodes and among edges**: Message passing-based HGNNs only model interactions between nodes and edges, without direct interactions among the nodes or direct interactions among the edges.
>
> To solve all these three limitations, we carefully design co-representations to disseminate the single output $z_{e_1}$ to multiple outputs $(m_{v_1, e_1}, m_{v_2, e_1}, \ldots, m_{v_n, e_1})$, which can solve the above three limitations in a unified framework. Additionally, we demonstrate that the CoNHD framework is more expressive than message passing-based HGNNs including ED-HNN (see Proposition 3 in our manuscript). As demonstrated in Table 1 and Figure 3, our method also significantly outperformed other baselines, including ED-HNN, on all datasets without sacrificing the efficiency, which proves the superiority of our improvement.

---

> ### Author Response · Authors · 2024-11-20
> **Response to Question 1**
>
> **Q1:** Comparison to GNNs on the star-expansion graph
>
> It is indeed an option to model the ENC problem as the edge prediction problem on the star-expansion graph, where the nodes and edges in the original hypergraph are treated as vertices in this new graph (to avoid confusion with the nodes in the hypergraph, we use "vertices" to refer to the new nodes in the expansion graph). However, this solution suffers from the following weaknesses.
>
> **(1) Overlooking the heterogeneity in hypergraphs.**
>
> Performing traditional GNNs on the star-expansion graph treats all the vertices as the same type. However, the vertices can be divided into two types: from node and from hyperedge in the original hypergraph. Processing these two types of vertices in the same overlooks the heterogeneity and affects the performance. In contrast, our method clearly separates the within-edge and within-node interactions, which can leverage such heterogeneity and improve the performance.
>
> **(2) Still suffering from the three main limitations.**
>
> This graph-based solution still relies on single node and edge representations, and suffers from the three main limitations discussed in our paper, including non-adaptive representation size, non-adaptive messages, and insufficient interactions among nodes or edges, which are addressed in our method.
>
> We have performed experiments with GNNs on the star-expansion graph and reported the performance in **Table 1** of our original manuscript. **The proposed CoNHD method significantly outperforms the GNN-based methods.**

---

> > ### Author Response · Authors · 2024-11-20
> > **Response to the Minor Issues**
> >
> > **Minor Corrections**
> >
> > Thank you for the detailed corrections. We have added the reference to Problem 1 and changed the text in Figure 3 accordingly.

---

> ### Author Response · Authors · 2024-11-20
> **References and Additional Remarks**
>
> **References**
>
> [a] Satorras et al. E(n) equivariant graph neural networks. In *ICML*, 2021.
>
> [b] Wang et al. Equivariant hypergraph diffusion neural operators. In *ICLR*, 2023.
>
> ---
>
> We hope the responses solve your concern and questions, and we are happy to discuss them if you need further clarification. Thank you!

---

> ### Author Response · Authors · 2024-11-29
>
> Dear Reviewer wbP8,
>
> Thank you for your valuable time and effort in reviewing our work. We have made an effort to address the concerns and questions you raised. In particular, we have clarified the novelty and contributions of our method compared to prior approaches and provided an analysis of the limitations in graph-based solutions. We sincerely hope that our responses have adequately resolved your concerns.
>
> Your support is very important to our paper. We would greatly appreciate it if you could kindly consider raising your score. Should you have any additional questions, we would be more than happy to engage in further discussions. Thank you once again for your thoughtful review and valuable suggestions.
>
> Best regards,
>
> Authors of Submission 5147

---

> > ### Author Response · Authors · 2024-12-02
> > **Gentle Reminder: Discussion Deadline Approaching**
> >
> > Dear Reviewer wbP8,
> >
> > As the discussion period is nearing its conclusion, we would like to confirm whether our responses have sufficiently addressed your questions and concerns. If there are any additional questions you would like to discuss, we would be happy to engage further.
> >
> > Thank you for your valuable feedback and efforts in improving the quality of our manuscript.
> >
> > Best regards,
> >
> > Authors of Submission 5147

---

### Author Response · Authors · 2024-11-20
**General Response**

We sincerely thank all the reviewers and the area chair for the time and effort in evaluating our manuscript. We are encouraged by the following positive remarks on our work:

- **Important Task:** The studied ENC task is interesting **[# Reviewer JdfB]**, important and under-explored **[# Reviewer JdfB/y4P5]**, and has many real-world applications **[# Reviewer y4P5]**.

- **Sound Solution:** The CoNHD method is technically sound **[# Reviewer wbP8/y4P5]** with improved expressiveness and innovative adaptive representation size mechanism **[# Reviewer y4P5]**.

- **Competitive Performance:** The proposed method demonstrates superior performance through comprehensive experiments **[# Reviewer wbP8/y4P5/WHeD]**.

- **Clarity and Readability:** The manuscript is well-written and easy to follow **[# Reviewer JdfB/WHeD]**.

---

We also deeply appreciate the constructive suggestions and insightful comments from the reviewers. We reply to individual reviewers to address their concerns and questions point by point. We have revised our manuscript accordingly, with the edits marked in $\textcolor{blue}{\text{blue}}$. Below we summarize the key revisions made to our updated manuscript:

-   Added baseline results for LEGCN and MultiSetMixer across ten ENC datasets. **[# Reviewer JdfB]**

-   Conducted ablation experiments to assess the effectiveness of direct interactions. **[# Reviewer WHeD]**

-   Incorporated space complexity analysis of the proposed CoNHD method. **[# Reviewer JdfB]**

-   Clarified the high-level intuition behind our method in Section 4. **[# Reviewer y4P5]**

-   Expanded discussion on the real-world significance of the ENC task. **[# Reviewer WHeD]**

-   Moved the ADMM part to the appendix. **[# Reviewer y4P5]**

-   Improved the overall clarity and readability, and corrected minor errors. **[# Reviewer y4P5/wbP8]**

To avoid confusion when referring to other sections of the original manuscript, most additional results and discussions have been placed at the end of the appendix. We will restructure the manuscript and further improve the readability in the final camera-ready version.

---

We hope our responses adequately address the concerns of the reviewers, and we are happy to have further discussions to address any additional questions. Thank you again for taking time to help improve the quality of our work!

---

### Meta-Review · Area_Chair_eZfR · 2024-12-21

**Metareview:**

This paper proposes a new method, CoNHD, that models both within-edge and within-node interactions as multi-input multi-output functions to tackle edge-dependent node classification tasks. The problem is interesting. The proposed CoNHD model outperforms baseline methods. The paper is well-organized and easy to follow. Several issues were raised by reviewers, including incremental novelty, a lack of rigorous guarantees to address the mentioned limitations, and unconvincing applicability to real-world applications. Reviewers are generally negative about this work.

**Additional Comments On Reviewer Discussion:**

After discussion, reviewers reach a consensus of rejection.

---

### Decision · Program_Chairs · 2025-01-22

Reject